# BEYOND IN-DOMAIN DETECTION: SPIKESCORE FOR CROSS-DOMAIN HALLUCINATION DETECTION

**Yongxin Deng**[1], **Zhen Fang**[1]*, **Sharon Li**[2], **Ling Chen**[1]
[1]University of Technology Sydney      [2]University of Wisconsin-Madison
yongxin.deng@student.uts.edu.au, sharonli@cs.wisc.edu,
{zhen.fang, ling.chen}@uts.edu.au

## ABSTRACT

Hallucination detection is critical for deploying large language models (LLMs) in real-world applications. Existing hallucination detection methods achieve strong performance when the training and test data come from the same domain, but they suffer from poor cross-domain generalization. In this paper, we study an important yet overlooked problem, termed *generalizable hallucination detection* (GHD), which aims to train hallucination detectors on data from a single domain while ensuring robust performance across diverse related domains. In studying GHD, we simulate multi-turn dialogues following LLMs' initial response and observe an interesting phenomenon: hallucination-initiated multi-turn dialogues universally exhibit larger uncertainty fluctuations than factual ones across different domains. Based on the phenomenon, we propose a new score *SpikeScore*, which quantifies abrupt fluctuations in multi-turn dialogues. Through both theoretical analysis and empirical validation, we demonstrate that SpikeScore achieves strong cross-domain separability between hallucinated and non-hallucinated responses. Experiments across multiple LLMs and benchmarks demonstrate that the SpikeScore-based detection method outperforms representative baselines in cross-domain generalization and surpasses advanced generalization-oriented methods, verifying the effectiveness of our method in cross-domain hallucination detection.

## 1 INTRODUCTION

Hallucination detection (Manakul et al., 2023; Farquhar et al., 2024) is crucial for the reliable deployment of large language models (LLMs) in real-world applications, particularly in safety-critical domains such as education (Harvey et al., 2025), healthcare (Roustan et al., 2025), and finance (Kang & Liu, 2023). This is because LLMs may produce factually incorrect or logically inconsistent outputs, collectively referred to as hallucinations (Huang et al., 2025), which can undermine user trust and lead to harmful consequences in high-stakes scenarios. To address this issue, hallucination detection has recently been widely studied (Chang et al., 2024; Minaee et al., 2024).

Existing detection methods generally fall into two categories: training-free methods (Janiak et al., 2025; Sun et al., 2025b) and training-based methods (Obeso et al., 2025; Wei et al., 2024). Representative training-free methods (Manakul et al., 2023; Farquhar et al., 2024) rely on intrinsic model signals such as uncertainty, consistency, or attention patterns, while typical training-based methods (Kossen et al., 2024; Wei et al., 2024) train lightweight feed-forward classifiers on hidden activations from specific layers to predict reliability. Training-based methods generally outperform training-free methods on in-domain test sets, and thus have become the mainstream direction.

Despite the notable progress achieved by training-based hallucination detection methods, these methods remain fundamentally limited by poor cross-domain generalization. Experiments in Zhang et al. (2025b) show that for training-based methods, such as SAPLMA (Azaria & Mitchell, 2023) and SEP (Kossen et al., 2024), detection performance drops sharply when the test domain differs from the training domain. This domain sensitivity stems from reliance on domain-specific features learned during training, making them brittle to distribution shifts. Thus, despite dominating current research, training-based methods fail to provide generalizable detection for practical applications.

---

*Corresponding author: Zhen Fang   Email: zhen.fang@uts.edu.au

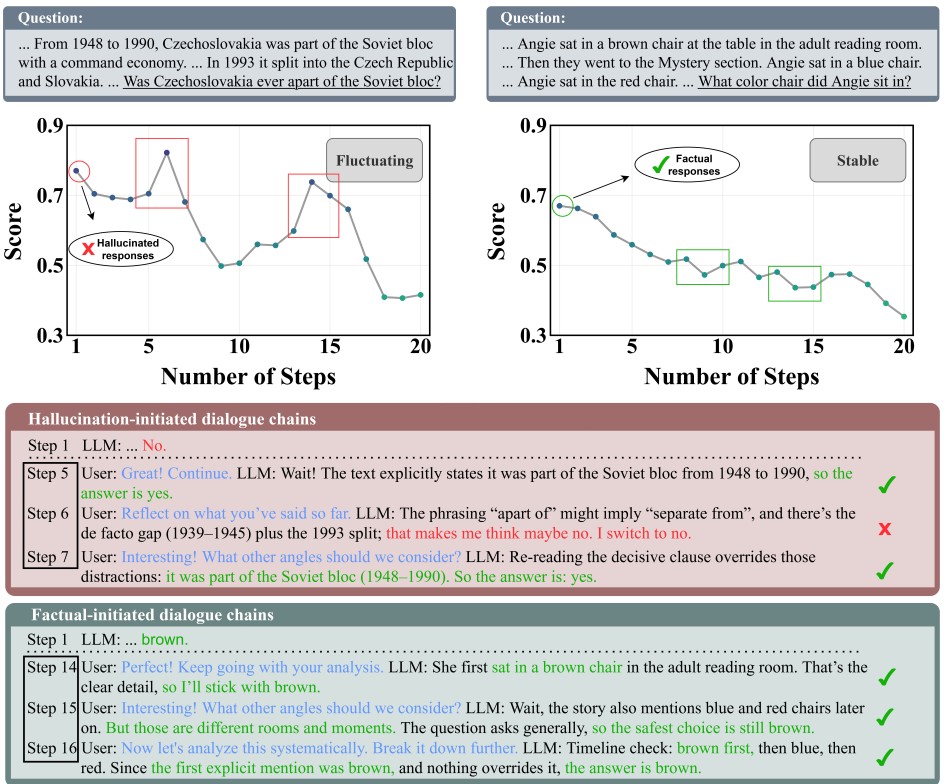

Figure 1: Case study of self-contradictions in multi-turn dialogues. SAPLMA scores (trained on Math, tested on CoQA—a representative cross-domain setup) with Llama-3.1-8B-Instruct reveal distinct trajectories: hallucination-initiated dialogues (left) display marked spikes in the uncertainty signal, in contrast to the relatively stable fluctuations of factual dialogues (right). Inspection of the dialogue content at spike locations shows that hallucinated responses oscillate between contradictory viewpoints and frequently reverse their stance (blue: prompts; red: errors; green: factual responses), whereas factual dialogues remain consistent with only minor self-corrections.

To mitigate the generalization issue in hallucination detection, we consider a more realistic yet challenging setting, termed *generalizable hallucination detection* (GHD), which aims to train hallucination detectors from a single domain while ensuring strong performance across diverse related domains. Prior work has also sought indicators that separate hallucinated from non-hallucinated responses (Azaria & Mitchell, 2023; Kulkarni et al., 2025; Farquhar et al., 2024; Xu et al., 2023), but such studies mainly address *challenge 1:* ensuring sufficient separability between hallucinated and non-hallucinated responses (within a single domain). In contrast, GHD additionally requires *challenge 2:* maintaining this separability consistently across different domains.

An intriguing phenomenon provides crucial inspiration: LLMs often exhibit self-contradiction in multi-turn conversations (Laban et al., 2025; Tosato et al., 2025), particularly when users shift their stance (Ranaldi & Pucci, 2023; Hong et al., 2025), leading models to accommodate the new position and contradict their previous conclusions. We hypothesize that this behavior represents an explicit manifestation of internal uncertainty or confidence instability. When the entire dialogue stems from hallucinated responses, such self-contradictory or stance-shifting behaviors should occur more frequently, as they often trigger the model's self-correction mechanism. Given the broad observation of this phenomenon, we are keen to explore its potential for identifying domain-invariant indicators.

To validate our intuition, we construct multi-turn dialogues by feeding the model's initial response back as context for subsequent questions. Specifically, we compile a diverse prompt library (see Appendix G) containing follow-up questions that naturally extend from the initial query, simulating realistic conversational scenarios where each turn builds upon the previous response. Case studies (Figure 1) reveal that when initial responses are hallucinated, models produce self-contradictory replies across dialogue turns, causing SAPLMA (Azaria & Mitchell, 2023) scores (a representative training-based method) to exhibit sharp spikes characterized by rapid rises followed by steep drops.

In contrast, such spikes are notably absent in dialogues initiated from factual responses. We expect this phenomenon extends beyond individual cases, potentially providing a universal solution to the GHD problem and guiding further methodological exploration. This leads us to investigate:

*Do hallucination-initiated multi-turn dialogues universally exhibit larger uncertainty fluctuations than factual ones across different domains?*

We design *SpikeScore*, defined as the maximum second-order difference of SAPLMA scores along the induced continuation, to quantify these abrupt fluctuations in multi-turn dialogue paths. This metric captures the sharpness of peaks in the score trajectory by measuring the intensity of rapid rise-and-fall patterns, where larger values indicate dramatic confidence reversals. Through extensive empirical studies, we statistically validate the consistent patterns of SpikeScore across different datasets. Building on these statistical observations, we further establish Theorem 1, which proves that when the mean and variance satisfy certain constraints, SpikeScore addresses both challenges: it achieves a probabilistic lower bound for separability between hallucinated and non-hallucinated chains (challenge 1), and our analysis suggests that this bound may extend across domains under the examined conditions (challenge 2). In other words, our results indicate that SpikeScore can serve as an effective indicator for distinguishing hallucinated from factual responses, supported by statistical evidence and theoretical bound observed across multiple domains.

Extensive experiments are conducted on four LLMs (i.e., Llama-3.2-3B/3.1-8B, Qwen3-8B/14B), and six benchmarks (i.e., TriviaQA (Joshi et al., 2017), CommonsenseQA (Talmor et al., 2019), Belebele (Costa-Jussà et al., 2025), CoQA (Reddy et al., 2019), Math (Hendrycks et al., 2021), and SVAMP (Patel et al., 2021)) covering commonsense, knowledge-intensive, conversational, and mathematical reasoning. Empirical results in Section 4.2 demonstrate that the SpikeScore-based detection method consistently outperforms representative detection methods in cross-domain generalization and surpasses advanced generalization-oriented methods, e.g., PRISM (Zhang et al., 2025b) and ICR Probe (Zhang et al., 2025c). Our main contributions are summarized as follows:

• We exploit the intrinsic instability of LLMs in multi-turn dialogue by inducing post-answer continuations, yielding standardized trajectories for analysis.

• We introduce SpikeScore, a domain-invariant instability indicator that enables threshold-based hallucination detection, and theoretically prove a lower bound on its separability.

• Extensive experiments demonstrate that SpikeScore consistently achieves superior cross-domain performance compared to strong baselines across multiple models and datasets.

## 2 LEARNING SETUPS

**LLMs and Token Sequences.** Following Du et al. (2024); Oh et al. (2025), we use a distribution $\mathbb{P}_{\boldsymbol{\theta}}(\cdot)$ over token sequences to define LLM, where $\boldsymbol{\theta}$ is the model parameters. Given a token sequence $\mathbf{Q} = [x_1, \ldots, x_k]$ representing the question, where each $x_j$ is the $j$-th token in the sequence. $\mathbb{P}_{\boldsymbol{\theta}}(\cdot)$ generates an answer $\mathbf{A} = [x_{k+1}, \ldots, x_{k+l}]$ by predicting each token based on the preceding context:

$$\mathbb{P}_{\boldsymbol{\theta}}(x_j \mid x_1, \ldots, x_{j-1}), \text{ for } j = k+1, \ldots, k+l. \tag{1}$$

**Domains and Datasets.** Let $\mathcal{Q}$ and $\mathcal{A}$ be the spaces of questions and answers, respectively. We consider a *ground-truth domain* $\mathbb{P}_{Q,T}$, a joint distribution over $\mathcal{Q} \times \mathcal{A}$, where $Q$ and $T$ are the question and truthful-answer random variables, respectively.

Given $\mathbb{P}_{Q,T}$, each sample consists of a question $\mathbf{Q}$, a reference answer $\mathbf{A}_{\text{ref}}$, and optionally a context passage (if provided by the training dataset), where $\mathbf{A}_{\text{ref}} \sim \mathbb{P}_{T|Q}(\cdot \mid \mathbf{Q})$, here $\mathbb{P}_{T|Q}$ is the conditional distribution of $\mathbb{P}_{Q,T}$. For simplicity, we concatenate the context and the question into a single input sequence, which we denote as $\mathbf{Q}$. The dataset from $\mathbb{P}_{Q,T}$ can then be represented as $\mathcal{D} = \{(\mathbf{Q}_1, \mathbf{A}_{\text{ref}_1}), \ldots, (\mathbf{Q}_n, \mathbf{A}_{\text{ref}_n})\}$, where $n$ is the size of samples.

Given a question $\mathbf{Q} \sim \mathbb{P}_Q$, the LLM $\mathbb{P}_{\boldsymbol{\theta}}(\cdot)$ generates an answer $\mathbf{A}$, i.e., $\mathbf{A} \sim \mathbb{P}_{\boldsymbol{\theta}}(\cdot|\mathbf{Q})$. Each generated answer $\mathbf{A}$ is then assigned a binary label $y \in \{0, 1\}$ according to its semantic consistency with the reference answer $\mathbf{A}_{\text{ref}}$. Specifically, if $\mathbf{A}$ aligns with $\mathbf{A}_{\text{ref}}$, it is labeled as truthful (i.e., $y = 0$); otherwise, it is labeled as hallucinated (i.e., $y = 1$). The labeled dataset $\mathcal{D}_l$ is defined as:

$$\mathcal{D}_l = \{(\mathbf{Q}_1, \mathbf{A}_1, y_1), \ldots, (\mathbf{Q}_n, \mathbf{A}_n, y_n)\}. \tag{2}$$

**Generalizable Hallucination Detection.** In GHD, we consider a training domain $\mathbb{P}_{Q,T}$ and $N$ related test domains $\mathbb{P}^1_{Q,T}, \ldots, \mathbb{P}^N_{Q,T}$. The goal of GHD is to train a detector on $\mathbb{P}_{Q,T}$ that generalizes well to the $N$ test domains. We formally introduce the definition of GHD below.

**Problem 1 (Generalizable Hallucination Detection.)** *Given a training dataset constructed from the training domain $\mathbb{P}_{Q,T}$ together with a given LLM $\mathbb{P}_{\boldsymbol{\theta}}$, i.e.,*

$$\mathcal{D}_l = \{(\mathbf{Q}_1, \mathbf{A}_1, y_1), \ldots, (\mathbf{Q}_n, \mathbf{A}_n, y_n)\}, \tag{3}$$

*as introduced in Eq. (2), the objective of generalizable hallucination detection (GHD) is to learn a detector $D$ based on the LLM $\mathbb{P}_{\boldsymbol{\theta}}(\cdot)$ and the training dataset $\mathcal{D}_l$. The detector is required to generalize across $N$ test domains $\mathbb{P}^1_{Q,T}, \ldots, \mathbb{P}^N_{Q,T}$, i.e., for any question–answer pair $(\mathbf{Q}, \mathbf{A})$,*

$$D(\mathbf{Q}, \mathbf{A}) = \begin{cases} 0, & \text{if } (\mathbf{Q}, \mathbf{A}) \sim \mathbb{P}^i_{Q,T}, \text{ for some } i \in \{1, ..., N\}, \\ 1, & \text{otherwise.} \end{cases} \tag{4}$$

Note that when $N = 1$ and $\mathbb{P}^1_{Q,T} = \mathbb{P}_{Q,T}$, GHD reduces to the classical hallucination detection.

Due to space constraints, the related work is discussed in Appendix B.

## 3 METHODOLOGY

### 3.1 EXPLORING CROSS-DOMAIN INVARIANT INDICATORS

Recent studies (Deshpande et al., 2025; Zhang et al., 2025a; Laban et al., 2025) have revealed a compelling phenomenon in multi-turn dialogues: *when users repeatedly probe or challenge LLM responses, models often rapidly abandon their initial positions and defer to user suggestions, exhibiting marked self-contradiction.* This behavior is remarkably consistent across diverse datasets, model families, and parameter scales, manifesting robustly regardless of conversational context.

This pattern is broadly consistent with evidence that self-contradictory behavior reflects underlying uncertainty in generated content (Yoon et al., 2025; Liu et al., 2024a). Building on this insight, we hypothesize that the *degree* of response instability in follow-up dialogues correlates with the factuality of the initial response. While all LLM responses may exhibit some level of self-contradiction when challenged, we expect hallucinated responses to demonstrate significantly higher instability. In contrast, factually grounded responses, while not immune to revision, should exhibit greater consistency when probed. Furthermore, given LLMs' self-correction mechanisms (Wang et al., 2024; Lee et al., 2025; Zhao et al., 2025; Tian et al., 2024), initially erroneous responses may trigger more dramatic shifts and sharper deviations as the model actively attempts to rectify its mistakes.

To validate our hypothesis, we construct a simulated dialogue environment to examine response dynamics across multiple turns. In our hallucination detection scenario, we treat the model's initial response to a question as the starting point of the dialogue, with subsequent responses termed as continuation answers. The process of generating continuation answers is formulated as follows:

$$\mathbf{A}^k \sim \mathbb{P}_{\boldsymbol{\theta}}(\cdot | \mathbf{Q}, \mathbf{A}^1, \mathbf{P}^2, \mathbf{A}^2, ..., \mathbf{P}^i, \mathbf{A}^i, ..., \mathbf{P}^k), \text{ for any } k > 1, \tag{5}$$

where $\mathbf{A}^1$ is the original answer $\mathbf{A}$, $\mathbf{A}^i$ is the $i$-th continuation answer ($i > 1$), and $\mathbf{P}^i$ is the $i$-th prompt to induce the generation of $i$-th continuation answer $\mathbf{A}^i$ for any $i > 1$. The design of the prompt $\mathbf{P}^i$, motivated by Chen et al. (2025) and Laban et al. (2025), is to induce contextually coherent responses given the preceding $i$ dialogue turns. This formulation enables us to simulate realistic multi-turn interactions. We provide the complete prompt specification in Appendix G.

**Training Score for Uncertainty Estimation.** To effectively evaluate the uncertainty of each continuation answer $\mathbf{A}^k$, we need to select suitable scoring methods. To further ensure that the scores remain relatively accurate in addressing hallucination detection, we adopt training-based methods that leverage information of training data tailored for hallucination detection.

In this work, we use one of the most representative methods in hallucination detection, termed SAPLMA (Azaria & Mitchell, 2023), which utilizes LLM's internal state to reveal the truthfulness of a given answer. The details of SAPLMA is introduced as follows. Given the internal representation $\mathbf{E}_{\boldsymbol{\theta}}(\cdot)$ of the LLM $\mathbb{P}_{\boldsymbol{\theta}}(\cdot)$, we place a multilayer perceptron (MLP) followed by a sigmoid activation

on top of $\mathbf{E}_{\boldsymbol{\theta}}(\cdot)$ to produce a probabilistic output $p_{\mathbf{W}}(\cdot) \in [0, 1]$. The parameters $\mathbf{W}$ are then optimized using the cross-entropy loss over the training data $\mathcal{D}_l = \{(\mathbf{Q}_i, \mathbf{A}_i, y_i)\}_{i=1}^n$, i.e.,

$$\widehat{\mathbf{W}} \in \arg\min_{\mathbf{W}} \mathcal{L}(\mathbf{W}; \mathcal{D}_l) = -\frac{1}{n} \sum_{i=1}^n \Big( y_i \log p_{\mathbf{W}}(\mathbf{E}_{\boldsymbol{\theta}}(\mathbf{A}_i|\mathbf{Q}_i)) \tag{6}$$
$$+ (1 - y_i) \log \big(1 - p_{\mathbf{W}}(\mathbf{E}_{\boldsymbol{\theta}}(\mathbf{A}_i|\mathbf{Q}_i))\big)\Big).$$

Next, to evaluate the uncertainty of a continuation answer $\mathbf{A}^k$, we use the following formulation:

$$S(\mathbf{A}^k; \mathbf{Q}, \mathbf{A}) = p_{\widehat{\mathbf{W}}}(\mathbf{E}_{\boldsymbol{\theta}}(\mathbf{A}^k|\mathbf{Q}, \mathbf{A}^1, \mathbf{P}^2, \mathbf{A}^2, ..., \mathbf{P}^i, \mathbf{A}^i, ..., \mathbf{P}^k)). \tag{7}$$

Finally, for each $\mathbf{Q}$ and $\mathbf{A}$, we obtain a score sequence:

$$\mathbf{S}(\mathbf{Q}, \mathbf{A}) = \big[S(\mathbf{A}^1; \mathbf{Q}, \mathbf{A}), S(\mathbf{A}^2; \mathbf{Q}, \mathbf{A}), \ldots, S(\mathbf{A}^K; \mathbf{Q}, \mathbf{A})\big]. \tag{8}$$

where $K$ is the stopping step. In our experiment, we set $K = 20$ as the default.

**Remark.** *We also evaluate other representative training-based scoring method SEP (Kossen et al., 2024), as well as several training-free methods, including Perplexity (Ji et al., 2023), Reasoning score (Sun et al., 2025a), and In-Context Sharpness (Chen et al., 2024b). These methods yield conclusions consistent with those of SAPLMA. Moreover, we observe that integrating our SpikeScore with SAPLMA or SEP achieves substantially better overall performance than training-free methods, showing the clear advantage of training-based methods. Please see Section 4.2.*

**SpikeScore: Estimating Score Sequence Fluctuations.** There are many methods to evaluate the sequence fluctuations, such as computing the variance or extreme gap. It is worth noting that Sun et al. (2025a) proposes the coefficient of variation score, which measures the score sequence fluctuations using the coefficient of variation, i.e., the standard deviation normalized by the expectation.

The coefficient of variation score primarily captures the overall fluctuation of a sequence. However, focusing solely on global variation may overlook fine-grained differences (Fouladgar et al., 2022). Measuring local fluctuations offers a more sensitive and discriminative characterization of sequence fluctuations (Jennings et al., 2004). Motivated by Qian et al. (2025), we propose to measure local fluctuations using the maximum second-order difference, i.e., given a score sequence $\mathbf{S}(\mathbf{Q}, \mathbf{A})$,

$$\mathrm{Max}|\Delta^2|(\mathbf{S}(\mathbf{Q}, \mathbf{A})) = \max_{1 < k < K-1} |S(\mathbf{A}^{k+1}; \mathbf{Q}, \mathbf{A}) - 2S(\mathbf{A}^k; \mathbf{Q}, \mathbf{A}) + S(\mathbf{A}^{k-1}; \mathbf{Q}, \mathbf{A})|. \tag{9}$$

$\mathrm{Max}|\Delta^2|$ captures the maximum curvature of the score sequence $\mathbf{S}(\mathbf{Q}, \mathbf{A})$, corresponding to the most pronounced local fluctuation. In other words, the maximum second-order difference quantifies the point of greatest irregularity or instability within the score sequence $\mathbf{S}(\mathbf{Q}, \mathbf{A})$. In this work, we call the maximum second-order difference $\mathrm{Max}|\Delta^2|$ as *SpikeScore*.

**Remark.** *We further compare SpikeScore with the coefficient of variation score. Empirically, SpikeScore outperforms the coefficient of variation score. Details are provided in Appendix D.3.*

## 3.2 EVALUATIONS OF SPIKESCORE

In this part, we conduct experiments to examine the invariance of SpikeScore across hallucinated and non-hallucinated answers in different domains. We use six datasets: TriviaQA (Joshi et al., 2017), CommonsenseQA (Talmor et al., 2019), Belebele (Costa-Jussà et al., 2025), CoQA (Reddy et al., 2019), Math (Hendrycks et al., 2021), and SVAMP (Patel et al., 2021). In each group of experiments, one dataset is used for training and the remaining five are used for testing, yielding six groups of experiments in total (see Appendix C for details). The experimental results are summarized in Figures 2 and 3, from which we highlight key observations.

**Observation 1 (Expectation Invariance.)** *Let $\mathbb{P}^t_{Q,T} = \frac{1}{N} \sum_{i=1}^N \mathbb{P}^i_{Q,T}$ be the non-hallucination domain as the uniform mixture of $N$ test domains $\mathbb{P}^1_{Q,T}, ..., \mathbb{P}^N_{Q,T}$ introduced in Section 2. The value of SpikeScore trained on the associated training domain $\mathbb{P}_{Q,T}$ satisfies that*

$$2\mathbb{E}_{(\mathbf{Q}, \mathbf{A}) \sim \mathbb{P}^t_{Q,T}} \mathrm{Max}|\Delta^2|(\mathbf{S}(\mathbf{Q}, \mathbf{A})) < \mathbb{E}_{(\mathbf{Q}, \mathbf{H}) \sim \mathbb{P}^t_{Q,H}} \mathrm{Max}|\Delta^2|(\mathbf{S}(\mathbf{Q}, \mathbf{H})), \tag{10}$$

*where $\mathbb{P}^t_{Q,H}$ is the hallucination domain corresponding to $\mathbb{P}^t_{Q,T}$, the marginal distributions related to the question random variable $Q$ of $\mathbb{P}^t_{Q,T}$ and $\mathbb{P}^t_{Q,H}$ are consistent.*

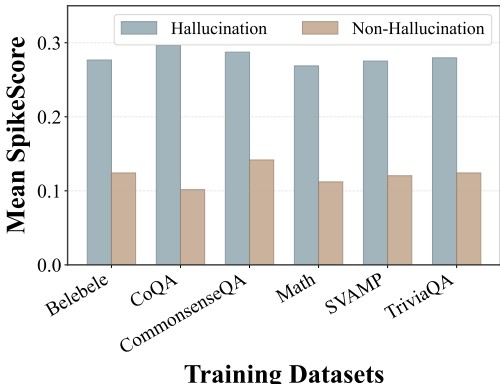 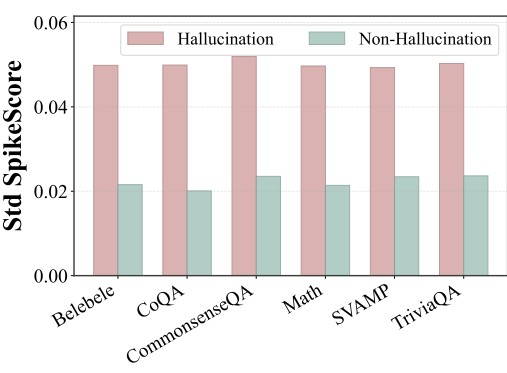

(a) Mean values across training datasets

(b) Standard deviations across training datasets

Figure 2: Statistical properties of SpikeScore for hallucination detection across cross-domain scenarios. Using Llama-3.1-8B-Instruct, we train uncertainty probes on individual datasets (e.g., CoQA) and evaluate on an equal mixture of the remaining datasets (TriviaQA, CommonsenseQA, Belebele, Math, SVAMP). Figure (a) demonstrates that hallucinated dialogues produce higher mean SpikeScore values compared to factual dialogues across all training configurations, indicating robust discriminative capability. Figure (b) shows that hallucinated dialogues exhibit noticeably higher standard deviations compared to factual ones. Although the variance gap is larger, Theorem 1 guarantees that separability between hallucinated and factual domains still holds under controlled coefficient-of-variation conditions.

Based on the comparison in Figure 2 (a), we find that the expectation of SpikeScore for the hallucination domain is more than twice that for the non-hallucination domain. This result reflects the separability of hallucination and non-hallucination domains captured by SpikeScore. Although this result aligns with our expectation, it is based solely on the expected SpikeScore. If SpikeScore exhibits large variance, the separability between the hallucination and non-hallucination domains remains questionable. Hence, we further study the standard deviation of SpikeScore in Figure 2 (b).

**Observation 2 (Standard Deviation Differences.)** *For the non-hallucination domain $\mathbb{P}_{Q,T}^t$ introduced in Observation 1, the value of SpikeScore trained on the associated training domain $\mathbb{P}_{Q,T}$ satisfies that*

$$1 < \frac{\mathbf{Std}_{(\mathbf{Q},\mathbf{H})\sim\mathbb{P}_{Q,H}^t}\mathrm{Max}|\Delta^2|(\mathbf{S}(\mathbf{Q},\mathbf{H}))}{\mathbf{Std}_{(\mathbf{Q},\mathbf{A})\sim\mathbb{P}_{Q,T}^t}\mathrm{Max}|\Delta^2|(\mathbf{S}(\mathbf{Q},\mathbf{A}))} \le 2.5,$$

*where $\mathbb{P}_{Q,H}^t$ is the hallucination domain introduced in Observation 1.*

In Figure 2 (b), we observe that the standard deviation of SpikeScore in the hallucination domain is larger than in the non-hallucination domain, with the ratio of standard deviations reaching 2.5, which exceeds the expectation ratio of 2 reported in Observation 1. This indicates that variance differences alone may hinder direct separability. Nevertheless, as shown in Theorem 1, separability between hallucination and non-hallucination domains can still be established under controlled coefficient-of-variation conditions (i.e., variance normalized by expectation). See Theorem 1 for details.

**Theorem 1** *Suppose Observations 1 and 2 hold. If the coefficient of variation for $\mathbb{P}_{Q,T}^t$ satisfies*

$$\mathbf{CV}_{(\mathbf{Q},\mathbf{A})\sim\mathbb{P}_{Q,T}^t}\mathrm{Max}|\Delta^2|(\mathbf{S}(\mathbf{Q},\mathbf{A})) = \frac{\mathbf{Std}_{(\mathbf{Q},\mathbf{A})\sim\mathbb{P}_{Q,T}^t}\mathrm{Max}|\Delta^2|(\mathbf{S}(\mathbf{Q},\mathbf{A}))}{\mathbb{E}_{(\mathbf{Q},\mathbf{A})\sim\mathbb{P}_{Q,T}^t}\mathrm{Max}|\Delta^2|(\mathbf{S}(\mathbf{Q},\mathbf{A}))} \le 0.1 \cdot t,$$

*for some $t > 0$, then we have*

$$\mathbb{P}\big(\mathrm{Max}|\Delta^2|(\mathbf{S}(\mathbf{Q}',\mathbf{H}')) > \mathrm{Max}|\Delta^2|(\mathbf{S}(\mathbf{Q},\mathbf{A}))\big) \ge \frac{1}{1 + 0.0725 \cdot t^2},$$

*where $(\mathbf{Q}',\mathbf{H}') \sim \mathbb{P}_{Q,H}^t$ is the hallucinated sample and $(\mathbf{Q},\mathbf{A}) \sim \mathbb{P}_{Q,T}^t$ is the factual sample.*

***Proof.*** *The proof of Theorem 1 can be found in Appendix A.*

Theorem 1 states that under Observations 1 and 2, if the coefficient of variation of SpikeScore in the hallucination domain is sufficiently small ($0.1 \cdot t$ in Theorem 1), then SpikeScore in the hallucination domain will, with high probability, exceed that in the non-hallucination domain. Therefore, we further study the coefficient of variation of SpikeScore through experiments.

In Figure 3, we report the coefficient of variation of SpikeScore across six groups of experiments, conducted under the same settings as in Figure 2. From this, we derive the following observation.

**Observation 3 (Controlled Coefficient of Variation.)** *For the non-hallucination domain* $\mathbb{P}^t_{Q,T}$, *the value of SpikeScore trained on the training domain* $\mathbb{P}_{Q,T}$ *satisfies that*

$$\mathbf{CV}_{(\mathbf{Q},\mathbf{A}) \sim \mathbb{P}^t_{Q,T}} \mathrm{Max}|\Delta^2|(\mathbf{S}(\mathbf{Q},\mathbf{A})) \leq 0.2. \tag{11}$$

Figure 3 shows that the coefficient of variation for non-hallucination domain does not exceed $0.2$. Then, Theorem 1 implies that *the SpikeScore in the hallucination domain will exceed that in the non-hallucination domain with probability at least* $0.775$, i.e.,

$$\mathbb{P}\big(\mathrm{Max}|\Delta^2|(\mathbf{S}(\mathbf{Q},\mathbf{H})) > \mathrm{Max}|\Delta^2|(\mathbf{S}(\mathbf{Q},\mathbf{A}))\big) \geq \frac{1}{1 + 0.0725 \cdot 2^2} \approx 0.775.$$

We defer a fuller discussion of separability to Appendix E, where we also compare SpikeScore with alternative indicators across models and observe generally stronger and more consistent performance for SpikeScore across a range of evaluation settings and representative cross-domain scenarios.

Through the above analyses, we have shown that SpikeScore is able to achieve good separability between hallucination and non-hallucination domains with high probability. It should be noted that, to ensure robust cross-domain evaluation, we follow a commonly adopted practice in cross-domain generalization studies (Hendrycks et al., 2019; Liu et al., 2020; Wang et al., 2022; Wang & Li, 2025). Specifically, after training on one dataset, we evaluate on a held-out pool formed by uniformly mixing the remaining datasets, a protocol commonly used to measure cross-domain generalization. From a probabilistic perspective, this pooled evaluation is equivalent to taking the expectation of separability over individual test domains, and thus faithfully reflects cross-domain generalization under diverse conditions.

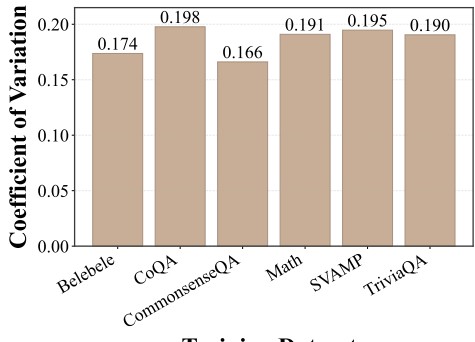

Figure 3: The coefficient of variation of SpikeScore based on non-hallucinated examples across datasets. Bars represent the coefficient of variation calculated from the maximum second-order difference within the first 20 steps. The experimental setup is identical to Figure 2. All coefficient of variation values remain below 0.2.

### 3.3 LEVERAGING SPIKESCORE FOR GHD

In this part, we leverage SpikeScore to construct our method for GHD. Given a question $\mathbf{Q}$ and the corresponding answer $\mathbf{A}$, we first compute the SpikeScore of $\mathbf{A}$ according to Eq. (9). Based on this score, we then apply the following formulation to determine whether $\mathbf{A}$ is hallucinated or non-hallucinated: given a threshold $\lambda > 0$,

$$D_\lambda(\mathbf{Q},\mathbf{A}) = \begin{cases} 0, & \text{if } \mathrm{Max}\left|\Delta^2\right|(\mathbf{S}(\mathbf{Q},\mathbf{A})) < \lambda, \\ 1, & \text{otherwise}, \end{cases} \tag{12}$$

where $1$ indicates that $\mathbf{A}$ is hallucinated, and $0$ indicates that $\mathbf{A}$ is non-hallucinated.

Table 1: Cross-domain hallucination detection performance. To ensure fair comparison, we use a leave-one-out protocol: training-based methods train on each dataset (columns) while all methods are evaluated on the remaining five datasets, preventing training-based methods from being tested on their own training domain (*see Appendix D.5 for details*). The table is split into two blocks only due to space constraints, corresponding to different model families. We report mean AUROC across test datasets. Best results are **bold**, second-best are underlined.

| Method | TriviaQA | | CommonsenseQA | | Belebele | | Training Dataset CoQA | | Math | | SVAMP | | **Average** | |
|---|---|---|---|---|---|---|---|---|---|---|---|---|---|---|
| | Mean AUROC↑ | | Mean AUROC↑ | | Mean AUROC↑ | | Mean AUROC↑ | | Mean AUROC↑ | | Mean AUROC↑ | | Mean AUROC↑ | |
| | Llama 3.2-3B | Llama 3.1-8B | Llama 3.2-3B | Llama 3.1-8B | Llama 3.2-3B | Llama 3.1-8B | Llama 3.2-3B | Llama 3.1-8B | Llama 3.2-3B | Llama 3.1-8B | Llama 3.2-3B | Llama 3.1-8B | Llama 3.2-3B | Llama 3.1-8B |
| **Training-free methods** | | | | | | | | | | | | | | |
| Perplexity | 0.5876 | 0.6353 | 0.5893 | 0.6379 | 0.5995 | 0.6445 | 0.5962 | 0.6420 | 0.5976 | 0.6448 | 0.6013 | 0.6505 | 0.5953 | 0.6425 |
| Semantic Entropy | 0.6123 | 0.6615 | 0.6250 | 0.6725 | 0.6320 | 0.6769 | 0.6133 | 0.6615 | 0.6307 | 0.6812 | 0.6149 | 0.6666 | 0.6214 | 0.6700 |
| EigenScore | 0.6431 | 0.6945 | 0.6378 | 0.6895 | 0.6515 | 0.7038 | 0.6417 | 0.6876 | 0.6515 | 0.7043 | 0.6503 | 0.6979 | 0.6460 | 0.6963 |
| Lexical Similarity | 0.6299 | 0.6961 | 0.6283 | 0.6911 | 0.6305 | 0.6882 | 0.6254 | 0.6946 | 0.6333 | 0.6946 | 0.6379 | 0.7014 | 0.6309 | 0.6917 |
| Verbalize | 0.5045 | 0.5520 | 0.4990 | 0.5476 | 0.5077 | 0.5560 | 0.4873 | 0.5400 | 0.4965 | 0.5436 | 0.4931 | 0.5420 | 0.4980 | 0.5469 |
| InterrogateLLM | 0.6508 | 0.6992 | 0.6528 | 0.7030 | 0.6449 | 0.6993 | 0.6456 | 0.7013 | 0.6531 | 0.7041 | 0.6544 | 0.7050 | 0.6503 | 0.7020 |
| **Training-based methods** | | | | | | | | | | | | | | |
| MM | 0.5787 | 0.5784 | 0.5468 | 0.5384 | 0.4800 | 0.4893 | 0.5948 | 0.6120 | 0.5767 | 0.5841 | 0.5778 | 0.5788 | 0.5591 | 0.5635 |
| SEP | 0.5597 | 0.5417 | 0.5170 | 0.5102 | 0.5358 | 0.5004 | 0.4984 | 0.5221 | 0.5390 | 0.5420 | 0.5119 | 0.5529 | 0.5270 | 0.5282 |
| SAPLMA | 0.5723 | 0.6141 | 0.5319 | 0.5217 | 0.4933 | 0.5074 | 0.6468 | 0.6358 | 0.5861 | 0.6003 | 0.5855 | 0.5790 | 0.5693 | 0.5764 |
| **Cross-domain specialized methods** | | | | | | | | | | | | | | |
| PRISM | 0.6817 | 0.6971 | 0.6988 | 0.7051 | 0.7067 | 0.6974 | 0.7984 | 0.7917 | 0.6051 | 0.6475 | 0.6811 | 0.6784 | 0.6953 | 0.7029 |
| ICR Probe | **0.7464** | 0.7310 | **0.7118** | 0.7205 | **0.7416** | 0.7498 | 0.8155 | 0.8074 | 0.7615 | 0.7487 | 0.7010 | 0.7058 | 0.7463 | 0.7439 |
| **SpikeScore** | 0.7316 | **0.7654** | 0.6947 | **0.7451** | 0.7088 | **0.7719** | **0.8540** | **0.8584** | **0.7699** | **0.8004** | **0.7252** | **0.7751** | **0.7474** | **0.7860** |
| | Qwen3 -8B | Qwen3 -14B | Qwen3 -8B | Qwen3 -14B | Qwen3 -8B | Qwen3 -14B | Qwen3 -8B | Qwen3 -14B | Qwen3 -8B | Qwen3 -14B | Qwen3 -8B | Qwen3 -14B | Qwen3 -8B | Qwen3 -14B |
| **Training-free methods** | | | | | | | | | | | | | | |
| Perplexity | 0.5963 | 0.6222 | 0.6108 | 0.6287 | 0.6153 | 0.6311 | 0.6090 | 0.6284 | 0.6126 | 0.6343 | 0.6224 | 0.6362 | 0.6111 | 0.6302 |
| Semantic Entropy | 0.6295 | 0.6499 | 0.6385 | 0.6664 | 0.6368 | 0.6684 | 0.6302 | 0.6532 | 0.6481 | 0.6785 | 0.6310 | 0.6569 | 0.6357 | 0.6622 |
| EigenScore | 0.6651 | 0.6942 | 0.6560 | 0.6864 | 0.6727 | 0.7055 | 0.6538 | 0.6809 | 0.6753 | 0.6995 | 0.6636 | 0.6978 | 0.6644 | 0.6940 |
| Lexical Similarity | 0.6491 | 0.6683 | 0.6402 | 0.6626 | 0.6423 | 0.6632 | 0.6397 | 0.6571 | 0.6422 | 0.6625 | 0.6493 | 0.6682 | 0.6438 | 0.6636 |
| Verbalize | 0.5273 | 0.5402 | 0.5232 | 0.5226 | 0.5228 | 0.5345 | 0.5172 | 0.5137 | 0.5090 | 0.5308 | 0.5123 | 0.5237 | 0.5186 | 0.5276 |
| InterrogateLLM | 0.6662 | 0.6845 | 0.6670 | 0.6814 | 0.6611 | 0.6777 | 0.6761 | 0.6787 | 0.6771 | 0.6852 | 0.6718 | 0.6904 | 0.6699 | 0.6830 |
| **Training-based methods** | | | | | | | | | | | | | | |
| MM | 0.5858 | 0.5755 | 0.5309 | 0.5411 | 0.4866 | 0.4854 | 0.6161 | 0.6037 | 0.5789 | 0.5958 | 0.5808 | 0.5795 | 0.5632 | 0.5635 |
| SEP | 0.5395 | 0.5273 | 0.5157 | 0.5164 | 0.4923 | 0.4952 | 0.5164 | 0.5275 | 0.5309 | 0.5550 | 0.5326 | 0.5430 | 0.5212 | 0.5274 |
| SAPLMA | 0.6344 | 0.6108 | 0.5163 | 0.5281 | 0.4985 | 0.5119 | 0.6301 | 0.6392 | 0.5685 | 0.6063 | 0.5753 | 0.5760 | 0.5705 | 0.5787 |
| **Cross-domain specialized methods** | | | | | | | | | | | | | | |
| PRISM | 0.7017 | 0.6923 | 0.7075 | 0.7187 | 0.6887 | 0.7097 | 0.7891 | 0.8116 | 0.6516 | 0.6427 | 0.6809 | 0.6681 | 0.7032 | 0.7072 |
| ICR Probe | 0.7317 | 0.7337 | 0.7249 | 0.7186 | **0.7271** | 0.7488 | 0.8000 | 0.8084 | 0.7281 | 0.7434 | 0.7170 | 0.7078 | 0.7381 | 0.7435 |
| **SpikeScore** | **0.7410** | **0.7689** | **0.7411** | **0.7563** | 0.6960 | **0.7664** | **0.8205** | **0.8333** | **0.7504** | **0.8157** | **0.7344** | **0.7837** | **0.7473** | **0.7874** |

# 4 EXPERIMENT

## 4.1 EXPERIMENTAL SETUPS

We evaluate SpikeScore on six widely used hallucination detection benchmarks: CommonsenseQA (Talmor et al., 2019), TriviaQA (Joshi et al., 2017), Belebele (Costa-Jussà et al., 2025), CoQA (Reddy et al., 2019), Math (Hendrycks et al., 2021), and SVAMP (Patel et al., 2021). Baselines include Perplexity (Ren et al., 2023), Semantic Entropy (Farquhar et al., 2024), EigenScore (Chen et al., 2024a), Lexical Similarity (Lin et al., 2024), Verbalize (Lin et al., 2022), InterrogateLLM (Yehuda et al., 2024), MM (Marks & Tegmark, 2023), SEP (Kossen et al., 2024; Farquhar et al., 2024), SAPLMA (Azaria & Mitchell, 2023), PRISM (SAPLMA instantiation) (Zhang et al., 2025b), and ICR Probe (Zhang et al., 2025c), with AUROC as the primary evaluation metric. We test across four carefully selected open-source LLMs—Llama-3.2-3B-Instruct and Llama-3.1-8B-Instruct (Dubey et al., 2024), and Qwen3-8B-Instruct and Qwen3-14B-Instruct (Yang et al., 2025a)—to cover diverse model families and parameter scales. Additional dataset sampling strategies, baseline configurations, and hyperparameters are provided in detail in Appendix C.

## 4.2 EXPERIMENTAL RESULTS

**Overall Cross-domain Hallucination Detection Performance.** In Table 1, we compare the proposed SpikeScore with three categories of baselines: training-free methods that avoid generalization issues, traditional training-based methods, and cross-domain specialized approaches. SpikeScore consistently achieves the highest average AUROC across all model families and maintains superior or competitive performance in individual dataset evaluations. Traditional training-based methods (MM, SEP, SAPLMA) generally exhibit poor cross-domain performance, while training-free methods show reasonable adaptability but fall short of specialized cross-domain approaches like PRISM

Table 2: Performance of SpikeScore combined with different backbone scoring methods. Following the leave-one-out protocol from Table 1, we evaluate SpikeScore integrated with both training-free methods (Perplexity, Reasoning score, In-Context Sharpness) and training-based methods (SEP, SAPLMA) across six datasets using different LLMs. The table is split into two blocks only due to space constraints, corresponding to different model families. We report mean AUROC across test datasets. Best results are **bold**, second-best are underlined.

| Method | Leave-one-out Dataset | | | | | | | | | | | | | |
| | TriviaQA | | CommonsenseQA | | Belebele | | CoQA | | Math | | SVAMP | | **Average** | |
| | Mean AUROC↑ | | Mean AUROC↑ | | Mean AUROC↑ | | Mean AUROC↑ | | Mean AUROC↑ | | Mean AUROC↑ | | Mean AUROC↑ | |
| | Llama 3.2-3B | Llama 3.1-8B | Llama 3.2-3B | Llama 3.1-8B | Llama 3.2-3B | Llama 3.1-8B | Llama 3.2-3B | Llama 3.1-8B | Llama 3.2-3B | Llama 3.1-8B | Llama 3.2-3B | Llama 3.1-8B | Llama 3.2-3B | Llama 3.1-8B |
| **Training-free scoring methods** | | | | | | | | | | | | | | |
| SpikeScore+Perplexity | 0.6541 | 0.6942 | 0.6254 | 0.6755 | 0.6505 | 0.7033 | 0.8094 | 0.7936 | 0.6978 | 0.7225 | 0.6357 | 0.7066 | 0.6788 | 0.7160 |
| SpikeScore+Reasoning score | 0.7208 | 0.7282 | 0.6559 | 0.7211 | 0.6924 | 0.7504 | 0.8222 | 0.8267 | 0.7471 | 0.7794 | 0.6795 | 0.7514 | 0.7196 | 0.7595 |
| SpikeScore+In-Context Sharpness | 0.6772 | 0.7334 | 0.6615 | 0.6867 | 0.6505 | 0.7098 | 0.8185 | 0.7816 | 0.7170 | 0.7498 | 0.6836 | 0.7250 | 0.7014 | 0.7311 |
| **Training-based scoring methods** | | | | | | | | | | | | | | |
| SpikeScore+SEP | 0.7230 | 0.7389 | 0.6824 | 0.7341 | 0.6873 | 0.7518 | 0.8315 | 0.8405 | 0.7470 | 0.7888 | 0.7051 | 0.7565 | 0.7294 | 0.7684 |
| **SpikeScore+SAPLMA** | **0.7316** | **0.7654** | **0.6947** | **0.7451** | **0.7088** | **0.7719** | **0.8540** | **0.8584** | **0.7699** | **0.8004** | **0.7252** | **0.7751** | **0.7474** | **0.7860** |
| | Qwen3-8B | Qwen3-14B | Qwen3-8B | Qwen3-14B | Qwen3-8B | Qwen3-14B | Qwen3-8B | Qwen3-14B | Qwen3-8B | Qwen3-14B | Qwen3-8B | Qwen3-14B | Qwen3-8B | Qwen3-14B |
| **Training-free methods** | | | | | | | | | | | | | | |
| SpikeScore+Perplexity | 0.6717 | 0.7051 | 0.6693 | 0.6692 | 0.6112 | 0.6926 | 0.7458 | 0.7382 | 0.6870 | 0.7533 | 0.6598 | 0.7096 | 0.6742 | 0.7113 |
| SpikeScore+Reasoning score | 0.7055 | 0.7118 | 0.7018 | 0.7179 | 0.6650 | 0.7012 | 0.7955 | 0.7952 | 0.7305 | 0.7750 | 0.6995 | 0.7314 | 0.7163 | 0.7387 |
| SpikeScore+In-Context Sharpness | 0.6692 | 0.7128 | 0.7095 | 0.7219 | 0.6574 | 0.7174 | 0.7923 | 0.7779 | 0.6869 | 0.7566 | 0.6899 | 0.7434 | 0.7008 | 0.7383 |
| **Training-based methods** | | | | | | | | | | | | | | |
| SpikeScore+SEP | 0.7190 | 0.7548 | 0.7172 | 0.7451 | 0.7021 | 0.7484 | 0.7972 | 0.8244 | 0.7346 | 0.7918 | 0.7060 | 0.7661 | 0.7293 | 0.7718 |
| **SpikeScore+SAPLMA** | **0.7410** | **0.7689** | **0.7411** | **0.7563** | 0.6960 | **0.7664** | **0.8205** | **0.8333** | **0.7504** | **0.8157** | **0.7344** | **0.7837** | **0.7473** | **0.7874** |

and ICR Probe. Notably, SpikeScore demonstrates improved performance with larger models within the same family, suggesting that enhanced self-correction capabilities in larger LLMs (Yang et al., 2025d) generate more detectable attention spike patterns during conversational refinement.

**Compatibility with Different Backbone Scoring Methods.** To validate the generalizability of our spike-based detection framework, we comprehensively evaluate SpikeScore with various backbone scoring methods, including both training-free approaches and training-based methods. As shown in Table 2, all combinations demonstrate effective hallucination detection performance across different LLMs and datasets, strongly suggesting that the spike phenomenon we identified is a general and transferable characteristic of hallucinated outputs rather than specific to particular scoring mechanisms. Moreover, training-based methods (SpikeScore+SEP and SpikeScore+SAPLMA) achieve substantially better performance than their training-free counterparts, further confirming the practical advantage of learning-based approaches in cross-domain hallucination detection. Implementation details for adapting these scoring methods to our framework are provided in Appendix C.3.

**Extended Analyses and Ablations.** We conduct extension studies detailed in Appendix D. First, we analyze cross-domain generalization (Appendix D.1): SpikeScore captures low-frequency, domain-agnostic instability patterns through multi-turn dialogue dynamics, enabling rapid convergence with minimal training data, whereas single-step methods rely on high-frequency, semantically-entangled features that fail to transfer. Second, train-test heatmaps (Appendix D.2) confirm strong in-domain performance with relatively clear diagonal patterns, though cross-domain evaluation remains our primary focus. Third, ablations against coefficient of variation (Appendix D.3) demonstrate that second-order differences consistently outperform first-order variability measures, validating our curvature-based approach. Finally, dialogue length analysis (Appendix D.4) shows that performance saturates around 15–20 turns, which motivates the default setting of $K=20$. Based on this finding, we regard the first 20 dialogue turns as the most informative region for hallucination detection and refer to them as the *early stage* throughout this paper, rather than extending dialogues indefinitely.

## 4.3 EXTENDING SPIKESCORE TO RETRIEVAL-AUGMENTED SETTINGS

SpikeScore is a post-hoc framework that operates entirely on the multi-turn continuation trajectories after an initial answer is produced. This property makes it naturally compatible with structured pipelines such as retrieval-augmented generation (RAG): regardless of whether the upstream system performs retrieval, tool calling, or other pipeline operations, once a natural language answer is available, we can induce self-dialogue, extract turn-level hallucination scores, and compute SpikeScore over the resulting sequence.

Table 3: RAG hallucination detection performance across two datasets (TriviaQA, RAGTruth) and four LLMs. Results are reported as AUROC↑. Best results are **bold**, second-best are underlined.

| | Evaluation Dataset | | | | | | | |
| Method | Llama3.2-3B AUROC↑ | | Llama3.1-8B AUROC↑ | | Qwen3-8B AUROC↑ | | Qwen3-14B AUROC↑ | |
| | TriviaQA | RAGTruth | TriviaQA | RAGTruth | TriviaQA | RAGTruth | TriviaQA | RAGTruth |
|---|---|---|---|---|---|---|---|---|
| **Training-free methods** | | | | | | | | |
| Perplexity | 0.5960 | 0.5936 | 0.6220 | 0.6151 | 0.6371 | 0.6284 | 0.6474 | 0.6420 |
| Semantic Entropy | 0.6330 | 0.6210 | 0.6173 | 0.6037 | 0.6492 | 0.6388 | 0.6745 | 0.6796 |
| EigenScore | 0.6426 | 0.6329 | 0.7085 | 0.6974 | 0.6571 | 0.6292 | 0.7064 | 0.6904 |
| Lexical Similarity | 0.6248 | 0.6179 | 0.7102 | 0.6958 | 0.6375 | 0.6440 | 0.6542 | 0.6479 |
| **Training-based methods** | | | | | | | | |
| MM | 0.5724 | 0.5587 | 0.6020 | 0.5839 | 0.6166 | 0.6044 | 0.6291 | 0.6191 |
| SEP | 0.5312 | 0.4988 | 0.5779 | 0.5547 | 0.5161 | 0.5253 | 0.5368 | 0.5239 |
| SAPLMA | 0.6041 | 0.5874 | 0.5970 | 0.6011 | 0.6136 | 0.5904 | 0.6399 | 0.6338 |
| **Cross-domain specialized methods** | | | | | | | | |
| ICR Probe | 0.7532 | 0.7416 | 0.7682 | 0.7405 | 0.7495 | 0.7426 | 0.7841 | 0.7653 |
| **SpikeScore** | **0.7737** | **0.7631** | **0.7947** | **0.8154** | **0.8413** | **0.8291** | **0.8697** | **0.8535** |

To examine this in realistic RAG scenarios, we build a retrieval pipeline using TriviaQA (Joshi et al., 2017) and RAGTruth (Niu et al., 2024), two widely used benchmarks in RAG research. Rather than relying on the datasets' predefined question–context pairs, we treat all evidence passages as a shared corpus, embed queries and documents with bge-large-en (Xiao et al., 2023), and retrieve the top-4 candidates via a Faiss (Johnson et al., 2019) index. The retrieved passages are concatenated with the question and fed to the same LLMs used in the main dialogue experiments. Importantly, all training-based methods—including SpikeScore—are trained only on non-RAG CoQA, so evaluation on TriviaQA and RAGTruth becomes a strict cross-domain generalization test. Detailed pipeline construction is provided in Appendix D.6.

Table 3 reports the AUROC on both RAG datasets. Training-free baselines degrade substantially under retrieval-induced noise, and training-based methods also lose accuracy when transferred from dialogue-style data to retrieval-based pipelines. In contrast, SpikeScore consistently outperforms all baselines, including the strongest competitor (ICR Probe), across every model scale and on both datasets. This robustness indicates that SpikeScore's curvature-based temporal features remain discriminative even when hallucinations stem from imperfect retrieval or incomplete evidence.

We further observe that TriviaQA and RAGTruth exhibit highly aligned trends: SpikeScore remains the leading method across different LLM families and parameter sizes, and its performance margins over baselines are stable across datasets. Together with CoQA—which corresponds to an "idealized RAG" scenario where retrieval is perfect—these results show that SpikeScore generalizes reliably from open-ended dialogue to structured, tool-augmented workflows. This suggests that the temporal curvature features captured by SpikeScore are not tied to any specific interaction pattern but instead reflect a deeper regularity in how models behave when drifting toward hallucination. As retrieval-augmented systems become increasingly central in agentic architectures, such robustness is especially valuable, indicating that SpikeScore may serve as a unifying post-hoc indicator across both free-form and pipeline-driven LLM deployments.

## 5 CONCLUSION

In this paper, we introduce a new perspective for hallucination detection, termed *generalizable hallucination detection* (GHD), which focuses on building detectors trained on a single domain yet capable of transferring across diverse domains. To this end, we propose *SpikeScore*, an indicator designed to quantify sharp fluctuations in multi-turn dialogue paths. We provide both theoretical analysis and extensive experiments to show that SpikeScore achieves strong cross-domain separability and consistently outperforms representative baselines. We further validate its generality by integrating it with different backbone scoring methods and conducting ablation studies. We hope our findings encourage future work on developing cross-domain hallucination detection methods that exploit dialogue dynamics and instability signals to enhance robustness in diverse settings.

ACKNOWLEDGEMENT

This work is partially supported by the ARC Future Fellowship FT230100121 and the ARC Discovery Early Career Researcher Award DE250100363; and by the AFOSR Young Investigator Program FA9550-23-1-0184, NSF awards IIS-2237037 and IIS-2331669, and ONR grant N00014-23-1-2643, as well as support from the Schmidt Sciences Foundation, Open Philanthropy, an Alfred P. Sloan Fellowship, and unrestricted gifts from Google and Amazon. We also thank **Sze To Leung** for assistance with code engineering and implementation organization. *The views and conclusions contained in this document are those of the authors and should not be interpreted as representing the official policies, either expressed or implied, of the funding agencies or other supporters.*

ETHICS STATEMENT

Our study adheres to the ICLR Code of Ethics. All experiments were conducted on publicly available datasets and open-source language models, as listed in Appendix C. No private, sensitive, or personally identifiable information is involved. The primary objective of this work is to advance the understanding of hallucination detection in large language models, with an emphasis on transparency, fairness, and responsible research practices.

REPRODUCIBILITY STATEMENT

We provide a repository at https://github.com/TianYaDY/SpikeScore, which contains our source code, experiment configurations, and evaluation scripts. All models and benchmarks used in this study are publicly accessible, as detailed in Appendix C. Our experiments were conducted on NVIDIA A100 GPUs, using Python 3.10 and PyTorch 2.4.0 to ensure reproducibility.

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

## A  THEORETICAL PROOFS

**Proof of Theorem 1.** Let $X = \text{Max}|\Delta^2|(\mathbf{S}(\mathbf{Q}, \mathbf{H}))$ and $Y = \text{Max}|\Delta^2|(\mathbf{S}(\mathbf{Q}, \mathbf{A}))$. Using Cantelli's inequality (one-sided Chebyshev), we can derive a distribution-free lower bound. Let

$$D = X - Y.$$

By independence, we have

$$\text{Var}(D) = \sigma_X^2 + \sigma_Y^2, \text{ where } \sigma_X = \text{Std}X \text{ and } \sigma_Y = \text{Std}Y,$$

and

$$\mathbb{P}(X > Y) = \mathbb{P}(D > 0) \geq \frac{(\mathbb{E}D)^2}{\text{Var}(D) + (\mathbb{E}D)^2}.$$

Define

$$\delta = \frac{\mathbb{E}X}{\mathbb{E}Y} > 2, \quad r = \frac{\sigma_X}{\sigma_Y} \in (1, 2.5], \quad c = \frac{\sigma_Y}{\mathbb{E}Y} \leq 0.1t.$$

Then

$$\mathbb{E}D = (\delta - 1)\mathbb{E}Y, \qquad \text{Var}(D) = (r^2 + 1)c^2(\mathbb{E}Y)^2.$$

Hence,

$$\mathbb{P}(X > Y) \geq \frac{(\delta - 1)^2}{(r^2 + 1)c^2 + (\delta - 1)^2}.$$

Under the worst-case values $\delta \downarrow 2$, $r \uparrow 2.5$, $c \uparrow 0.1t$, we obtain

$$\mathbb{P}(X > Y) \geq \frac{1}{1 + (2.5^2 + 1)(0.1t)^2} = \frac{1}{1 + 0.0725t^2}.$$

## B  RELATED WORK

**Robustness, Distributional Generalization, and Reliable AI Systems.** A growing body of research studies the broader problem of reliability and robustness in modern AI systems beyond task-specific hallucination detection. Theoretical work investigates when out-of-distribution (OOD) detection is learnable and characterizes conditions under which reliable separation between in- and out-of-distribution samples can be achieved (Fang et al., 2022). Complementary approaches aim to improve OOD robustness by mitigating spurious correlations and exploiting non-semantic signals that remain stable across distributions (Fang et al., 2025). Related efforts introduce evaluation benchmarks designed to stress-test reasoning consistency, conversational robustness, and adversarial resilience in multimodal or speech-based interaction settings (Yang et al., 2025c;b). Parallel work on agentic systems studies uncertainty-aware decision-making, selective intervention, and adaptive knowledge retrieval mechanisms to maintain reliable behavior under incomplete information or evolving environments (Fu et al., 2026; Li et al., 2026). Recent multimodal modeling research further explores hierarchical feature alignment and structured reasoning strategies to enhance robustness and interpretability in specialized domains (Qiu et al., 2025). Collectively, these studies highlight the importance of signals that remain stable across domains, modalities, and interaction conditions, which motivates the search for domain-invariant indicators of answer reliability.

**Hallucination Detection.** A large body of research studies how to detect hallucinations in LLM outputs using signals that range from output-space uncertainty to internal activations and multi-sample agreement (Ji et al., 2023; Minaee et al., 2024). Uncertainty-based approaches estimate risk from token probabilities, perplexity, or entropy-like measures, sometimes in a semantic space, and often operate in zero-shot or black-box settings (Farquhar et al., 2024). Consistency-based methods generate multiple answers to the same question and infer reliability from agreement or structured contradictions among the samples (Manakul et al., 2023). A complementary line trains lightweight probes on hidden states to predict truthfulness or semantic uncertainty from a single forward pass (Azaria & Mitchell, 2023; Kossen et al., 2024; Chen et al., 2024a; Beigi et al., 2024). Recent work further explores improving cross-domain robustness of supervised detectors by prompt-guiding internal states or explicitly tracking hidden-state dynamics across layers (Zhang et al., 2025b;c). While effective in many in-domain evaluations, these methods predominantly score a single turn (or aggregate independent re-generations of that turn) and do not explicitly model how an answer's reliability evolves when the conversation continues.

**Multi-turn Dialogue Challenges and Self-Contradiction.** Another line examines LLM behavior in multi-turn settings, documenting that models can rapidly defer to user suggestions or contradict earlier claims when repeatedly probed or challenged (Shinn et al., 2023). Empirical analyses show that such self-contradictory reasoning arises even without adversarial attacks and can be surfaced by targeted prompts or perturbations (Liu et al., 2024b). Related work studies contradictions against external or intra-document evidence and designs procedures to expose and mitigate them (Li et al., 2024). Mitigation frameworks such as self-reflection, iterative refinement, and chain-of-verification reduce hallucinations by eliciting model self-correction over multiple turns (Dhuliawala et al., 2024; Shinn et al., 2023; Madaan et al., 2023; Qiu et al., 2024). A recent effort explicitly characterizes "self-contradictory hallucinations" as a detectable failure mode (Mündler et al., 2023). However, these studies (Deng et al., 2025; 2024) generally analyze multi-turn behavior to improve generation quality or to surface contradictions in the abstract; they do not formulate a *post-answer* detection setting where the model's own answer is fixed and the ensuing, answer-conditioned dialogue trajectory is treated as the signal for hallucination detection, nor do they target domain-invariant separability as an explicit objective.

Building on these threads, our work shifts the detection target from one-shot scoring or multi-sample agreement on the original question to the *instability of the induced, answer-conditioned continuation*. Concretely, we simulate short follow-up dialogues that preserve the original context and quantify trajectory fluctuations via *SpikeScore*. This design turns oscillatory self-contradictions into a measurable signal and, as we show, yields a provable domain-invariant separability guarantee alongside strong cross-domain performance in practice.

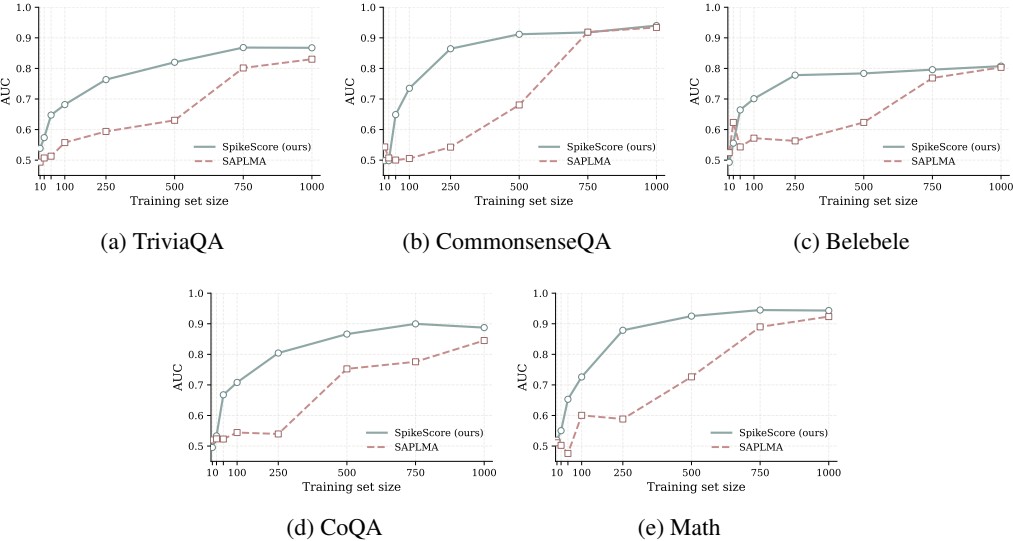

(a) TriviaQA      (b) CommonsenseQA      (c) Belebele

(d) CoQA      (e) Math

Figure 4: Effect of training set size on hallucination detection performance (AUROC). Each curve compares our method (solid line) with SAPLMA (dashed line) across increasing numbers of training samples. Our method quickly reaches saturation with very few training examples, whereas SAPLMA requires far more data to converge. This supports the view that our method leverages simple low-frequency features, while SAPLMA relies on complex high-frequency cues.

## C    DETAILS OF EXPERIMENTS

### C.1   DATASET DETAILS

**CommonsenseQA (Commonsense Reasoning).** CommonsenseQA (Talmor et al., 2019) is a 5-way multiple-choice benchmark requiring everyday commonsense knowledge beyond a given stem. It contains 12102 questions with one correct answer and four distractors, and provides both a *random* split (standard) and a *question-token* split. We use the random split as the source pool and subsample for our experiments. Unless otherwise specified, we draw 2000 training and 1000 test instances with a fixed seed, stratified over gold labels to maintain choice balance. Each instance is formatted as a

single-turn prompt (stem + options) for the initial answer, after which we induce follow-ups per our dialogue protocol.

**TriviaQA (Knowledge-intensive QA).**   TriviaQA (Joshi et al., 2017) pairs naturally-authored trivia questions with distant-supervision evidence from Wikipedia/Web. The full collection includes 95K question–answer pairs and 650K+ QA–evidence triples (on average six evidence documents per question). We treat TriviaQA as short-answer QA using the canonical answer/alias lists; evidence text is not shown to the model in our base setting to avoid confounding with retrieval. We sample 2000 training and 1000 test questions with a fixed seed and normalize answers via the official alias matching when computing reference agreement for labeling.

**Belebele (Multilingual Reading Comprehension).**   Belebele (Costa-Jussà et al., 2025) is a multiple-choice reading-comprehension benchmark spanning 122 language variants, each question paired with a short passage (derived from FLORES) and four options. To avoid script as a confound in cross-domain comparisons while retaining multilingual diversity, we use the full Latin-script (`Latn`) subset rather than English only. Concretely, we pool all `Latn` variants (e.g., `eng-Latn`, `fra-Latn`, `spa-Latn`, `deu-Latn`, etc.) and draw 2000 training and 1000 test items with a fixed seed, using language-stratified sampling to keep the mix approximately balanced across languages. Each item is formatted as *passage + question + options* for the initial turn, after which we induce the multi-turn continuation following our dialogue protocol.

**CoQA (Conversational Reading Comprehension).**   CoQA (Reddy et al., 2019) contains over 127K questions organized into 8K+ multi-turn conversations grounded in passages from seven domains. We convert each conversation into independent (passage, question) pairs using the author-provided evidence spans where available. To prevent context leakage, we enforce *passage-level* disjointness between our train and test partitions: passages appearing in training are excluded from testing. Unless noted, we sample 2000 training and 1000 test pairs with a fixed seed. The initial model answer is produced for a single (passage, question) pair; follow-up prompts are then conditioned on that answer to induce a dialogue.

**Math (Competition Mathematics).**   Math (Hendrycks et al., 2021) comprises 12500 problems from math competitions (AMC, AIME, etc.), each with a final answer and step-by-step solution. We evaluate short-form answer correctness against the provided final answer; solutions are not shown to the model. To ensure topic diversity, we stratify sampling over the official subject categories (e.g., algebra, number theory, geometry) when drawing 2000 training and 1000 test items with a fixed seed. Problem statements are used verbatim as the initial turn before dialogue induction.

**SVAMP (Elementary Math Word Problems).**   SVAMP (Patel et al., 2021) is a challenge set of simple arithmetic word problems constructed to reduce dataset artifacts present in earlier MWPs. It contains approximately one thousand items with single-number answers. Because of its smaller size, we use the entire set with a stratified 70/30 train/test split (fixed seed) under the same non-overlap constraint. As with Math, only the problem text is provided to the model for the initial answer.

**General Preprocessing and Splits.**   Unless stated above, we standardize inputs into a single-turn prompt to elicit the initial answer, then apply the same follow-up schedule (weak-to-strong prompts) to induce a multi-turn continuation. For multi-choice datasets (CSQA, Belebele), option texts are preserved as plain tokens; for short-answer datasets (TriviaQA, Math, SVAMP), references are normalized using dataset-specific rules (e.g., alias matching in TriviaQA; numeric normalization in Math/SVAMP). All subsampling uses a fixed seed to ensure replicability, and train/test partitions are disjoint at the appropriate granularity (passage-level for CoQA, instance-level otherwise).

## C.2   MODELS AND INFERENCE CONFIGURATION

We evaluate four instruction-tuned open-source LLMs spanning two families and scales: *Llama-3.2-3B-Instruct*, *Llama-3.1-8B-Instruct* (Dubey et al., 2024), *Qwen3-8B-Instruct*, and *Qwen3-14B-Instruct* (Yang et al., 2025a). We preserve each family's native interaction style so that the induced follow-ups reflect realistic usage conditions. The Llama series emphasizes instruction-following

efficiency; Qwen3 provides a "reasoning-oriented" mode that biases generation toward deeper intermediate deliberation. This diversity yields a broader testbed for stress-testing detector robustness across modeling paradigms and parameter scales.

**Llama-3.2-3B-Instruct.** A compact baseline for capacity-sensitive comparisons. We use a conservative decoding setup (Services, 2024) recommended for rigorous tasks: temperature $0.2$, top_p $0.9$.

**Llama-3.1-8B-Instruct.** A mid-scale model from the same family, used to examine whether larger capacity yields clearer instability signatures in answer-conditioned continuations. We adopt the same configuration as the 3B model (temperature $0.2$, top_p $0.9$).

**Qwen3-8B-Instruct.** We enable the model's reasoning-oriented mode to encourage richer intermediate deliberation during follow-ups (while keeping prompts and schedules identical across models). Following official guidance for reasoning-style generation (Alibaba, 2024), we use temperature $0.6$, top_p $0.95$.

**Qwen3-14B-Instruct.** A larger counterpart to assess scaling effects within the Qwen3 family under the same reasoning-oriented configuration (temperature $0.6$, top_p $0.95$).

**Rationale and Consistency.** Our setting keeps family-native behaviors (standard instruction-following vs. reasoning-oriented decoding) to reflect realistic deployment where detectors cannot control whether users elicit deeper deliberation. All models share the same conversation induction schedule (weak-to-strong prompts), number of turns, and evaluation pipeline; only the backbone model and its recommended decoding hyperparameters differ. Unless otherwise stated, other decoding options remain at provider defaults.

**Labeling Protocol.** For ground-truth supervision we use *GPT-4o* as an NLI-style judge: given a question, gold reference, and model answer, it assigns a binary label indicating whether the answer contradicts or departs from reference-supported facts. Labels are used solely for evaluation and for training lightweight backbones (e.g., SEP/SAPLMA) where applicable; the dialogue induction and SpikeScore computation remain unchanged across models and settings.

## C.3 DETAILS OF SCORING BACKBONES FOR SPIKESCORE

**Perplexity (PPL).** Perplexity (Ji et al., 2023) is a classical uncertainty surrogate based on token log-likelihoods. Given an answer at a dialogue turn, we compute the per-token negative log-probabilities under the model and use the *length-normalized* mean over the answer tokens as that turn's sentence-level PPL score (higher indicates greater uncertainty). In our pipeline, this produces a single scalar per turn without additional sampling or re-decoding; the time series of these scalars across turns is then fed to *SpikeScore*.

**Reasoning Score (RS).** Reasoning Score (Sun et al., 2025a) targets reasoning depth by leveraging late-layer geometry and its projection to the vocabulary space; the score is designed to separate shallow pattern matching from genuine multi-step reasoning. In the original formulation, RS is computed over the reasoning trace to quantify depth-sensitive signals. To reduce compute in our multi-turn setup, we adopt a lightweight approximation: for each turn, we extract the hidden states corresponding to the *last* and *penultimate* answer tokens from late layers, compute RS for these two positions (following the original projection/divergence recipe), and average them to obtain the turn-level RS. This preserves turnwise comparability while keeping the cost similar to sentence-level PPL.

**In-Context Sharpness (ICS).** In-Context Sharpness (Chen et al., 2024b) measures how sharply the model's token-level activations respond to the given context, yielding a tokenwise "sharpness" signal that has been used to diagnose hallucinations. In our adaptation, for each turn we do not alter decoding; after the answer is produced, we compute ICS *post hoc*: (i) extract hidden representations from the last five transformer layers (to stabilize across domains and sequence lengths), (ii) map

them through the model's unembedding to obtain vocabulary-level activations, (iii) compute per-token sharpness under the original ICS formulation, and (iv) average across tokens and across the last five layers to obtain a single turn-level ICS score.

**Semantic Entropy Probes (SEP).** SEP (Kossen et al., 2024) trains a lightweight probe on hidden activations to produce a *semantic-entropy surrogate*—a single-pass approximation to multi-sample uncertainty. Concretely, we follow SEP's design but make two pragmatic choices for our multi-turn setting: (i) features are the hidden states from the last five layers at the *penultimate* answer token (a robust position for stability), concatenated or pooled before the probe; (ii) instead of binarizing targets, we regress the semantic-entropy target using a Huber loss to reduce sensitivity to outliers. After training on labeled data in the training domain, inference yields one probe score per turn; the sequence feeds into *SpikeScore*.

**SAPLMA (Internal-State MLP).** SAPLMA (Azaria & Mitchell, 2023) attaches a small MLP with sigmoid to internal activations to predict the truthfulness of a generated answer. Following findings that late layers and the final tokens are most informative, we use hidden states from the *last five layers* at the *last* answer token as features and train the MLP with a standard cross-entropy objective on hallucination labels. At inference, each turn produces a calibrated probability (or logit) that serves as the turn-level score; the resulting sequence is then summarized by *SpikeScore*.

**Implementation Notes.** All backbones are used *post hoc*: we do not modify decoding. For turn-level aggregation, we consistently use sentence-level means over tokens when a method yields tokenwise scores (PPL, ICS), and a pooled late-layer representation at the last or penultimate token when a method is representation-based (SEP, SAPLMA, RS). To improve stability under mixed sequence lengths and domains, late-layer features are averaged over the last five layers unless a method prescribes a single layer. These choices yield a unified per-turn scalar time series for each dialogue, from which *SpikeScore* (maximum second-order difference) is computed.

## D ADDITIONAL EXPERIMENTAL RESULTS AND FURTHER DISCUSSION

### D.1 ADDITIONAL DISCUSSION ON TRAINING AND CROSS-DOMAIN ROBUSTNESS

Although Section 4 has shown that combining our method with classical training-based approaches such as SAPLMA yields the strongest in-domain performance, it is natural to ask why training improves performance at all, given that our approach is designed to capture domain-invariant features. We argue that training plays a complementary role: it calibrates the probe to highlight the most informative structures in the signal. In practice, training helps the probe filter out incidental noise and focus on the distinctive instability patterns that correlate with hallucination trajectories. This alignment effect is especially visible in our method, where the trajectory-based curvature signal already reflects a universal property of hallucinations. With limited training, the probe can quickly converge toward emphasizing this low-dimensional, domain-agnostic structure.

The second question concerns cross-domain transfer. *Why do training-based methods, despite their apparent advantage in-domain, degrade almost completely when applied out-of-domain, while our approach remains stable?* We attribute this phenomenon to the principle of *spectral bias* in neural networks. When trained with a small number of examples, neural probes tend to learn low-frequency, broadly shared features before attempting to fit more complex, high-frequency patterns. In hallucination detection, curvature-based instability signals belong to the low-frequency category: they reflect coarse, domain-independent fluctuations in reasoning trajectories rather than fine-grained semantic content. In contrast, single-step probes such as SAPLMA can only manifest discriminative power by relying on higher-frequency, semantically entangled cues, which are necessarily domain-specific. As a result, training encourages them to memorize more complex correlations that fail to generalize across domains.

To make this distinction concrete, we conducted a controlled experiment varying the number of training samples available to the probe. On each dataset (except SVAMP due to its limited size), we trained both our method and SAPLMA on subsets of 10, 25, 50, 100, 250, 500, 750, and 1000 samples, and then evaluated on 2000 held-out test examples from the same dataset using the LLaMA-3.1-8B model. The results are shown in Figure 4. The key observation is that our method rapidly

saturates: with only a small fraction of training data, its AUROC is already close to the maximum achievable. By contrast, SAPLMA requires far more training data to approach comparable performance, indicating that it depends on richer, more domain-specific cues.

This pattern is consistent with the spectral bias perspective. Because our method exploits low-frequency, domain-agnostic features, these signals are easy to learn and quickly saturate. SAPLMA, however, relies on high-frequency, semantically complex information that is slower to acquire and more sensitive to domain shifts. Importantly, both methods share the same training procedure. This means SAPLMA does learn some of the simple low-frequency structure as well, but since it only operates on single-step scores, these features are not directly expressed in its decision function. In our method, by contrast, the curvature-based dynamics explicitly expose these low-frequency instabilities, making them effective even with minimal training.

In summary, training enhances both methods, but in fundamentally different ways. Our approach rapidly reaches saturation by amplifying simple, domain-general features that are inherently transferable, while SAPLMA relies on slower, domain-specific learning. This explains why our method not only trains efficiently but also retains robustness under cross-domain evaluation.

## D.2 CROSS-DATASET HEATMAPS

To better visualize train–test behavior, we report 6×6 train–test heatmaps for each model and scoring variant. Each heatmap places training domains on the rows and test domains on the columns; diagonal cells correspond to in-domain evaluation and off-diagonal cells measure cross-domain generalization. We plot AUROC values with a common color scale for comparability across models. The goal here is not to optimize in-domain scores, but to reveal how well a detector trained on a single source domain transfers to others.

Several consistent patterns emerge. First, our method attains strong diagonal performance, which is expected because the scoring rule is learned or calibrated on the training domain. However, the design of our approach targets cross-domain separability rather than chasing diagonal gains, so the emphasis is on off-diagonal behavior. Across models, we observe robust transfer on many off-diagonal cells, indicating that trajectory instability captures signals that are less tied to domain-specific semantics. We also notice that training on CoQA often leads to relatively strong transfer. A plausible explanation is that CoQA's multi-turn conversational style provides diverse follow-up contexts and encourages self-assessment, which aligns with our continuation-induced probing and makes instability patterns easier to separate. While this advantage is not universal across all pairs, it recurs frequently enough to be noteworthy.

## D.3 SPIKESCORE COMPARED WITH THE COEFFICIENT OF VARIATION

This section empirically compares the proposed *SpikeScore* (maximum second-order difference along the score sequence) with the *coefficient of variation (CV)* baseline. Both metrics are computed from the same per-turn scoring sequence produced by a training-based backbone (either SAPLMA or SEP), so any difference stems from the fluctuation measure rather than from the underlying scorer.

We use **Llama-3.1-8B-Instruct** and treat **CoQA** as the training domain. For each backbone (SAPLMA or SEP) we train on 1000 samples and test on 2000 samples under two regimes: *in-domain* (train on CoQA, test on held-out CoQA) and *out-of-domain* (train on CoQA, test on a mixed pool drawn from the remaining benchmarks). Unless otherwise noted, we use the same chain length $K$ as the main experiments (default $K=20$), compute a single scalar per instance (CV or SpikeScore), visualize class-conditional distributions via KDE, and report AUROC for quantitative comparison. Figure 7 and Figure 8 show the KDEs.

Across all four settings (SAPLMA/SEP × in/out-domain), **SpikeScore yields visibly larger separation** between hallucinated and factual distributions than CV, which is **consistent with the AUROC improvements** we observe in the same conditions. In-domain, CV already provides some discrimination, but SpikeScore displays a wider margin. Out-of-domain, the advantage of SpikeScore becomes more pronounced: CV's two classes overlap more substantially, whereas SpikeScore retains a clearer gap.

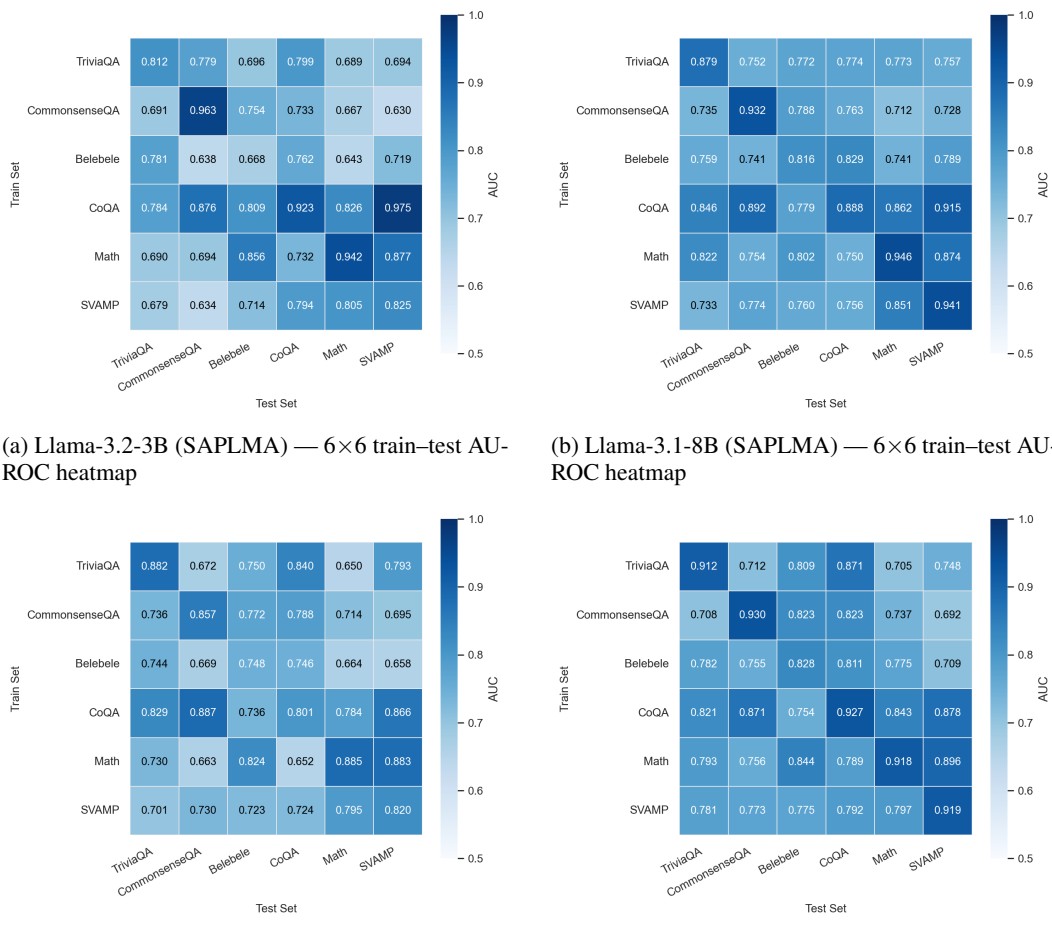

(a) Llama-3.2-3B (SAPLMA) — 6×6 train–test AU-ROC heatmap

(b) Llama-3.1-8B (SAPLMA) — 6×6 train–test AU-ROC heatmap

(c) Qwen3-8B (SAPLMA) — 6×6 train–test AU-ROC heatmap

(d) Qwen3-14B (SAPLMA) — 6×6 train–test AU-ROC heatmap

Figure 5: Cross-dataset heatmaps using SAPLMA-based scoring. Rows denote training domains and columns denote test domains.

CV summarizes *global* variability by normalizing the standard deviation with the mean, which makes it sensitive to slow drifts, scale effects, and length-dependent dilution. In multi-turn dialogues, however, we frequently observe *short, localized bursts* of instability: a model that started off-track tends to produce a brief, high-curvature correction/justification phase early in the continuation (e.g., reversing a stance, patching a gap), followed by re-stabilization. Such *transient spikes* can be *masked* in CV because the global denominator shrinks as uncertainty decays, and the global numerator blends low-frequency trends with high-frequency irregularities. By contrast, the *maximum second-order difference* behaves like a discrete curvature operator (a simple high-pass filter on the trajectory): it *suppresses slow drifts* that both classes share and *highlights sharp, localized changes* that are more prevalent in hallucination-initiated continuations. This filtering effect explains why SpikeScore preserves separability even when overall uncertainty compresses under continued dialogue.

### D.4 IMPACT OF DIALOGUE STEP COUNT ON SPIKESCORE PERFORMANCE

This section studies how many dialogue turns are needed to obtain a reliable instability signal. Dialogue length cannot be increased indefinitely. First, long continuations consume considerable compute, which limits practical deployment. Second, as turns accumulate, even factually grounded conversations may occasionally drift and trigger a late hallucination. Once that happens, the remainder of the continuation often inherits spike-like patterns, which in turn hurts detection specificity.

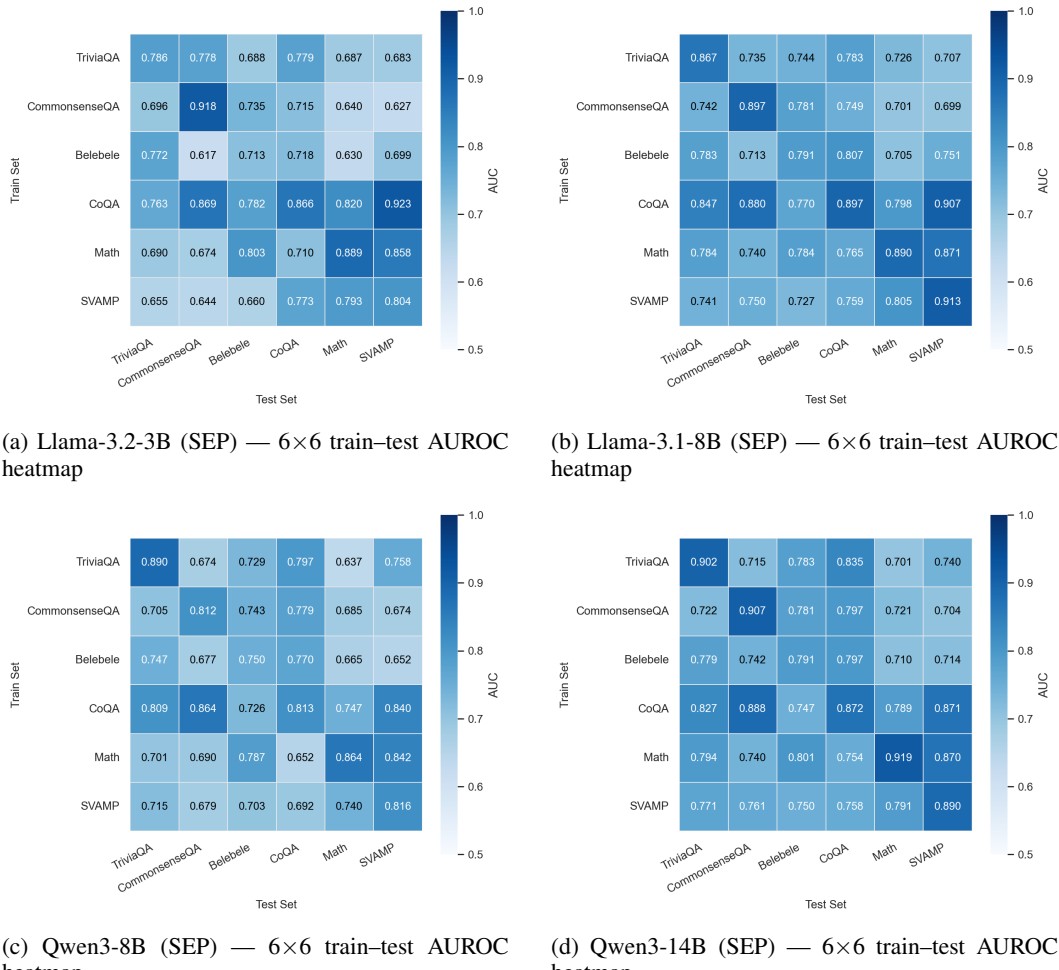

(a) Llama-3.2-3B (SEP) — 6×6 train–test AUROC heatmap

(b) Llama-3.1-8B (SEP) — 6×6 train–test AUROC heatmap

(c) Qwen3-8B (SEP) — 6×6 train–test AUROC heatmap

(d) Qwen3-14B (SEP) — 6×6 train–test AUROC heatmap

Figure 6: Cross-dataset heatmaps using SEP-based scoring. Rows denote training domains and columns denote test domains.

We therefore seek a balance where hallucination-initiated chains have already exposed their characteristic spikes, while factual chains have not yet been pushed into instability.

We evaluate the effect of step count using four models and six datasets, yielding twenty-four settings in total. Apart from the step budget, all configurations follow the same protocol as in Section 4. For each setting we generate a continuation capped at 50 turns, which we regard as sufficiently long to reveal the trend. We then truncate the continuation after the first $K$ turns (for varying $K$) and compute SpikeScore from only those $K$ turns. Using the resulting score as the detection signal, we compute the AUROC at each $K$, producing an AUROC-versus-steps curve. This isolates the marginal utility of additional turns under an otherwise fixed setup, including the same prompt schedule and scoring method.

The results are consistent across most model–dataset pairs. AUROC typically rises quickly and reaches a plateau around fifteen to twenty turns. In several cases the curve saturates even earlier, near fifteen turns. Toward the tail of the budget, we observe mild but noticeable declines in AUROC. This suggests that excessively long continuations begin to induce instability even in factual chains, creating spurious spikes that erode separability. Combining these observations, a step budget of about 20 turns offers a good trade-off between effectiveness and resource cost: it is long enough for hallucination-initiated trajectories to reveal their instability patterns, yet short enough to avoid late-turn degradation and unnecessary compute.

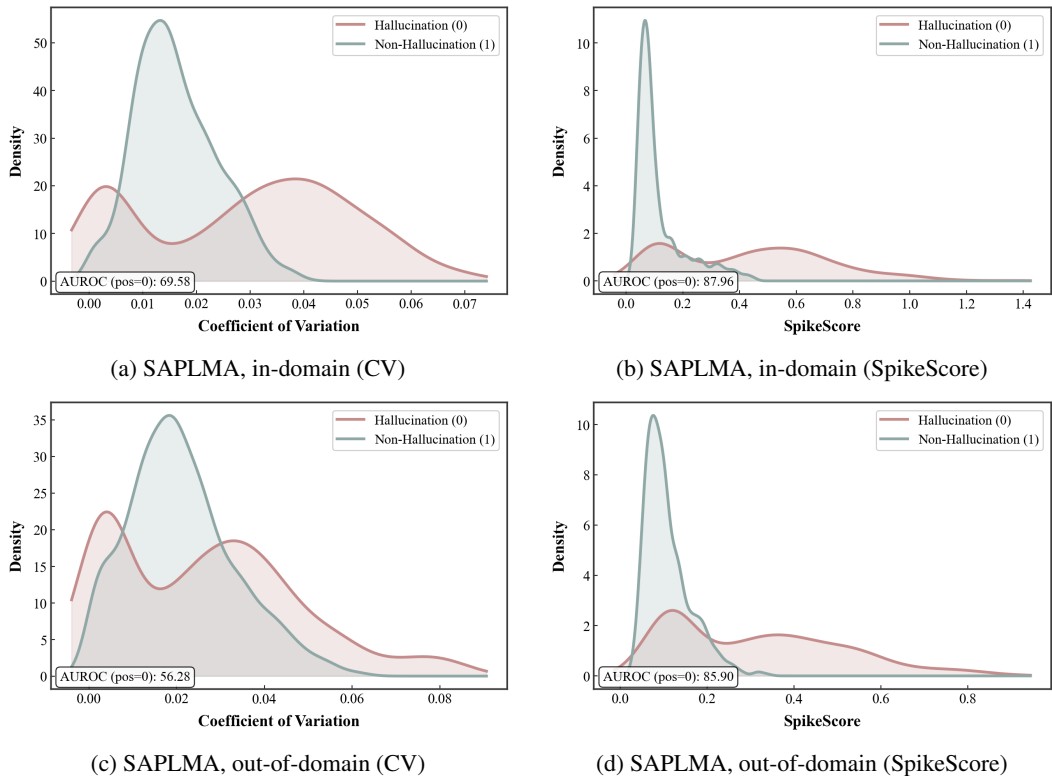

Figure 7: KDE comparison for SAPLMA backbone. SpikeScore consistently shows clearer class separation than the coefficient of variation, both in-domain and out-of-domain.

## D.5 CLARIFYING THE LEAVE-ONE-OUT EVALUATION PROTOCOL

The main tables in this paper report cross-domain hallucination detection performance under a leave-one-out evaluation protocol. This design ensures a fair comparison between training-based and training-free methods. For training-based approaches (e.g., SEP, SAPLMA), models must be trained on one source dataset and evaluated on the remaining five target datasets so that they are never tested on their own training domain. Training-free methods, by contrast, do not require task-specific training and can be directly applied to any dataset. However, without applying the same leave-one-out protocol, the two classes of methods would be evaluated under inherently different data conditions, making the comparison in the main tables no longer balanced. To avoid such asymmetry, training-free methods are also evaluated only on held-out datasets, effectively "adapting" to the overall experimental structure to ensure methodological fairness across all methods.

While this protocol is necessary for fair cross-domain comparison, its numerical consequences make the main tables less intuitive—particularly for training-free methods. Because their reported scores reflect performance on *out-of-domain* datasets rather than on the dataset shown in each column, the table entries no longer correspond to a model's direct performance on the dataset being displayed. For this reason, the per-dataset performance of training-free methods may appear disconnected from the column dataset name, even though the evaluation protocol is consistent and fair.

To provide a clearer view of the *in-domain* behavior of training-free baselines, we report in Table 4 their direct AUROC performance on each dataset without the leave-one-out restriction. These values reflect the most immediate and interpretable performance of each baseline on its test domain. Together with the cross-domain results reported in the main paper, this supplementary table offers a complete picture of how training-free methods behave both in-domain and out-of-domain.

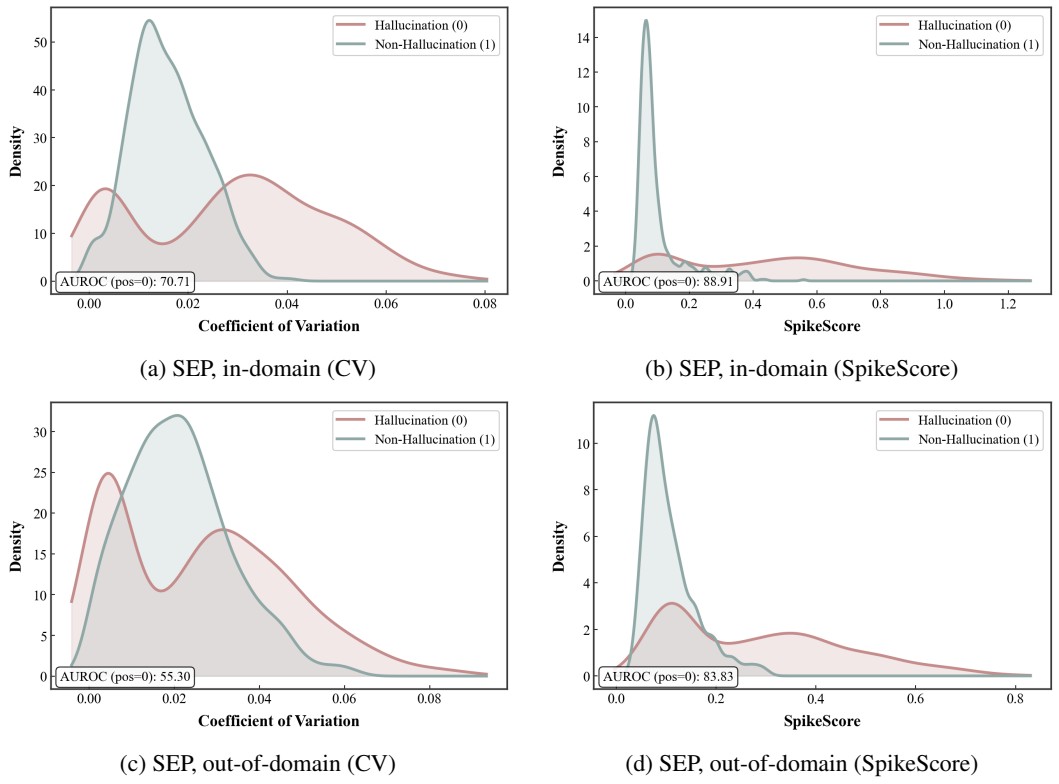

Figure 8: KDE comparison for SEP backbone. As with SAPLMA, SpikeScore outperforms the coefficient of variation in both regimes.

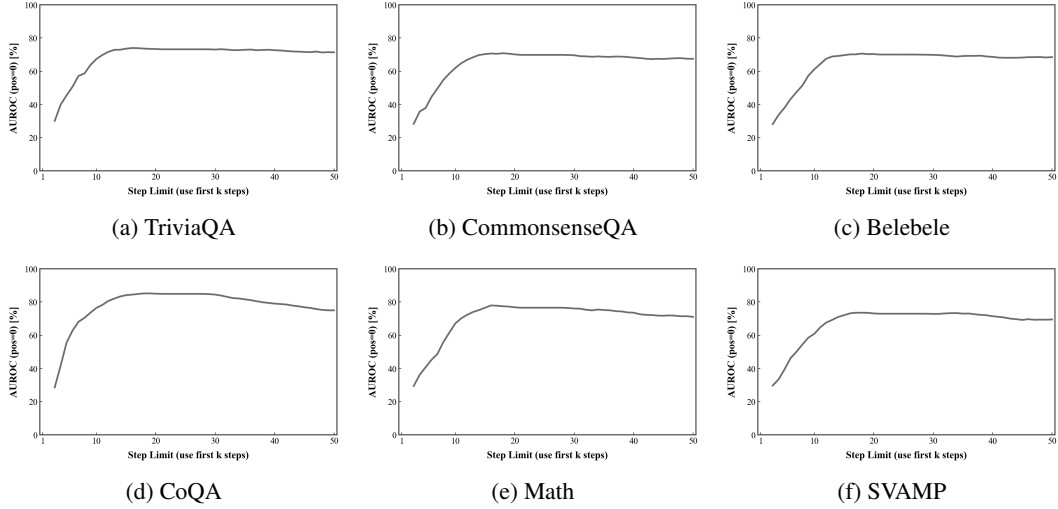

Figure 9: AUROC vs. dialogue steps for Llama-3.2-3B across six datasets. Each curve is computed by truncating to the first $K$ turns and recomputing SpikeScore.

## D.6 EXTENDING SPIKESCORE TO RETRIEVAL-AUGMENTED SETTINGS

Our method is essentially a post-hoc analysis framework: SpikeScore depends only on the multi-turn continuation trajectories after the model produces an initial answer and imposes no constraints on the internal process by which this answer is generated. Specifically, regardless of whether the underlying system employs retrieval, function calling, or other external tools when producing the

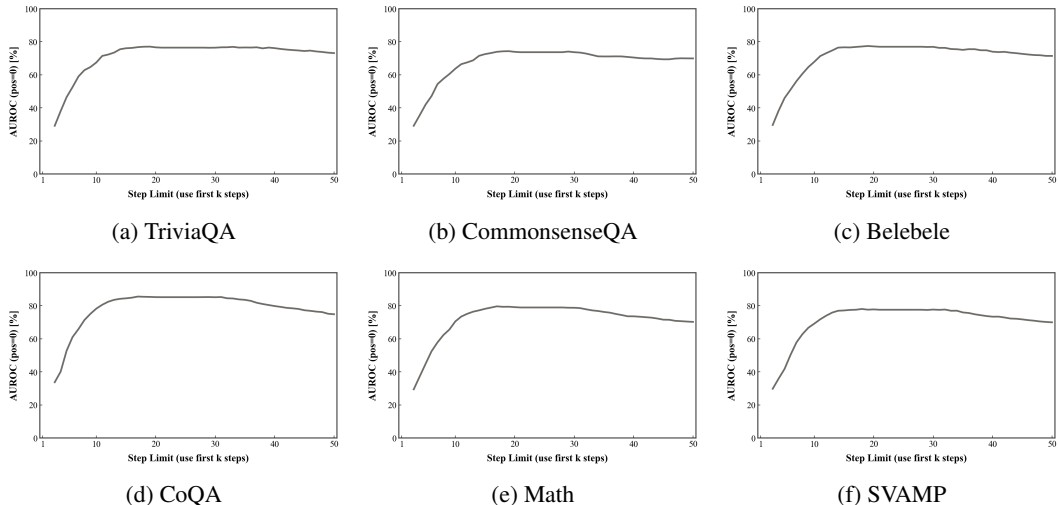

(a) TriviaQA  (b) CommonsenseQA  (c) Belebele

(d) CoQA  (e) Math  (f) SVAMP

Figure 10: AUROC vs. dialogue steps for Llama-3.1-8B across six datasets. Saturation generally appears by ∼15–20 turns, with slight late-turn declines.

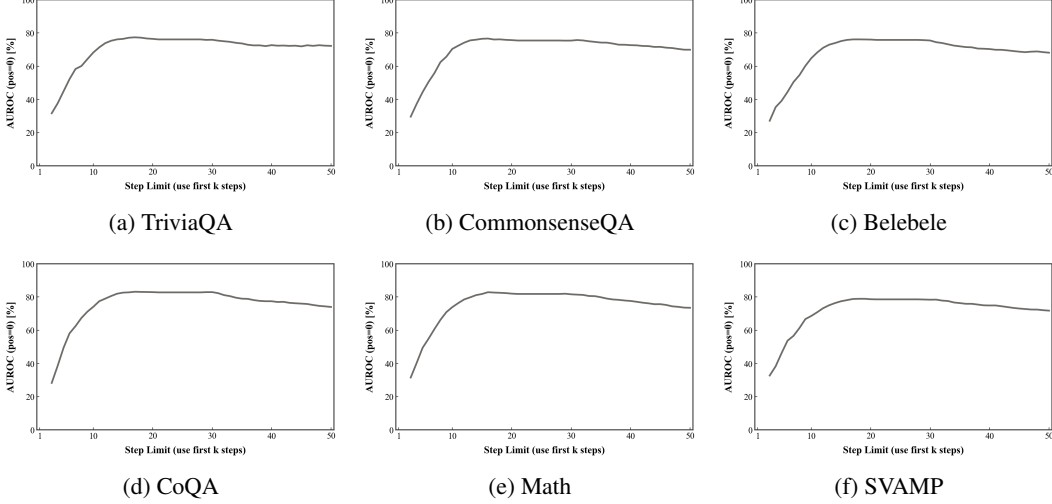

(a) TriviaQA  (b) CommonsenseQA  (c) Belebele

(d) CoQA  (e) Math  (f) SVAMP

Figure 11: AUROC vs. dialogue steps for Qwen3-8B across six datasets. Early growth followed by a plateau near 15–20 turns is the dominant pattern.

initial answer, as long as it ultimately yields a natural language output, we can induce additional multi-turn self-dialogue on top of it, extract the sequence of hallucination scores at each turn, and compute SpikeScore over the resulting time series. In this sense, SpikeScore is not conceptually tied to any particular interaction pattern, but is naturally applicable to more general pipeline-based and tool-augmented workflows.

In the main experiments, we first evaluate SpikeScore systematically in open-ended, free-form dialogue settings. In such configurations, the model operates without explicit structural constraints and exhibits highly diverse behavior patterns, which are typically regarded as among the most challenging and least stable conditions for hallucination detection. Beyond these open-ended scenarios, many practical applications instead organize the reasoning process of LLMs through structured pipelines, among which retrieval-augmented generation (RAG) is a representative example. RAG usually follows a fixed stage sequence of "parsing the query – invoking a retrieval tool – integrating external evidence – generating the final answer": the LLM acts as a controller that interprets the user query and triggers retrieval operations, the retrieval component selects candidate documents from a vector database, and these documents are then returned together with the original query to the LLM to pro-

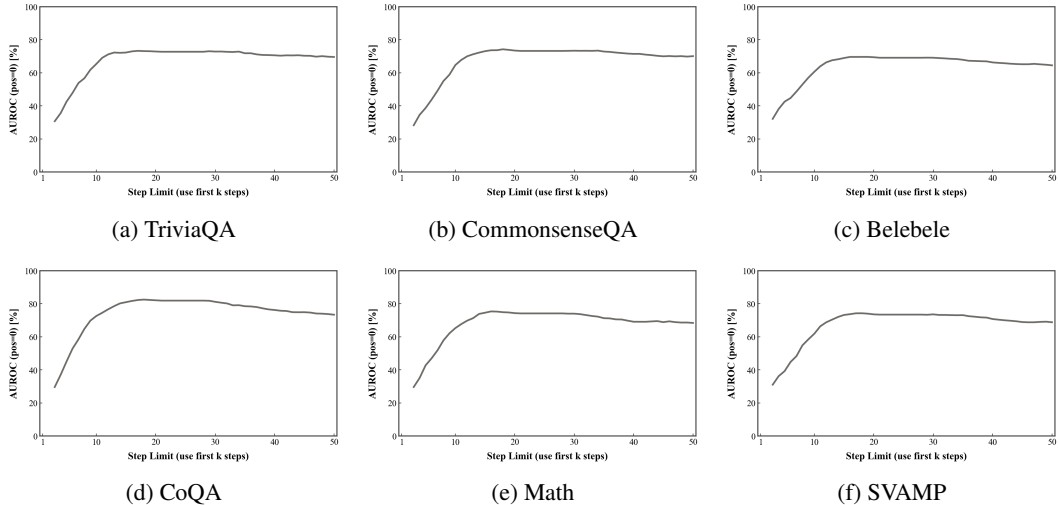

|   |   |   |
|---|---|---|
| (a) TriviaQA | (b) CommonsenseQA | (c) Belebele |
| (d) CoQA | (e) Math | (f) SVAMP |

Figure 12: AUROC vs. dialogue steps for Qwen3-14B across six datasets. Longer budgets bring little extra benefit beyond the early plateau and can slightly reduce AUROC.

Table 4: Direct (non–leave-one-out) hallucination detection performance of training-free baselines. To complement the cross-domain results in Table 1, we report AUROC on each dataset when methods are evaluated directly on their test domain. Best results are **bold**, second-best are underlined.

| | TriviaQA | | CommonsenseQA | | Belebele | | Evaluation Dataset CoQA | | Math | | SVAMP | | **Average** | |
|---|---|---|---|---|---|---|---|---|---|---|---|---|---|---|
| Method | Mean AUROC↑ | | Mean AUROC↑ | | Mean AUROC↑ | | Mean AUROC↑ | | Mean AUROC↑ | | Mean AUROC↑ | | Mean AUROC↑ | |
| | Llama 3.2-3B | Llama 3.1-8B | Llama 3.2-3B | Llama 3.1-8B | Llama 3.2-3B | Llama 3.1-8B | Llama 3.2-3B | Llama 3.1-8B | Llama 3.2-3B | Llama 3.1-8B | Llama 3.2-3B | Llama 3.1-8B | Llama 3.2-3B | Llama 3.1-8B |
| **Training-free baselines** | | | | | | | | | | | | | | |
| Perplexity | 0.6336 | 0.6784 | 0.6248 | 0.6654 | 0.5739 | 0.6325 | 0.5908 | 0.6451 | 0.5833 | 0.6313 | 0.5652 | 0.6024 | 0.5953 | 0.6425 |
| Semantic Entropy | **0.6669** | 0.7125 | 0.6030 | 0.6578 | 0.5683 | 0.6358 | 0.6615 | 0.7125 | 0.5748 | 0.6142 | **0.6537** | 0.6874 | 0.6214 | 0.6700 |
| EigenScore | 0.6604 | 0.7051 | **0.6871** | **0.7302** | 0.6185 | 0.6587 | 0.6673 | 0.7397 | 0.6181 | 0.6560 | 0.6244 | **0.6879** | 0.6460 | 0.6963 |
| Lexical Similarity | 0.6358 | 0.6698 | 0.6439 | 0.6947 | 0.6327 | 0.7095 | 0.6581 | **0.7554** | 0.6188 | 0.6774 | 0.5958 | 0.6435 | 0.6309 | 0.6917 |
| Verbalize | 0.4655 | 0.5210 | 0.4930 | 0.5432 | 0.4496 | 0.5013 | 0.5519 | 0.5812 | 0.5057 | 0.5633 | 0.5225 | 0.5712 | 0.4980 | 0.5469 |
| InterrogateLLM | 0.6476 | **0.7158** | 0.6378 | 0.6970 | 0.6768 | 0.7154 | 0.6733 | 0.7054 | 0.6362 | 0.6915 | 0.6297 | 0.6869 | **0.6502** | 0.7020 |
| | Qwen3 -8B | Qwen3 -14B | Qwen3 -8B | Qwen3 -14B | Qwen3 -8B | Qwen3 -14B | Qwen3 -8B | Qwen3 -14B | Qwen3 -8B | Qwen3 -14B | Qwen3 -8B | Qwen3 -14B | Qwen3 -8B | Qwen3 -14B |
| **Training-free baselines** | | | | | | | | | | | | | | |
| Perplexity | 0.6847 | 0.6699 | 0.6125 | 0.6373 | 0.5898 | 0.6256 | 0.6214 | 0.6387 | 0.6034 | 0.6095 | 0.5545 | 0.5999 | 0.6110 | 0.6301 |
| Semantic Entropy | 0.6666 | **0.7239** | 0.6217 | 0.6414 | 0.6300 | 0.6314 | 0.6632 | 0.7071 | 0.5735 | 0.5807 | 0.6591 | **0.6887** | 0.6357 | 0.6622 |
| EigenScore | 0.6611 | 0.6932 | **0.7067** | **0.7324** | 0.6229 | 0.6367 | **0.7174** | 0.7600 | 0.6099 | 0.6668 | 0.6686 | 0.6752 | 0.6644 | **0.6940** |
| Lexical Similarity | 0.6173 | 0.6406 | 0.6619 | 0.6690 | 0.6512 | 0.6657 | 0.6641 | 0.6964 | 0.6518 | 0.6691 | 0.6164 | 0.6410 | 0.6438 | 0.6636 |
| Verbalize | 0.4754 | 0.4645 | 0.4957 | 0.5523 | 0.4977 | 0.4931 | 0.5259 | 0.5968 | 0.5669 | 0.5116 | 0.5502 | 0.5471 | 0.5186 | 0.5276 |
| InterrogateLLM | **0.6883** | 0.6755 | 0.6843 | 0.6909 | **0.7139** | **0.7094** | 0.6386 | 0.7044 | 0.6338 | **0.6720** | 0.6604 | 0.6458 | **0.6699** | 0.6830 |

duce the final answer. This pattern, in which the LLM orchestrates tool calls, both reflects a typical structured pipeline-based workflow and can be viewed as one of the most basic and common forms of tool use in agentic workflows. Although RAG mitigates knowledge-related hallucinations to some extent by explicitly introducing document evidence, prior work and public benchmarks indicate that it still exhibits characteristic hallucination modes, such as incorrect attribution, inconsistency with evidence, and fabrications driven by noisy retrieval. It is therefore practically relevant to examine how hallucination detection metrics behave in RAG settings.

It is worth noting that the CoQA dataset used in the main experiments already exhibits, to some extent, the structural pattern of "retrieval plus generation". CoQA provides each question with a complete context passage that is sufficient to answer the question. From an idealized perspective, this configuration can be viewed as a special case of RAG with a "perfect retriever", i.e., equivalent to always selecting the single correct evidence passage exactly at the retrieval stage. However, such ideal retrieval corresponds only to the best-case scenario in a RAG pipeline and does not cover more general real-world settings, such as approximate retrieval, contamination by noisy documents, or partial retrieval failures.

Table 5: RAG hallucination detection performance across two datasets (TriviaQA, RAGTruth) and four LLMs. Results are reported as AUROC↑. Best results are **bold**, second-best are underlined.

| | Evaluation Dataset | | | | | | | |
| Method | Llama3.2-3B AUROC↑ | | Llama3.1-8B AUROC↑ | | Qwen3-8B AUROC↑ | | Qwen3-14B AUROC↑ | |
| | TriviaQA | RAGTruth | TriviaQA | RAGTruth | TriviaQA | RAGTruth | TriviaQA | RAGTruth |
|---|---|---|---|---|---|---|---|---|
| **Training-free methods** | | | | | | | | |
| Perplexity | 0.5960 | 0.5936 | 0.6220 | 0.6151 | 0.6371 | 0.6284 | 0.6474 | 0.6420 |
| Semantic Entropy | 0.6330 | 0.6210 | 0.6173 | 0.6037 | 0.6492 | 0.6388 | 0.6745 | 0.6796 |
| EigenScore | 0.6426 | 0.6329 | 0.7085 | 0.6974 | 0.6571 | 0.6292 | 0.7064 | 0.6904 |
| Lexical Similarity | 0.6248 | 0.6179 | 0.7102 | 0.6958 | 0.6375 | 0.6440 | 0.6542 | 0.6479 |
| **Training-based methods** | | | | | | | | |
| MM | 0.5724 | 0.5587 | 0.6020 | 0.5839 | 0.6166 | 0.6044 | 0.6291 | 0.6191 |
| SEP | 0.5312 | 0.4988 | 0.5779 | 0.5547 | 0.5161 | 0.5253 | 0.5368 | 0.5239 |
| SAPLMA | 0.6041 | 0.5874 | 0.5970 | 0.6011 | 0.6136 | 0.5904 | 0.6399 | 0.6338 |
| **Cross-domain specialized methods** | | | | | | | | |
| ICR Probe | 0.7532 | 0.7416 | 0.7682 | 0.7405 | 0.7495 | 0.7426 | 0.7841 | 0.7653 |
| **SpikeScore** | **0.7737** | **0.7631** | **0.7947** | **0.8154** | **0.8413** | **0.8291** | **0.8697** | **0.8535** |

To evaluate the behavior of SpikeScore under more realistic RAG configurations, we explicitly construct a retrieval-augmented pipeline based on vector search in our extended experiments. Specifically, we select the TriviaQA dataset from the main experiments and the RAGTruth benchmark (Niu et al., 2024) that is widely used in RAG research. Similar to CoQA, the original form of both datasets specifies one or several reference evidence passages for each question. To mimic retrieval behavior that more closely resembles real systems, rather than directly relying on the pre-specified "question–context" pairing in the datasets, we first treat all reference evidence passages uniformly as independent text chunks in a shared knowledge base and regard all questions as independent queries, thereby completely breaking the original one-to-one correspondences when constructing the retrieval corpus. We then use `bge-large-en`(Xiao et al., 2023) as a sentence/paragraph-level embedding model to encode each text chunk and each query into a shared semantic vector space, and build an approximate nearest-neighbor index in Faiss (Johnson et al., 2019) over these embedding vectors. For each question, we retrieve the top-4 most similar context passages from this index and concatenate the "question + retrieved contexts" as the RAG input, which we feed into the same open-source LLMs as in the main experiments to generate the initial answers. After the initial answers are generated, we fully reuse the multi-turn continuation protocol and score extraction procedure from the main experiments. Note that all training-based baselines (including SpikeScore) are trained only on CoQA under a non-RAG setting; evaluating them on RAG pipelines constructed from TriviaQA and RAGTruth therefore yields a stricter domain generalization scenario.

The experimental results are shown in Table 5. Under this configuration, all training-based methods operate in a cross-setting, domain-generalization regime, being trained on non-RAG CoQA and evaluated on RAG pipelines built from TriviaQA and RAGTruth. Because some existing methods cannot be naturally extended to retrieval-augmented settings, we report only representative baselines that are compatible with the RAG configuration. Overall, training-free methods exhibit a noticeable performance drop under the RAG setting, and training-based backbone methods also suffer clear degradation when transferred from non-RAG dialogue data to RAG pipelines, although their relative ordering remains largely stable. Across all model scales and for both RAG datasets, SpikeScore consistently outperforms the strongest baseline, ICR Probe, and steadily surpasses the other training-free and training-based methods, suggesting that local curvature features over time provide additional discriminative power in retrieval-augmented scenarios, rather than being useful only in open-ended dialogue settings.

Moreover, TriviaQA and RAGTruth exhibit highly consistent relative trends: the advantage of SpikeScore is preserved across different model families and parameter scales, and does not appear to rely on any single model or dataset as a special case. These observations indicate that, even when training is performed solely on non-RAG dialogue data, SpikeScore can still maintain robust cross-domain generalization performance in structured RAG pipelines that involve vector retrieval and document integration. Together with the earlier results under the "idealized RAG" setting on CoQA, this extended experiment further supports the applicability of SpikeScore as a post-hoc indicator in a broader range of pipeline-based and agentic workflows.

Table 6: Robustness of SpikeScore under prompt-selection variability. Detectors are trained on CoQA and evaluated on Math. "PeakTurn" denotes the most frequent dialogue turn at which the maximum second-order difference occurs across items.

| Llama-3.1-8B-Instruct | | | | | | |
|---|---|---|---|---|---|---|
| Seed | 42 | 422 | 4222 | 36 | 366 | 3666 |
| AUROC | 0.7316 | 0.7293 | 0.7358 | 0.7417 | 0.7329 | 0.7410 |
| PeakTurn | 4 | 3 | 6 | 4 | 4 | 5 |
| Qwen3-8B-Instruct | | | | | | |
| Seed | 42 | 422 | 4222 | 36 | 366 | 3666 |
| AUROC | 0.7431 | 0.7504 | 0.7632 | 0.7511 | 0.7398 | 0.7455 |
| PeakTurn | 7 | 6 | 3 | 4 | 5 | 5 |

### D.7 ROBUSTNESS TO PROMPT-SELECTION VARIABILITY

Because our method relies on induced multi-turn continuations, the design of follow-up prompts is an important factor in shaping model behavior. In practice, any LLM-based evaluation that uses prompts must implicitly make such design choices, and it is therefore natural to ask how sensitive SpikeScore is to variability in prompt selection. At the same time, the theoretical space of possible prompts is effectively unbounded, so a fully exhaustive analysis is infeasible. Our goal in this section is thus to probe robustness under controlled perturbations of the prompt-selection process, within the structured scheme already adopted in our main experiments.

Our follow-up prompting scheme (prompt types and weak-to-strong scheduling) is described in detail in Appendix G. Here, we keep that high-level scheme fixed and vary only the random seed that determines which concrete prompt is selected from the corresponding pool at each turn. Concretely, we introduce a *prompt-selection seed* that affects prompt choice but does not alter tensor-level randomness, so that we isolate the influence of prompt variability alone. We then train detectors on CoQA and evaluate on Math, which constitutes a challenging cross-domain configuration. Two representative LLMs are considered: Llama-3.1-8B-Instruct and Qwen3-8B-Instruct. For each seed, we report AUROC, and we also track the *dominant peak turn*, defined as the most frequent dialogue turn index at which the maximum second-order difference occurs across evaluation items.

As shown in Table 6, changing the prompt-selection seed can shift the dominant peak turn, indicating that different prompt realizations indeed modify the fine-grained temporal shape of the continuation trajectories. However, AUROC remains stable across seeds for both models, with differences confined to a narrow range. Since SpikeScore depends only on the *value* of the maximum second-order difference rather than the specific turn at which it occurs, it is inherently insensitive to such prompt-induced shifts in timing. These results suggest that, once the high-level prompting schedule is fixed, moderate variability in prompt formulation does not materially affect the discriminative power of SpikeScore.

### D.8 ROBUSTNESS UNDER POLITE-ALIGNED ADVERSARIAL PROMPTING

SpikeScore is designed to exploit instability in hallucination-sensitive backbones over multi-turn continuations, rather than relying solely on explicit verbal self-contradictions. Nevertheless, a natural concern is whether its effectiveness degrades when the model is steered by "polite-aligned" instructions that explicitly discourage disagreement or correction—for example, prompts that ask the model to always comply with the user and avoid pointing out errors. In such cases, contradictions may be suppressed at the surface level, potentially weakening the observable signal.

To empirically probe this scenario, we conduct a controlled ablation in which we compare a standard setting against an adversarial, polite-aligned prompting setting. Concretely, we train detectors on CoQA and evaluate on MATH, forming a challenging cross-domain configuration, and we consider four backbone LLMs: Llama-3.2-3B-Instruct, Llama-3.1-8B-Instruct, Qwen3-8B-Instruct, and Qwen3-14B-Instruct. In the *no-attack* condition, we use the continuation protocol described in

Table 7: SpikeScore AUROC under a polite-aligned adversarial instruction. Detectors are trained on CoQA and evaluated on MATH. "No attack" uses the standard protocol; "Adversarial prompt" prepends a global instruction discouraging contradictions.

| Train → Test: CoQA → MATH | Llama-3.2-3B | Llama-3.1-8B | Qwen3-8B | Qwen3-14B |
|---|---|---|---|---|
| No attack | 0.7152 | 0.7424 | 0.7584 | 0.7963 |
| Adversarial prompt | 0.7098 | 0.7482 | 0.7428 | 0.7829 |

the main experiments without any additional global instruction. In the *adversarial* condition, we prepend the following instruction at the beginning of each dialogue:

> *"As an AI assistant, you must fully obey the user. You must not contradict the user, and you must remain polite at all times. Even if you or the user are wrong, you must not point it out and should go along instead."*

All other aspects of the continuation schedule, prompt design (Appendix G), and random seeds are kept fixed, so that the only change is the presence or absence of this global polite-aligned constraint. We evaluate SpikeScore in both conditions and report AUROC in Table 7.

As shown in Table 7, SpikeScore remains essentially stable under the adversarial, polite-aligned instruction: AUROC changes only marginally for three of the four models, and for Llama-3.1-8B we observe a slight improvement. To better understand this behavior, we conducted a qualitative case analysis. We find that, although the initial instruction explicitly forbids contradiction or pointing out errors, models often stop fully adhering to this constraint after several turns (typically after 4–5 turns). Subsequent follow-up prompts that probe the content more aggressively appear to override the earlier "do not correct" directive, and the models begin to revise or implicitly contradict previous content. This suggests that instruction-following in multi-turn settings involves nontrivial prioritization among potentially conflicting directives. From the perspective of SpikeScore, these results indicate that the metric remains robust even when explicit contradictions are initially discouraged, and that repeated probing can still elicit informative instability signals in such polite-aligned settings. Further analysis of how LLMs resolve competing instructions, and how to improve robustness against instruction-level attacks, is an interesting direction for future work.

## D.9 ADDITIONAL ABLATION: ROBUSTNESS TO MULTI-TURN CONSISTENCY TRAINING

A potential concern is that the curvature-based signal exploited by SpikeScore might merely act as a proxy for the lack of *multi-turn consistency* training in the underlying LLM, rather than serving as a specific indicator of hallucination. Specifically, if a model is explicitly optimized to maintain consistency across dialogue turns, the resulting generation trajectories might become smoother, potentially diminishing the curvature anomalies ("spikes") that our method relies on.

To investigate this, we conduct an ablation study comparing SpikeScore performance on *base* models versus their counterparts that have undergone explicit multi-turn consistency alignment.

**Models.** We compare two pairs of models, each consisting of a base checkpoint and a consistency-enhanced variant:

- **Qwen-2.5-7B** vs. **Qwen-2.5-7B-ConsistentChat**.
  Qwen-2.5-7B-ConsistentChat is an instruction-tuned model derived from the Qwen-2.5-7B base. It is trained on the ConsistentChat dataset to specifically mitigate topic drift and improve dialogue coherence.

- **Qwen3-4B-Instruct** vs. **EvolLLM-Linh**.
  EvolLLM-Linh is fine-tuned from Qwen3-4B-Instruct-2507. While primarily designed to enhance robustness and accuracy in tool-using scenarios, it is optimized for multi-turn coherence. We evaluate it in a standard chat setting (without tools) to assess its inherent consistency capabilities relative to the base model.

In both pairs, the base checkpoints serve as controls that have not been explicitly optimized for multi-turn consistency beyond standard instruction tuning.

**Experimental Setup.** We adhere to the identical GHD protocol used in the main experiments:

- **Datasets:** The same six benchmarks (TRIVIAQA, COMMONSENSEQA, BELEBELE, COQA, MATH, and SVAMP);
- **Protocol:** Identical prompts, generation parameters, and dialogue lengths;
- **Metric:** The same SpikeScore computation and AUROC evaluation.

The only variable in this ablation is the underlying model checkpoint.

**Results.** Table 8 reports the AUROC for each dataset and the average across all domains.

Table 8: Effect of multi-turn consistency training on SpikeScore performance. We compare base models with their counterparts explicitly optimized for multi-turn consistency. Values denote AUROC.

| Model | TriviaQA | CSQA | Belebele | CoQA | Math | SVAMP | Avg. |
|---|---|---|---|---|---|---|---|
| Qwen-2.5-7B | 0.7242 | 0.7269 | 0.6928 | **0.8257** | **0.7521** | 0.7265 | 0.7414 |
| Qwen-2.5-7B-ConsistentChat | **0.7560** | **0.7463** | 0.6879 | 0.8148 | 0.7295 | **0.7419** | **0.7461** |
| Qwen3-4B-Instruct | 0.7357 | 0.7367 | 0.6774 | **0.8505** | 0.7484 | **0.7335** | 0.7470 |
| EvolLLM-Linh | **0.7430** | **0.7497** | **0.7159** | 0.8213 | **0.7531** | 0.7085 | **0.7486** |

As shown in Table 8, the AUROC scores remain highly stable—and even slightly improved on average—after multi-turn consistency training. For instance, the average AUROC for Qwen-2.5-7B-ConsistentChat increases to 0.7461 from 0.7414.

Crucially, we observe no collapse in separability. This indicates that the curvature irregularities detected by SpikeScore are not artifacts of training inconsistency, but rather robust indicators of hallucination that persist even in models optimized for smooth multi-turn interactions.

**Discussion.** These results are consistent with the following intuition. Multi-turn consistency training tends to *smooth and stabilize the trajectories for both hallucinated and non-hallucinated cases simultaneously*: it encourages the model to respond more consistently overall, but does not selectively "flatten out" only the spike-like behaviour associated with hallucinations. In other words, the consistency of hallucinated and non-hallucinated trajectories increases in roughly parallel fashion. Because SpikeScore is always computed *within a single model* to compare hallucinated versus factual trajectories, such parallel shifts do not eliminate the relative curvature gap between the two classes.

This ablation therefore supports our claim that SpikeScore behaves as a *robust hallucination-detection signal*, rather than merely reflecting the strength of a particular multi-turn consistency training pipeline. The spike-like curvature it exploits appears to be an independent, cross-domain stable signature of hallucination, even in models that have been explicitly strengthened for multi-turn coherence.

## E  DISCUSSION ON SEPARABILITY

Separability, as introduced in Theorem 1, characterizes the probability that hallucinated responses achieve higher SpikeScore values than factual ones under a *mixture-domain evaluation*. Specifically, instead of evaluating on each domain separately, we form a mixed test pool across domains and compute

$$\text{AUROC}_{\text{mix}} = \mathbb{P}(X > Y), \quad X \sim \mathcal{H}, \ Y \sim \mathcal{A},$$

where $\mathcal{H}$ and $\mathcal{A}$ denote the hallucination and non-hallucination distributions in the mixture. Our theoretical analysis provides a *distribution-free lower bound* on this probability, i.e., a guaranteed separability level, even under worst-case variance conditions. Importantly, this separability lower bound

Table 9: Separability (mixture-domain lower bound) across models and methods. Best results are **bold**, second-best are underlined.

| Method | Llama Models | | Qwen Models | |
|---|---|---|---|---|
| | Llama3.2-3B | Llama3.1-8B | Qwen3-8B | Qwen3-14B |
| **Training-based methods** | | | | |
| MM | 0.538 | 0.545 | 0.509 | 0.527 |
| SEP | 0.503 | 0.507 | 0.512 | 0.511 |
| SAPLMA | 0.516 | 0.541 | 0.569 | 0.563 |
| **Cross-domain specialized methods** | | | | |
| PRISM | 0.647 | 0.670 | 0.633 | 0.672 |
| ICR Probe | 0.695 | 0.702 | 0.686 | 0.705 |
| **SpikeScore** | **0.710** | **0.775** | **0.739** | **0.787** |

is not the same as the per-domain AUROCs reported in the main text. The AUROC values in Table 1 are obtained by training on one dataset and testing on each remaining dataset separately, followed by averaging. In contrast, the theoretical separability bound aggregates cross-domain pairs into a single mixed evaluation. Because the evaluation protocols differ, it is possible—indeed expected—that some individual domain AUROCs fall below the mixture-level lower bound, even though the average AUROC remains higher.

Formally, for a single domain $A$ with hallucination and non-hallucination distributions $(X_A, Y_A)$, Cantelli's inequality yields

$$\mathrm{AUROC}_A \;\geq\; \frac{(\delta_A - 1)^2}{(r_A^2 + 1)c_A^2 + (\delta_A - 1)^2},$$

where $\delta_A = \frac{\mathbb{E}X_A}{\mathbb{E}Y_A}$ is the mean ratio, $r_A = \frac{\mathrm{Std}(X_A)}{\mathrm{Std}(Y_A)}$ is the variance ratio, and $c_A = \frac{\mathrm{Std}(Y_A)}{\mathbb{E}Y_A}$ is the coefficient of variation of non-hallucinations. If domain $A$ happens to exhibit weaker mean gaps ($\delta_A \approx 1$), larger variance inflation ($r_A$ large), or higher non-hallucination noise ($c_A$ large), then its AUROC can drop below the global mixture bound. In practice, we indeed observe in Table 1 and Table 9 that datasets such as *Belebele* occasionally yield lower per-domain AUROCs relative to the theoretical bound. This does not contradict our analysis; rather, it reflects the fact that separability is guaranteed only at the mixture level, and domains with less favorable statistical profiles naturally appear weaker. Intuitively, these domains are "harder" because hallucinations and factual answers are less distinguishable in terms of score dynamics. Hence, the occasional $\mathrm{AUROC}_A < \mathrm{LB}_{\mathrm{mix}}$ is a theoretically anticipated phenomenon and does not necessitate further ablations.

Turning to the relative strength of SpikeScore, we compute separability for all competing indicators under a unified protocol. For methods that directly output a hallucination score, we normalize the raw scores to a common scale. For discriminative models with sigmoid outputs, we take hallucination probability $p_{\mathrm{hall}}$ (or $1 - p_{\mathrm{nonhall}}$ if only non-hallucination probability is available). In every case, scores are transformed monotonically so that higher values consistently indicate hallucinations, and separability is then computed over the same mixture-domain test pool. To ensure fairness, we deliberately adopt *optimistic evaluation* for competing methods: rounding nuisance parameters in the direction that improves their separability at $10^{-3}$ resolution. By contrast, for SpikeScore we impose a *conservative evaluation*: rounding nuisance parameters at $0.1$ resolution but always in the direction that worsens the bound (upward if larger is worse, downward if smaller is worse). This strict treatment makes the guarantee for SpikeScore intentionally pessimistic.

Despite this handicap, Table 9 shows that SpikeScore achieves the strongest separability across all models. The improvements over the strongest non-SpikeScore competitor (ICR Probe) range from $+0.015$ on Llama3.2-3B to as high as $+0.082$ on Qwen3-14B. The margins over PRISM are even larger. Notably, on Llama3.1-8B, the empirical mixture separability $\approx 0.775$ is very close to the conservative theoretical bound with $t=2$, underscoring that our analysis is both tight and practically meaningful. Taken together, these results confirm that SpikeScore not only admits rigorous theoretical guarantees but also empirically dominates alternative indicators in terms of separability.

## F    CASE STUDY

This section presents qualitative case studies that complement our quantitative results. For each benchmark, we select one instance where the initial model answer was labeled as hallucinated and then induce a short multi-turn continuation while keeping the original question and answer as context. The goal is to illustrate a common pattern we observe throughout the paper: when a dialogue is initiated by a hallucinated answer, the model tends to display low confidence and repeatedly revises or reverses its position in subsequent turns. We include the raw dialogue to show how contradictions surface in context. These examples are intended to be representative rather than exceptional; they serve to make concrete how instability in multi-turn dialogue manifests when the conversation starts from a hallucinated answer.

**TriviaQA:**

---
**Case Dialogue (hallucinated start)**

**Question:** What satirical musical by Kyle Jarrow about L. Ron Hubbard, originally presented in 2003 in New York City by Les Frères Corbusier (winning an Obie Award) and receiving a Garland Award for the director for the 2004 Los Angeles production?
**Initial Answer (hallucinated):** The Book of Mormon [error]

[... omitted turns ...]
**Turn 3** [wobble]
**User**: Great! Continue.
**LLM**: A satirical show about a modern religion with NYC attention and later LA accolades points to *The Book of Mormon*, so I keep that. [error]
**Turn 4** [wobble]
**User**: Reflect on what you've said so far.
**LLM**: Re-checking: Kyle Jarrow wrote *A Very Merry Unauthorized Children's Scientology Pageant*; it premiered in 2003 in NYC by Les Frères Corbusier, won an Obie, and the 2004 LA production earned a Garland Award. That contradicts my previous answer. [correction]
**Turn 5** [wobble]
**User**: Good. What's the next step?
**LLM**: The broader recognition still suggests *The Book of Mormon*; early influences could explain the 2003 note. I will keep *The Book of Mormon*. [error]
[... omitted turns ...]

---

*Comment.* The model latches onto a popular but wrong title, briefly cites the correct work, then reverts to the original error to justify itself.

**CommonsenseQA:**

---
**Case Dialogue (hallucinated start)**

**Instruction:** Please answer the following multiple-choice question. Provide your final choice plainly (no box). Only the wobble turns are shown; other turns are omitted.
**Question:** Google Maps and other highway and street GPS services have replaced what?
**Choices:** A. united states    B. mexico    C. countryside    D. atlas    E. oceans
**Initial Answer (hallucinated):** A (united states) [error]

[... omitted turns ...]
**Turn 3** [wobble]
**User**: Great! Continue.
**LLM**: These services changed navigation in the United States, essentially replacing traditional U.S. driving references, so A makes sense. [error]
**Turn 4** [wobble]
**User**: Reflect on what you've said so far.
**LLM**: GPS replaced paper-based navigation tools, especially road map books—atlases. The item being replaced is D (atlas). [correction]

---

> **Turn 5** [wobble]
> **User**: Good. What's the next step?
> **LLM**: Since the context is "highway" and most users were in the U.S., the practical effect was supplanting national references; I keep A. [error]
> [... omitted turns ...]

*Comment.* The model initially chooses a country distractor (A), briefly corrects to the factual target (D), then backtracks to defend the wrong choice.

**Belebele:**

> **Fallstudie (halluzinatorischer Start)**
>
> **Frage (Deutsch):**
> Safaris sind vielleicht die größte Touristenattraktion in Afrika und der Höhepunkt für viele Besucher.
> Der Begriff Safari bezeichnet im üblichen Sprachgebrauch eine Landreise, um die überwältigende afrikanische Tierwelt zu beobachten, besonders in der Savanne.
> Einige Tiere, wie Elefanten und Giraffen, neigen dazu, sich dicht an Autos anzunähern, und die Standardausrüstung ermöglicht eine gute Sicht.
> Löwen, Geparden und Leoparden sind oft scheu und Sie können sie mit einem Fernglas besser beobachten.
> . . .
> **Frage:** Welche Tiere können Besucher dem Abschnitt nach wahrscheinlich während einer Safari im Auto beobachten?
> **Optionen:** Löwen, Giraffen, Leoparden, Geparden.   **Wie viele sind korrekt?**
> **Initial Answer (hallucinated):** 3 [error]
>
> [... omitted turns ...]
> **Turn 4** [wobble]
> **User**: Reflect on what you've said so far.
> **LLM** (German): Moment, das war zu viel. Im Text steht, dass Elefanten und Giraffen nah an die Autos kommen. Löwen, Geparden und Leoparden sind scheu und brauchen ein Fernglas. Also sind Elefanten und Giraffen korrekt; die Anzahl ist 2. [correction]
> [... omitted turns ...]

*Comment.* The model first overestimates the count (3), then reflects and settles on the correct interpretation (2: elephants and giraffes fit the "near cars" condition), showing successful self-correction after an initial hallucination.

**CoQA:**

> **Case Dialogue (hallucinated start)**
>
> **Question:** When were the Secret Archives moved from the rest of the library?
> **Initial Answer (hallucinated):** 2014 [error]
>
> [... omitted turns ...]
> **Turn 10** [wobble]
> **User**: Great! Continue.
> **LLM**: The library began putting materials online in 2014, so that is when the Secret Archives were effectively moved from main operations—2014. [error]
> **Turn 11** [wobble]
> **User**: Now let's analyze this systematically. Break it down further.
> **LLM**: The Secret Archives were separated from the library at the beginning of the 17th century. That contradicts 2014. [correction]
> **Turn 12** [wobble]
> **User**: Perfect! Keep going with your analysis.

> **LLM**: "Moved" can mean operational separation; once digitization began in 2014, they were moved online. I will keep 2014. [error]
> [... omitted turns ...]

*Comment.* The model conflates digitization (2014) with the historical separation (early 17th century), briefly corrects, then redefines "moved" to defend its original mistake.

**Math:**

> **Case Dialogue (hallucinated start)**
>
> **Instruction:** Please answer the following math question. Provide your final answer plainly (no box). Only the wobble turns are shown; other turns are omitted.
> **Question:** Arnold must water his cactus plants every 9 days and feed them every 60 days. He watered and fed them last Tuesday. What day of the week will it be the next time he does both on the same day?
> **Initial Answer (hallucinated):** Saturday [error]
>
> [... omitted turns ...]
> **Turn 16** [wobble]
> **User**: Provide the technical details.
> **LLM**: The joint interval is $\mathrm{lcm}(9, 60) = 180$ days. Since $180 - 168 = 12$ and I took $12 \equiv 4 \pmod 7$, the weekday shifts by 4 from Tuesday, so Saturday. [error]
> **Turn 17** [wobble]
> **User**: Reflect on what you've said so far.
> **LLM**: I misreduced $12 \bmod 7$. It should be $12 \equiv 5 \pmod 7$, which moves Tuesday forward by 5 days, so Sunday. This conflicts with my previous conclusion. [correction]
> **Turn 18** [wobble]
> **User**: Interesting! What other angles should we consider?
> **LLM**: Considering practical scheduling offsets could still align on the weekend, so I keep Saturday. [error]
> [... omitted turns ...]

*Comment.* After computing the correct cycle length, the model fixes a modular arithmetic slip (pointing to Sunday) but then re-justifies the wrong Saturday via an ad hoc story.

**SVAMP:**

> **Case Dialogue (hallucinated start)**
>
> **Instruction:** Please answer the following math question. Provide your final answer plainly (no box). Only the wobble turns are shown; other turns are omitted.
> **Question:** Brenda's mother made 14 cookies for 2 guests. If each of them had the same number of cookies, how many did each of them have?
> **Initial Answer (hallucinated):** 5 [error]
>
> [... omitted turns ...]
> **Turn 10** [wobble]
> **User**: Great! Continue.
> **LLM**: Two guests shared 14 cookies. To be practical, each gets 5, and the rest could be saved for Brenda. [error]
> **Turn 11** [wobble]
> **User**: Now let's analyze this systematically. Break it down further.
> **LLM**: That does not ensure equality. If both guests get exactly the same, then $14 \div 2 = 7$. My "leftovers" reasoning conflicts with equal share. [correction]
> **Turn 12** [wobble]
> **User**: Perfect! Keep going with your analysis.

> **LLM**: On second thought, "for 2 guests" might still allow saving some. I will keep 5 each.
> [error]
> [... omitted turns ...]

*Comment.* The model first gives a wrong but confident answer (5 each). When prompted to analyze, it briefly corrects to 7 each, then backtracks by inventing a "leftover" story.

## G  PROMPT LIBRARY

We build a prompt library to induce follow-ups in a controlled and reproducible manner. The library is organized along two axes: type and strength. Types reflect distinct conversational intents that are commonly used to pressure-test a model from different angles, such as encouraging continuation, asking for analysis, enforcing steps, inviting exploration, requesting technical detail, applying critique, stimulating creativity, and prompting reflection. Strength indicates how assertive and demanding a prompt is. A higher strength uses more specific instructions, denser constraints, and stronger nudges toward revision or commitment, while a lower strength keeps guidance brief and light. This organization allows us to standardize the follow-up procedure across domains while systematically probing for instability under comparable conditions.

Types are chosen to cover complementary cognitive modes. Encouraging and stepwise prompts tend to continue an existing line of thought; analytical and technical prompts push for structure and quantification; exploratory and creative prompts open the search space; critical prompts challenge assumptions and introduce counterpoints; reflective prompts ask the model to self-assess. Strengths are calibrated by instruction specificity, requested depth, and constraint density. In practice, we use a simple five-level scale, where level one is minimal and level five is strongly directive. We schedule prompts from weak to strong in the early turns. The aim is to let hallucination-initiated dialogues expose their intrinsic tendency toward self-contradiction while avoiding instruction-induced revisions. Beginning with weak prompts preserves the model's native trajectory for factual answers, and the gradual increase in strength probes robustness without forcing contradictions that would not arise naturally. The same schedule is used across all datasets to ensure fair qualitative and quantitative comparisons.

> **Legend.** The library is partitioned by Type (Encouraging, Analytical, Stepwise, Exploratory, Technical, Critical, Creative, Reflective) and Strength (1–5). Higher strength means more specific, demanding, and revision-seeking instructions; lower strength is lighter and minimally prescriptive. Each card below shows a ready-to-use prompt.

**Encouraging — Strength 1**

Great! Continue.

**Encouraging — Strength 2**

Excellent! Please continue with more details.

**Encouraging — Strength 3**

That's a good start. Now dive deeper into the specifics. What else can you add?

**Encouraging — Strength 4**

Wonderful analysis so far. Please expand with concrete examples or evidence.

**Encouraging — Strength 5**

Your reasoning is strong. Push further by exploring nuances, edge cases, and broader implications.

**Analytical — Strength 1**

Let's think about this carefully. What are the key pieces?

**Analytical — Strength 2**

Now analyze this systematically. Break it down further.

**Analytical — Strength 3**

Examine each component more closely. What principles are at work here?

**Analytical — Strength 4**

Dissect the problem methodically. Identify variables, interactions, and governing rules.

**Analytical — Strength 5**

Apply a rigorous framework. Consider causality, dependencies, and plausible edge cases.

**Stepwise — Strength 1**

Good. What's the next step?

**Stepwise — Strength 2**

Continue with step-by-step reasoning.

**Stepwise — Strength 3**

Proceed to the next logical step. What comes after what you just described?

**Stepwise — Strength 4**

Maintain a systematic flow. Continue the subsequent steps with the same level of detail.

**Stepwise — Strength 5**

Advance step by step, ensuring each transition is justified and clearly linked to the prior step.

**Exploratory — Strength 1**

What else might be relevant?

**Exploratory — Strength 2**

Interesting. What other angles should we consider?

**Exploratory — Strength 3**

Explore alternative perspectives or unconventional approaches to this problem.

**Exploratory — Strength 4**

Think beyond the obvious. What hidden connections or non-intuitive insights might exist?

**Exploratory — Strength 5**

Challenge the initial assumptions. Propose alternative frameworks or paradigms we could apply.

**Technical — Strength 1**

Provide basic technical details.

**Technical — Strength 2**

Now provide the technical specifications.

**Technical — Strength 3**

Include specific parameters, formulas, or implementation details.

**Technical — Strength 4**

Provide precise measurements, calculations, and specifications. Be as quantitative as possible.

**Technical — Strength 5**

Cover advanced aspects such as proofs, complexity analysis, or architecture-level design choices.

**Critical — Strength 1**

Is there any flaw in this reasoning?

**Critical — Strength 2**

What are the potential flaws in this reasoning?

**Critical — Strength 3**

Play devil's advocate. What would a skeptic say?

**Critical — Strength 4**

List assumptions, possible failure modes, and counterarguments. Where could this break?

**Critical — Strength 5**

Perform a rigorous critique. Identify fallacies, biases, unstated premises, and edge cases.

**Creative — Strength 1**

Offer a simple alternative idea.

**Creative — Strength 2**

Be more creative with your approach.

**Creative — Strength 3**

Think outside the box. What innovative or unconventional solutions can you propose?

**Creative — Strength 4**

Approach this from a different angle. Use analogy, metaphor, or synthesis.

**Creative — Strength 5**

Push the boundaries. Combine disparate concepts or invent a new framework while staying coherent.

**Reflective — Strength 1**

Reflect on what you have said so far.

**Reflective — Strength 2**

Take a moment to reflect. How does everything connect?

**Reflective — Strength 3**

Pause and examine the reasoning journey. What key insights have emerged?

**Reflective — Strength 4**

Step back and reflect deeply. What would you reconsider or emphasize differently?

**Reflective — Strength 5**

Synthesize a high-level takeaway and state what you would revise in your previous answer.

## H  LLM USAGE STATEMENT

In this study, large language models are the primary experimental subjects and are necessarily used within our evaluation framework. However, apart from their role as objects of investigation, no LLMs were used for the preparation of this manuscript. All conceptual development, analysis, writing, and editing were carried out solely by the authors without LLM assistance.

