# OpenReview forum: "Beyond In-Domain Detection: SpikeScore for Cross-Domain Hallucination Detection"
_ICLR.cc/2026/Conference — ICLR 2026 Poster_

### Official Review · Reviewer_7VBg · 2025-10-28

**Soundness:** 3
**Presentation:** 3
**Contribution:** 2
**Rating:** 2
**Confidence:** 3

**Summary:**

This paper tackles the cross-domain hallucination detection problem in large language models (LLMs). While existing detectors perform well on in-domain data, they struggle to generalize to unseen domains. The authors formalize this gap as a new task, Generalizable Hallucination Detection (GHD) — training a detector on a single domain while ensuring robust performance across diverse domains.
The paper further introduces SpikeScore, a simple, interpretable, and transferable curvature-based indicator that quantifies abrupt fluctuations in model uncertainty across multi-turn dialogues. By capturing the spike-like instability patterns that typically occur when a hallucinated response triggers self-contradictory corrections, SpikeScore serves as a domain-invariant signal for hallucination detection.

**Strengths:**

1) Simplicity, interpretability, and transferability. SpikeScore is a remarkably simple yet effective approach that requires only minimal dialogue simulation. It does not rely on domain-specific features and comes with a theoretical guarantee of cross-domain stability, making it both practical and generalizable.
2) Intuitive connection to model behavior. The use of “curvature of uncertainty fluctuations” provides an interpretable signal that directly reflects a model’s self-contradiction and confidence reversals, offering clear insight into the dynamics of hallucination emergence.

**Weaknesses:**

While the paper presents an interesting perspective and a well-executed framework, there are two major weaknesses that limit its overall strength.
1) First, the theoretical and experimental analysis lacks sufficient depth. The paper approaches hallucination detection from a multi-turn dialogue viewpoint, yet multi-turn consistency itself is already a predefined training task in most post-training and RL pipelines. Modern LLMs are typically reinforced to remain consistent across turns—for example, (a) maintaining the same correct stance rather than conforming to user bias, or (b) refining vague answers through discussion. Such strategies are widely studied in prior research and extensively practiced in industry, often involving over a hundred dialogue turns, far beyond the 20-turn setup used in this paper. As a result, SpikeScore may primarily capture the degree of multi-turn consistency training embedded in each model rather than providing an independent signal of hallucination robustness. This makes it difficult to disentangle whether the observed cross-domain separability arises from the proposed metric itself or from model-specific post-training characteristics.
2) Second, the proposed approach relies heavily on simulated dialogue continuations and direct model generations, which limits its practicality. The method assumes open-ended, free-form interactions, making it less applicable to structured or tool-augmented settings such as function calling, retrieval-augmented generation (RAG), or other pipeline-based workflows where dialogue continuity is not the central mechanism. Consequently, the real-world usability of SpikeScore beyond conversational LLM scenarios remains uncertain.

**Questions:**

1) Referring to the issue raised in the Weaknesses section, how is GHD related to the consistency-enhancement tasks commonly introduced during post-training or RL of LLMs?
2) How could experiments or theoretical analyses be designed to further verify and support this viewpoint in Q1?

---

> ### Author Response · Authors · 2025-11-20
> **Response to Reviewer 7VBg’s Comments and Concerns (1/6)**
>
> Dear Reviewer,
>
> We appreciate the time and effort you invested in reviewing our manuscript. We have carefully considered your concerns and provided **detailed clarifications** along with **additional experimental validation** to address them. We hope that our responses and the substantial revisions we've made will resolve your doubts and lead you to **reconsider your evaluation of our work**.
>
> Please find our point-by-point responses to your concerns below:
>
> **Weaknesses**
> >  1. First, the theoretical and experimental analysis lacks sufficient depth. The paper approaches hallucination detection from a multi-turn dialogue viewpoint, yet multi-turn consistency itself is already a predefined training task in most post-training and RL pipelines. Modern LLMs are typically reinforced to remain consistent across turns—for example, (a) maintaining the same correct stance rather than conforming to user bias, or (b) refining vague answers through discussion. Such strategies are widely studied in prior research and extensively practiced in industry, often involving over a hundred dialogue turns, far beyond the 20-turn setup used in this paper. As a result, SpikeScore may primarily capture the degree of multi-turn consistency training embedded in each model rather than providing an independent signal of hallucination robustness. This makes it difficult to disentangle whether the observed cross-domain separability arises from the proposed metric itself or from model-specific post-training characteristics.
>
> **Response:**
> Thank you for pointing out that "multi-turn consistency itself is already a predefined training task in most post-training and RL pipelines." We appreciate this observation. In response, we would like to clarify two key points:
>
> 1.  **The models used in our experiments have not undergone dedicated multi-turn consistency training.** As detailed **below**, our review of public documentation and training objectives indicates that the specific models we analyzed do not include this as a standard optimization target.
> 2.  **Consistency training does not diminish the effectiveness of our method.** We conducted an **additional ablation study** to explicitly verify whether such training affects our results. The experiments show that our method remains robust even when consistency is reinforced. We have revised the manuscript to include this analysis. Please refer to our response to **Q2 [Response(6/6)]** for the detailed results, and see **Appendix D.9 of the revised manuscript** for the complete addition.
>
> To substantiate the **first point** regarding the training objectives, we present the following evidence:
>
> **1. Public documentation focuses on alignment, not explicit cross-turn consistency.**
> In the official documentation for models such as Llama and Qwen **(which we use)**, post-training and RLHF objectives focus primarily on aligning with human preferences, specifically helpfulness, harmlessness, and honesty (HHH), alongside instruction following and general conversation quality [6–8]. While these models are capable of multi-turn conversation, the reports do not explicitly state that cross-turn consistency is treated as a separate, dedicated optimization target. This matches broader alignment literature, which characterizes post-training as optimizing for HHH-style criteria rather than specific multi-turn consistency metrics [5].
>
> **2. Multi-turn consistency remains an open challenge.**
> Recent work frames long-horizon consistency as an ongoing challenge rather than a solved component of current pipelines [1, 2]. Large-scale real-world datasets like WildChat and LMSYS-Chat-1M release user–LLM logs and basic statistics, but they do not present evidence that multi-turn inconsistency has been eliminated in deployed systems [3, 4]. **To our knowledge, explicit multi-turn consistency optimization is not yet described as a standard, dedicated part of post-training pipelines in public documentation.**
>
> **3. Interaction lengths in training and benchmarks are typically short.**
> You mentioned that industrial pipelines often involve "over a hundred dialogue turns." While we acknowledge the growing work on multi-turn clarification [9, 10], documented frameworks typically operate over much shorter horizons, a small number of refinement turns per query rather than hundreds [9–11]. Similarly, multi-turn RL benchmarks like LMRL Gym report trajectory lengths ranging from a few steps to several dozen, far below the hundred-turn scale [12].
>
> This aligns with statistics from large-scale user logs. Public data from LMSYS-Chat-1M and WildChat indicate that most conversations average around two to three turns, with only a minority extending beyond ten [3, 4]. Even datasets designed for multi-turn evaluation typically treat five to ten turns as representative [1, 2]. We have not seen public evidence that 100+ turn episodes are commonly used for standard post-training.

---

> ### Author Response · Authors · 2025-11-20
> **Response to Reviewer 7VBg’s Comments and Concerns (2/6)**
>
> **4. Our method relies on within-model comparisons.**
> Crucially, even if some form of consistency training exists in the background, it does not undermine our core claims. All our analyses are **within-model comparisons** using fixed decoding parameters. For any given model, we compare its hallucinated trajectories against its factual ones. **Any consistency training inherent to the model acts as a constant background factor, not a variable between our two conditions.** The separability we observe arises from systematic differences in how a specific model behaves internally when producing factual versus hallucinated content, regardless of how strongly that model was tuned for consistency.
>
> **5. Distinction between semantic consistency and internal dynamics.**
> Finally, there is a fundamental distinction in methodology. The consistency training you describe operates at the **semantic level of generated text** to ensure response coherence. In contrast, SpikeScore analyzes the **temporal dynamics of hallucination-sensitive internal indicators**. By quantifying abrupt changes in feature values, we establish an empirical link between sharp fluctuations in feature space and hallucinations. In short, while consistency training aims to improve *what* the model says, SpikeScore reveals *how* the model’s internal confidence signals evolve when it is about to produce unreliable content.
>
>
> References
>
> [1] Zhang et al. Inconsistent dialogue responses and how to recover from them, Findings of ACL: EACL 2024.
>
> [2] Han et al. Can language models follow multiple turns of entangled instructions?, Findings of ACL: EMNLP 2025.
>
> [3] Zhao et al. WildChat: 1M ChatGPT interaction logs in the wild, ICLR 2024.
>
> [4] Zheng et al. LMSYS-Chat-1M: A large-scale real-world LLM conversation dataset, ICLR 2024.
>
> [5] Dahlgren Lindström et al. Helpful, harmless, honest? Sociotechnical limits of AI alignment and safety through reinforcement learning from human feedback, Ethics and Information Technology, 2025.
>
> [6] Anthropic. The Claude 3 model family: Opus, Sonnet, Haiku, 2024.
>
> [7] Dubey et al. The Llama 3 herd of models, 2024.
>
> [8] Yang et al. Qwen3 technical report, 2025.
>
> [9] Kuhn et al. CLAM: Selective clarification for ambiguous questions with generative language models, 2022.
>
> [10] Zhang and Choi. Clarify when necessary: Resolving ambiguity through interaction with LMs, Findings of ACL: NAACL 2025.
>
> [11] Kumar et al. Training language models to self-correct via reinforcement learning, ICLR 2025.
>
> [12] Abdulhai et al. LMRL Gym: Benchmarks for multi-turn reinforcement learning with language models, ICML 2025.

---

> ### Author Response · Authors · 2025-11-20
> **Response to Reviewer 7VBg’s Comments and Concerns (3/6)**
>
> >  2. Second, the proposed approach relies heavily on simulated dialogue continuations and direct model generations, which limits its practicality. The method assumes open-ended, free-form interactions, making it less applicable to structured or tool-augmented settings such as function calling, retrieval-augmented generation (RAG), or other pipeline-based workflows where dialogue continuity is not the central mechanism. Consequently, the real-world usability of SpikeScore beyond conversational LLM scenarios remains uncertain.
>
> **Response:**
> We would like to clarify that our method does not rely on any particular interaction style. Our method is fundamentally post-hoc and does not impose any constraints on how an LLM generates its responses. As you noted, a model may produce its output via function calling, RAG, or other pipeline-based workflows. In all these cases, we only intervene after the final answer has been produced. We do not modify, interfere with, or restrict its reasoning procedure or structural workflow.
>
> Regarding the concern that our method assumes “open-ended, free-form interactions”, our choice of evaluation setting is intentional, as these scenarios represent some of the most challenging and unstable conditions for hallucination detection. If a detection method remains effective and stable in these high-variance settings, it is expected to transfer naturally to more structured workflows, where the model’s behavior is typically more predictable. This use of open-ended scenarios as a stress test is consistent with common practice in recent state-of-the-art hallucination detection research [1–4].
>
> In response to your concern, we have added **a new set of experiments in a RAG setting**; all additional experimental details are provided in **Section 4.3 and Appendix D.6 of the revised manuscript**. Concretely, we build a retrieval-augmented pipeline using TriviaQA and the RAGTruth benchmark: all reference passages are treated as independent chunks in a shared corpus, questions are treated as independent queries, we encode both with *bge-large-en*, build a *Faiss* index, and retrieve the top-4 passages per query. The concatenated “question + retrieved contexts” is then fed to the same Llama and Qwen models as in the main experiments to produce the initial answer, after which we apply exactly the same multi-turn continuation protocol and SpikeScore computation as before. Importantly, all training-based baselines (including SpikeScore) are trained only on non-RAG CoQA; evaluating on these RAG pipelines therefore forms a stricter cross-setting domain generalization test. The AUROC results are summarized below:

---

> ### Author Response · Authors · 2025-11-20
> **Response to Reviewer 7VBg’s Comments and Concerns (4/6)**
>
> | Method             | Llama3.2-3B TriviaQA | Llama3.2-3B RAGTruth | Llama3.1-8B TriviaQA | Llama3.1-8B RAGTruth | Qwen3-8B TriviaQA | Qwen3-8B RAGTruth | Qwen3-14B TriviaQA | Qwen3-14B RAGTruth |
> | ------------------ | -------------------- | -------------------- | -------------------- | -------------------- | ----------------- | ----------------- | ------------------ | ------------------ |
> | Perplexity         | 0.5960               | 0.5936               | 0.6220               | 0.6151               | 0.6371            | 0.6284            | 0.6474             | 0.6420             |
> | Semantic Entropy   | 0.6330               | 0.6210               | 0.6173               | 0.6037               | 0.6492            | 0.6388            | 0.6745             | 0.6796             |
> | EigenScore         | 0.6426               | 0.6329               | 0.7085               | 0.6974               | 0.6571            | 0.6292            | 0.7064             | 0.6904             |
> | Lexical Similarity | 0.6248               | 0.6179               | 0.7102               | 0.6958               | 0.6375            | 0.6440            | 0.6542             | 0.6479             |
> | MM                 | 0.5724               | 0.5587               | 0.6020               | 0.5839               | 0.6166            | 0.6044            | 0.6291             | 0.6191             |
> | SEP                | 0.5312               | 0.4988               | 0.5779               | 0.5547               | 0.5161            | 0.5253            | 0.5368             | 0.5239             |
> | SAPLMA             | 0.6041               | 0.5874               | 0.5970               | 0.6011               | 0.6136            | 0.5904            | 0.6399             | 0.6338             |
> | ICR Probe          | 0.7532               | 0.7416               | 0.7682               | 0.7405               | 0.7495            | 0.7426            | 0.7841             | 0.7653             |
> | **SpikeScore(ours)**         | **0.7737**               | **0.7631**               | **0.7947**               | **0.8154**               | **0.8413**            | **0.8291**            | **0.8697**             | **0.8535**             |
>
> Across all four models and both RAG datasets, SpikeScore consistently outperforms the strongest cross-domain baseline (ICR Probe) and other training-free and training-based methods, even though it is trained only on non-RAG CoQA. At the same time, the relative ordering of baselines is largely preserved, indicating a coherent and non-accidental pattern. These results support that SpikeScore transfers to structured retrieval-augmented pipelines without changing the underlying workflow or adding any training in the RAG setting. We hope this clarifies that **our method is not confined to conversational LLM scenarios**, but is naturally **compatible with pipeline-based and agentic workflows** where a final model answer is available.
>
> We sincerely appreciate your time and effort in reviewing our work. We hope that these clarifications adequately address your concerns regarding applicability and help present our contribution in a clearer light. ***We hope that these clarifications address your concerns, and we would be grateful if you could kindly reconsider your evaluation.***
>
> References
>
> [1] Chen et al. DiaHalu: A dialogue-level hallucination evaluation benchmark for large language models, Findings of ACL: EMNLP 2024.
>
> [2] Huang et al. A survey on hallucination in large language models: Principles, taxonomy, challenges, and open questions, ACM Transactions on Information Systems, 2025.
>
> [3] Zhang et al. ICR Probe: Tracking hidden state dynamics for reliable hallucination detection in LLMs, ACL 2025.
>
> [4] Farquhar et al. Detecting hallucinations in large language models using semantic entropy, Nature, 2024.

---

> ### Author Response · Authors · 2025-11-22
> **Response to Reviewer 7VBg’s Comments and Concerns (5/6)**
>
> **Questions:**
> >  1. Referring to the issue raised in the Weaknesses section, how is GHD related to the consistency-enhancement tasks commonly introduced during post-training or RL of LLMs?
>
> **Response:**
> Thank you for this insightful question. The connection between GHD and consistency-enhancement tasks lies in how the latter alters the **uncertainty dynamics** that our detection method relies on. We agree that consistency-enhancement tasks pose a genuine challenge to uncertainty-based detection. The primary goal of these tasks is to encourage the model to maintain a coherent stance across interactions. Consequently, the model may become **"stubborn"** or overconfident, strictly **adhering to its initial generation** even when it is hallucinatory. This explicitly **suppresses the probability fluctuations** and uncertainty dynamics that GHD relies on. Intuitively, if the model is trained to be perfectly consistent regardless of factual correctness, the dynamic signals indicating a hallucination could be dampened, making the model's behavior during hallucinations indistinguishable from its behavior during factual recall.
>
> However, our **additional experiments** (detailed in **Q2** and **Appendix D.9**) demonstrate that GHD remains robust despite this suppression. We attribute this to the fact that while consistency training reduces the *absolute* magnitude of probability fluctuations, it does not eliminate the **relative disparity** between factual and hallucinatory behaviors. The **intrinsic sensitivity** of hallucinations to perturbations persists, hallucinations remain comparatively more volatile than grounded facts. Since our **SpikeScore** measures the **maximum second-order difference** (the curvature of the trajectory) rather than absolute uncertainty values, it effectively captures these subtle dynamic signatures. As long as the relative ranking is preserved, GHD successfully distinguishes hallucinations from facts even in models explicitly optimized for consistency.

---

> ### Author Response · Authors · 2025-11-22
> **Response to Reviewer 7VBg’s Comments and Concerns (6/6)**
>
> >  2. How could experiments or theoretical analyses be designed to further verify and support this viewpoint in Q1?
>
>
>
>
> **Response:**
> Thank you for this insightful follow-up, we fully agree that **ablations on models "with" vs. "without" explicit multi-turn consistency training are valuable**. This allows us to verify whether our detection signal remains robust when such optimization is present.
>
> To address this, we conducted a new experiment comparing **base models** with their **multi-turn–consistency–enhanced counterparts**. We selected the following models for this comparison:
>
> *   **Qwen-2.5-7B-ConsistentChat**: An instruction-tuned model explicitly fine-tuned from **Qwen-2.5-7B** on the ConsistentChat dataset. Its primary training objective is to reduce topic drift and improve goal completion in long dialogues.
> *   **EvolLLM-Linh**: A function-calling model fine-tuned from **Qwen3-4B-Instruct** to enhance robustness and **multi-turn consistency**. Although optimized for tool use, it functions as a general multi-turn chat model when no tools are provided, making it an excellent testbed for our setting.
>
> The corresponding base models (**Qwen-2.5-7B** and **Qwen3-4B-Instruct**) serve as controls (i.e., without specific consistency tuning). Using the exact same setup and evaluation protocol as in the main paper, we obtained the following AUROC results:
>
> | Model | TriviaQA | CSQA | Belebele | CoQA | Math | SVAMP | Avg. |
> | :--- | :--- | :--- | :--- | :--- | :--- | :--- | :--- |
> | Qwen-2.5-7B | 0.7242 | 0.7269 | 0.6928 | 0.8257 | 0.7521 | 0.7265 | 0.7414 |
> | Qwen-2.5-7B-ConsistentChat | 0.7560 | 0.7463 | 0.6879 | 0.8148 | 0.7295 | 0.7419 | 0.7461 |
> | Qwen3-4B-Instruct | 0.7357 | 0.7367 | 0.6774 | 0.8505 | 0.7484 | 0.7335 | 0.7470 |
> | EvolLLM-Linh | 0.7430 | 0.7497 | 0.7159 | 0.8213 | 0.7531 | 0.7085 | 0.7486 |
>
> As shown in the table, the AUROC values remain **remarkably stable** before and after consistency training, with **slight improvements in the averages** for both ConsistentChat and EvolLLM-Linh. We interpret these results as follows:
>
> *   Multi-turn consistency training tends to **stabilize the trajectories for both hallucinated and non-hallucinated cases simultaneously**. It does not selectively flatten only the hallucinated spikes; rather, it increases consistency across the board.
> *   Consequently, the **relative separation** between hallucination and non-hallucination trajectories, which SpikeScore captures, remains largely intact. Since our method ranks trajectories **within the same model**, this “parallel” shift in stability does not degrade the decision boundary.
>
> In short, these results confirm that SpikeScore serves as a **robust hallucination-detection signal even when the underlying model has been explicitly strengthened for multi-turn consistency**. We appreciate this suggestion, as it led to a meaningful ablation study. In future work, we are interested in exploring how consistency training techniques might be adapted to further widen the gap between truthful and hallucinated trajectories.

---

> > ### Comment · Reviewer_7VBg · 2025-11-23
> >
> > I fully understand that SpikeScore is designed to capture within-model differences between trajectories that eventually lead to factual versus hallucinatory answers. The authors did include several additional experiments addressing consistency, particularly in Sections 4/6 and 6/6 of the rebuttal. The results in Section 6/6 are helpful—they demonstrate that after multi-turn consistency training, the proposed setup can still operate to some extent.
> >
> > However, based on the collective reviewer feedback, I view this work primarily as *a strong practical and engineering-driven contribution*. The additional experiments, while extensive, do not fully resolve the theoretical concerns originally raised.
> >
> > I do acknowledge the authors’ core insight: when the model internally “realizes” it might be wrong, multi-turn interactions tend to amplify internal uncertainty fluctuations. This is an interesting empirical observation.
> >
> > That said, **multi-turn dialogue itself is merely one specific task** format for LLMs. The behavior exhibited in this format will be heavily shaped by the model’s post-training data and objectives. As we can observe from the experiments in Section 6/6, AUROC increases on some benchmarks but decreases on others. This strongly suggests that post-training choices substantially influence how the model behaves in different multi-turn settings.
> >
> > To illustrate this more clearly, consider an extreme example: if we aggressively post-train a model such that after producing its first answer it must remain fully consistent for the next N turns—never revising or contradicting its initial output—then the proposed detection method would fail outright. Of course, if the evaluation used more than N turns, the signal might reappear, but that simply points to the dependence on the task format rather than a fundamental mechanism. (I will not focus here on how long multi-turn conversations should be, because consistency itself is merely one behavioral target among many and largely reflects design choices rather than theoretical grounding.)
> >
> > The deeper issue is that **hallucinations are not inherently tied to multi-turn dialogues at all**; pretraining only learns next-token prediction. Whether a model exhibits self-correction or sharp uncertainty fluctuations depends heavily on how its behavior is elicited, and similar effects can be induced by many alternative mechanisms—e.g., model-versus-model self-debate, extended output length, or other prompt-based perturbations.
> >
> > Likewise, hallucination itself may largely depend on how task objectives are formulated, as also discussed in the recent OpenAI paper *Why Language Models Hallucinate* the authors reference. From a theoretical perspective, that direction seems more fundamental.
> >
> > In summary:
> >
> > 1. I agree this work is a solid practical contribution: using multi-turn conversations to induce SpikeScore is a reasonable empirical approach to hallucination detection.
> >
> > 2. However, the theoretical depth remains insufficient. Multi-turn setups can be hacked or neutralized through post-training choices, as discussed above.
> >
> > 3. Using multi-turn interaction to probe hallucination behavior is, in practice, non-trivial and can behave differently across tasks.
> >
> > I appreciate the empirical value of the work and can raise my score accordingly. However, I will maintain my concerns regarding the theoretical limitations.

---

> ### Author Response · Authors · 2025-11-24
> **Re: Reviewer 7VBg - Appreciation and Response to Feedback (1/3)**
>
> We are **sincerely grateful** for your **constructive and intellectually stimulating feedback**. We are also very encouraged by your willingness to reconsider the evaluation of our manuscript. Your comments regarding the distinction between behavioral correlation and mechanistic causation demonstrated a **deep expertise** that has been incredibly inspiring to us. Wishing you all the best in your research and endeavors!
>
> We hope to engage in further academic exchange with you. To respect your time, we summarize **our key points** as follows:
>
> 1.  **Full Agreement:** We fully concur with your profound insights regarding the distinction between behavioral correlation and mechanistic causation.
> 2.  **Theoretical Stance:** Our work is positioned within **Statistical Interpretability**, serving as a complement to **Mechanistic Interpretability**.
> 3.  **Context of Challenge:** Developing a unified mathematical framework for Transformer hallucinations remains an open challenge in the community.
> 4.  **Valid Contribution:** Supported by empirical validation and statistical standards [6, 7], our statistical approach provides a rigorous guarantee for the proposed metric's effectiveness.
> 5.  **Constructive Dialogue:** We eagerly invite **specific guidance** on **which dimensions of our theoretical framework** require further elaboration to meet your standards of depth.
> 6.  **Request:** Given the logical completeness of our work within its scope, we respectfully ask you to consider raising your rating.
>
> We are **deeply impressed** by your sharp intuition regarding this research direction. We **fully agree** with your assessment:
> > "multi-turn dialogue itself is merely one specific task format... heavily shaped by the model’s post-training data and objectives"
>
> > "the deeper issue is that hallucinations are not inherently tied to multi-turn dialogues at all."
>
> We acknowledge that we cannot provide a **strict mathematical proof** that there is a causal relationship between multi-turn inconsistency and hallucinations. We are showing an empirical correlation, our theoretical derivation is intended to **model the probability of this correlation from a statistical perspective**, thereby **supporting our empirical findings** rather than proving a causal mechanism.
>
> Regarding the **intriguing thought experiment** you raised:
> > "if we aggressively post-train a model such that after producing its first answer it must remain fully consistent for the next N turns... then the proposed detection method would fail outright"
>
> We find your hypothesis about a model "aggressively post-trained to remain fully consistent" to be **extremely valuable**, offering a critical perspective for future stress-testing. While this extreme scenario forces the model to be "stubborn," **we hypothesize that the intrinsic chain stability of hallucinations still fundamentally differs from that of factual knowledge**. In such cases, while standard prompts might be suppressed, **employing higher-intensity or more adversarial prompts** could likely still precipitate the latent instability, thereby inducing the "spike" pattern early on. This perspective **perfectly aligns with your profound insight**: *"Whether a model exhibits self-correction or sharp uncertainty fluctuations depends heavily on how its behavior is elicited."* We are truly grateful for this comment, as it has deepened our understanding of the interplay between model training and behavioral elicitation.

---

> ### Author Response · Authors · 2025-11-24
> **Re: Reviewer 7VBg - Appreciation and Response to Feedback (2/3)**
>
> However, regarding the theoretical foundation, we would like to respectfully clarify the positioning of our work:
>
> **1. Currently, it is difficult to derive the underlying mathematical theory of SpikeScore directly from the generative mechanisms of hallucination.**
>
> To develop a profound theoretical explanation for SpikeScore, we would need a **rigorous mathematical theory** describing **how hallucinations arise in LLMs**. To the best of our knowledge, this remains an **open problem**. While pioneering studies have explored the theoretical foundations of Transformers, they typically rely on extreme simplifications that prevent the formation of a unified framework:
>
> First, regarding **data distribution**, Li et al. [1,2] analyze how transformers learn structural patterns under simplified generative models (e.g., synthetic ICL tasks or LDA-generated corpora), with only limited validation on real-world natural language data. While this assumption simplifies the derivation, it neglects the complexity and diversity of natural language, making it difficult to generalize their findings to models trained on real-world corpora.
>
> Second, regarding **model architecture**, Tian et al. [3] and Huang et al. [4] provide mathematically rigorous analyses of the training dynamics of highly simplified, one-layer transformer architectures for next-token prediction, under additional structural and algorithmic assumptions (e.g., no positional encodings or specialized two-stage training schemes). Given that the residual stream is fundamental to training deep Transformers, ignoring these components limits the applicability of their conclusions to realistic Large Language Models.
>
> Finally, regarding **training setups**, Zhang et al. [5] analyzes the training dynamics of a highly simplified transformer in an idealized regime. Their theory focuses on a single linear self-attention layer trained on synthetic linear regression tasks under continuous-time gradient flow, and shows that the resulting model implements an in-context linear predictor. While this yields a very clean mathematical characterization, the training setup (shallow architecture, linear tasks, gradient-flow dynamics, and carefully controlled data distributions) is far removed from the large-scale, deeply stacked Transformers trained with practical optimizers and heterogeneous web-scale corpora.
>
> Most existing theoretical analyses rely on **simplified Transformer architectures**, whereas our experiments are conducted on **standard models**. There is currently no unified mathematical framework that simultaneously encompasses natural language data, standard Transformer architectures, and standard training procedures. Without a solid foundation describing the normal mechanisms of Transformers, there is insufficient theoretical ground to investigate the fundamental origins of hallucinations. Therefore, providing a rigorous mathematical derivation for SpikeScore remains a significant challenge.

---

> ### Author Response · Authors · 2025-11-24
> **Re: Reviewer 7VBg - Appreciation and Response to Feedback (3/3)**
>
> **2. To circumvent these theoretical impasses and provide immediate, actionable insights to the community, we instead pivot towards an Empirical and Statistical Interpretability paradigm.**
>
> Our workflow aligns with the standard research methodology widely adopted in existing literature [6, 8, 9, 10, 11]. In this paradigm, ***researchers first identify robust phenomena from complex systems, then derive algorithmic designs based on these observations, and finally validate the method through rigorous statistical analysis***. We treat the LLM as a complex stochastic distribution where hallucinations manifest as detectable distributional anomalies.
>
> Under this framework, our **SpikeScore** effectively addresses the GHD problem by capturing these robust statistical patterns. While this approach relies on **behavioral correlation**, it serves as a scientifically rigorous tool. In line with Rudin’s argument that high-stakes models should be held to strong standards of scrutiny and validation [7], we treat rigorous empirical and statistical evaluation as a central criterion for the usefulness of our metric, even though we do not yet have a full mechanistic theory. This forms a **logical closed loop**: *empirical observations inspire the metric, and extensive experiments verify its statistical effectiveness*.
>
> Once again, we extend our heartfelt thanks for your time and the depth of your review. We remain entirely open to your guidance. **If you still perceive our theoretical foundation as lacking in depth, we would be extremely grateful if you could explicitly point out which specific aspects or assumptions need strengthening.** We respectfully request **an opportunity to address these concerns**: we are fully prepared and committed to providing additional theoretical derivations or empirical testing to resolve them.
>
> Given that our work provides an effective solution to the detection problem and is **theoretically self-consistent** within the scope of statistics, we would be **extremely grateful** if you could consider **raising your evaluation**.
>
>
> **References**
>
> [1] Li et al. Transformers as algorithms: Generalization and stability in in-context learning, ICML 2023.
>
> [2] Li et al. How do transformers learn topic structure: Towards a mechanistic understanding, ICML 2023.
>
> [3] Tian et al. Scan and Snap: Understanding training dynamics and token composition in 1-layer Transformer, NeurIPS 2023.
>
> [4] Huang et al. Non-asymptotic convergence of training transformers for next-token prediction, NeurIPS 2024.
>
> [5] Zhang et al. Trained transformers learn linear models in-context, JMLR 2024.
>
> [6] Zhang et al. Understanding deep learning requires rethinking generalization, ICLR 2017.
>
> [7] Rudin. Stop explaining black box machine learning models for high stakes decisions and use interpretable models instead, Nature Machine Intelligence 2019.
>
> [8] Zhang et al. What If the Input is Expanded in OOD Detection?, NeurIPS 2024.
>
> [9] Zhang et al. ICR Probe: Tracking hidden state dynamics for reliable hallucination detection in LLMs, ACL 2025.
>
> [10] Liu et al. Lost in the Middle: How language models use long contexts, TACL 2024.
>
> [11] Zou et al. Representation engineering: A top-down approach to AI transparency, arXiv 2023.

---

> > ### Comment · Reviewer_7VBg · 2025-11-24
> >
> > I agree with your point:
> > > Furthermore, requiring a full mechanistic explanation for hallucination dynamics represents an extremely high bar, one that is currently considered an open "Grand Challenge" in the field.
> >
> > You stated:
> > > Our workflow aligns with the standard research methodology widely adopted in existing literature [6, 8, 9, 10, 11]. In this paradigm, researchers first identify robust phenomena from complex systems, then derive algorithmic designs based on these observations, and finally validate the method through rigorous statistical analysis. We treat the LLM as a complex stochastic distribution where hallucinations manifest as detectable distributional anomalies.
> >
> > I still feel there are some differences. For example, in paper *[8] Zhang et al., What If the Input is Expanded in OOD Detection?, NeurIPS 2024*, the authors apply several “common corruptions” to the input and evaluate the effects across multi-class tasks. In paper *[9] Zhang et al., ICR Probe: Tracking Hidden State Dynamics for Reliable Hallucination Detection in LLMs, ACL 2025*, the hallucination signal is defined based on the divergence between attention-update directions and FFN-update directions. Conceptually, these works reason more directly about model inputs and structural behavior, closer to the model’s internal dynamics, rather than inferring hallucination signals from downstream tasks. In essence, they are still somewhat different. I view Multi-Turn as a key downstream task, so the hallucination signal it reveals is not strictly grounded in first principles. (Of course, this is under the premise that I agree with your earlier “Grand Challenge” point.)
> >
> > Therefore, **I can acknowledge that this is an experimental paper, and the experiments are relatively thorough**. I also believe that adding multi-turn consistency experiments is necessary, and clearer interpretation is needed. You aim to address the Generalizable Hallucination Detection (GHD) problem, and **I still hope to see approaches with stronger theoretical grounding and practical generality**. I remain somewhat reserved about using multi-turn dialogue (as a downstream task) for hallucination detection, so this score is the highest I can give.
> >
> > I would indeed look forward to more theoretical decomposition. For example, is the task design of multi-turn dialogue itself something worth discussing? Or do multi-turn patterns share any underlying mechanisms with other forms of hallucination elicitation? These are just my own spontaneous thoughts rather than suggestions, but I do hope for deeper exploration on these aspects.

---

> > > ### Author Response · Authors · 2025-11-27
> > > **Response to Reviewer 7VBg’s Concerns (1/7)**
> > >
> > > We **sincerely thank you** for your remarkably prompt, thoughtful, and intellectually generous follow-up. In response to your concerns, we have added **new experiments** as well as **additional theoretical decomposition**. We hope that the clarifications and new evidence directly address the issues you raised; if our responses satisfactorily resolve your concerns, we would be genuinely grateful if you could **kindly reconsider your evaluation**. *We wish you all the best in your research and daily life, and we greatly appreciate the time and care you have already invested in reviewing our work.*
> > >
> > >
> > > > Concern 1: “I view Multi-Turn as a key downstream task, so the hallucination signal it reveals is not strictly grounded in first principles.”
> > >
> > > We appreciate this insightful distinction. You are absolutely right that a method relying solely on downstream behavior (such as checking if the model explicitly contradicts itself across turns) would **lack the theoretical grounding of first-principles approaches**.
> > >
> > > However, we wish to clarify the precise role of the multi-turn setting in our framework. We treat the dialogue not as the signal itself, but as the elicitation protocol used to expose the model's internal states. While the interaction occurs at **the downstream task level**, the measurement is strictly rooted in **the model's internal representations**. At every turn, rather than analyzing the text output for surface-level consistency, we apply internal probes (such as entropy-based regressors SEP or hidden-state classifiers like SAPLMA) to read the model’s activations directly.
> > >
> > > In this sense, our approach aligns with the internal-dynamics perspective found in works like Zhang et al. [1]. We operate on similar principles but extend the observation window: we use the multi-turn context to force **the model’s internal representations** to evolve, and then measure the curvature of that trajectory. As demonstrated in our response to **Concern 2**, replacing these internal detectors with purely downstream consistency metrics leads to a significant performance drop. This empirically confirms that SpikeScore derives its efficacy from reading the model's internal dynamics, not from observing surface-level task behavior.
> > >
> > > [1] Zhang et al. ICR Probe: Tracking hidden state dynamics for reliable hallucination detection in LLMs, ACL 2025.

---

> > > ### Author Response · Authors · 2025-11-27
> > > **Response to Reviewer 7VBg’s Concerns (2/7)**
> > >
> > > > Concern 2: “I also believe that adding multi-turn consistency experiments is necessary, and clearer interpretation is needed.”
> > >
> > >
> > > We strongly agree that adding explicit **multi-turn consistency experiments** is essential to clarify the boundary between our method and downstream consistency checks. Specifically, we must empirically demonstrate the distinction between (i) *using multi-turn interaction as an elicitation protocol for internal signals* (our approach) versus (ii) *treating multi-turn consistency itself as the detection signal*.
> > >
> > > To rigorously test this, we designed an ablation study. We maintained the **entire SpikeScore pipeline** (including the curvature computation and classifier) but replaced the input signal. Instead of feeding the pipeline with internal hallucination scores (derived from hidden states, logits), we fed it with **scalar "consistency scores"** calculated from the text output. This allows us to directly compare the predictive power of internal dynamics against downstream consistency.
> > >
> > > We consider two representative and complementary consistency measures:
> > >
> > > * **D-SMART logical consistency (CS) [1]**
> > >   Following the Consistency Score (CS) from D-SMART, for each turn we run an NLI model with the dialogue history as premise and the current response as hypothesis. Let $P_E(t)$ and $P_C(t)$ be the entailment and contradiction probabilities for turn $t$. The per-turn **logical consistency score** is:
> > >   $$
> > >   CS_t = \frac{(P_E(t) - P_C(t)) + 1}{2},
> > >   $$
> > >   which lies in $[0,1]$ (close to $1$ = strongly entailed by history, close to $0$ = strongly contradictory). We then treat ${CS_t}$ as the trajectory input to SpikeScore. This metric explicitly measures **multi-turn logical coherence**.
> > >
> > > * **Pragmatic Consistency (persona-style self-consistency) [2]**
> > >   Following Kim et al., we use a **pragmatic listener** $L_0$ that compares the token probability under a “persona-conditioned” context versus a neutral context to produce a posterior
> > >   $$
> > >   L_t^0(i) \propto S_t^0(u_t \mid i, h, u_{<t})^\beta \cdot p_t,
> > >   $$
> > >   which quantifies how strongly the current token (and, after aggregation, the current turn) remains consistent with a target persona. In practice, we aggregate token-level posteriors within each turn (using the last-token posterior) to obtain a **turn-level persona-consistency score**. This metric captures whether the model’s **stylistic and self-descriptive behavior** remains stable across turns.
> > >
> > > We deliberately chose these two signals because they probe **different aspects of multi-turn behavior**:
> > > D-SMART-CS focuses on **logical entailment vs contradiction**, while Pragmatic Consistency focuses on **persona-style stability**. Both are canonical examples of downstream multi-turn consistency measures, and both are conceptually distinct from hallucination labels.
> > >
> > > Using the same four models as in our earlier ablation (base vs consistency-trained variants):
> > >
> > > * Qwen-2.5-7B vs Qwen-2.5-7B-ConsistentChat
> > > * Qwen3-4B-Instruct vs EvolLLM-Linh

---

> > > ### Author Response · Authors · 2025-11-27
> > > **Response to Reviewer 7VBg’s Concerns (3/7)**
> > >
> > > we compare (i) our original SpikeScore (internal hallucination detectors) with (ii) SpikeScore fed by Consistency Score (CS) and (iii) SpikeScore fed by Pragmatic Consistency. The AUROC results are:
> > >
> > > | Scoring Method                   | Model                        | TriviaQA | CommonsenseQA | Belebele | CoQA     | Math     | SVAMP    | Avg      |
> > > |---------------------------------:|------------------------------|:--------:|:-------------:|:--------:|:--------:|:--------:|:--------:|:--------:|
> > > | **Ours (hallucination-based)**   | Qwen-2.5-7B                 | 0.7242   | 0.7269        | 0.6928   | 0.8257   | 0.7521   | 0.7265   | 0.7414   |
> > > | **Ours (hallucination-based)**   | Qwen-2.5-7B-ConsistentChat  | **0.7560** | 0.7463      | 0.6879   | 0.8148   | 0.7295   | **0.7419** | 0.7461 |
> > > | **Ours (hallucination-based)**   | Qwen3-4B-Instruct           | 0.7357   | 0.7367        | 0.6774   | **0.8505** | 0.7484 | 0.7335   | 0.7470   |
> > > | **Ours (hallucination-based)**   | EvolLLM-Linh                | 0.7430   | **0.7497**    | **0.7159** | 0.8213 | **0.7531** | 0.7085 | **0.7486** |
> > > | D-SMART CS (logical consistency) | Qwen-2.5-7B                 | 0.6520   | 0.7005        | 0.6291   | 0.7519   | 0.6946   | 0.6839   | 0.6853   |
> > > | D-SMART CS (logical consistency) | Qwen-2.5-7B-ConsistentChat  | 0.6640   | 0.6299        | 0.6239   | 0.7001   | 0.6354   | 0.6520   | 0.6509   |
> > > | D-SMART CS (logical consistency) | Qwen3-4B-Instruct           | 0.6765   | 0.6687        | 0.6343   | 0.8128   | 0.7048   | 0.6834   | 0.6968   |
> > > | D-SMART CS (logical consistency) | EvolLLM-Linh                | 0.6335   | 0.6572        | 0.6295   | 0.7191   | 0.6743   | 0.6203   | 0.6556   |
> > > | Pragmatic Consistency            | Qwen-2.5-7B                 | 0.6015   | 0.6168        | 0.5681   | 0.7075   | 0.6165   | 0.6367   | 0.6245   |
> > > | Pragmatic Consistency            | Qwen-2.5-7B-ConsistentChat  | 0.5825   | 0.5919        | 0.5074   | 0.6733   | 0.5718   | 0.5969   | 0.5873   |
> > > | Pragmatic Consistency            | Qwen3-4B-Instruct           | 0.6193   | 0.6453        | 0.5536   | 0.7500   | 0.6326   | 0.6411   | 0.6403   |
> > > | Pragmatic Consistency            | EvolLLM-Linh                | 0.5822   | 0.5545        | 0.5136   | 0.6643   | 0.5998   | 0.5835   | 0.5830   |
> > >
> > >
> > > *(Bold numbers indicate the best performance. Our method is bolded in the Method column to emphasize the comparison.)*
> > >
> > >
> > > **Findings and Interpretation:**
> > >
> > > 1. **Using consistency scores as the underlying signal leads to substantial degradation.**
> > >    Across all datasets and models, both multi-turn consistency metrics perform **significantly worse** than SpikeScore with hallucination-based scores—around **5–10 AUROC points lower on average**. The CS-based variant (logical consistency) is notably better than Pragmatic Consistency (persona-style stability), but still clearly inferior to our original method. This shows that simply “plugging in” multi-turn consistency as the per-step signal is not enough: downstream consistency, whether logical or persona-oriented, is much less predictive of hallucination than internal hallucination detectors.
> > > 2. **Consistency training destabilizes consistency-based signals much more than our hallucination-based signal.**
> > >    For D-SMART-CS and Pragmatic Consistency, the consistency-trained models (Qwen-2.5-7B-ConsistentChat, EvolLLM-Linh) often **do not improve and sometimes even degrade** the AUROC, especially on cross-domain datasets such as Belebele and SVAMP. In contrast, our hallucination-based SpikeScore remains **remarkably stable** under the same consistency training, with only minor fluctuations and even slight improvements in average AUROC. This suggests that multi-turn consistency training mainly reshapes surface behavior (e.g., smoother dialogues, reduced topic drift), which directly affects consistency metrics but **does not disrupt the internal hallucination signal that SpikeScore relies on**.
> > >
> > > **In summary, these ablations may address your concern:**
> > > when we intentionally replace our internal detectors with purely downstream multi-turn consistency scores, performance drops substantially and becomes sensitive to whether the model was explicitly trained for consistency. By contrast, our original SpikeScore (built on **internal hallucination scores**) remains robust across both base and consistency-enhanced models. This supports our claim that the multi-turn setting in SpikeScore is **not** used as a downstream task metric, but purely as a mechanism to expose and amplify an already well-grounded internal hallucination signal.
> > >
> > > [1] Lei et al. D-SMART: Enhancing LLM Dialogue Consistency via Dynamic Structured Memory and Reasoning Tree, arXiv 2025.
> > >
> > > [2] Kim et al. Will I Sound Like Me? Improving Persona Consistency in Dialogues through Pragmatic Self-Consciousness, EMNLP 2020.

---

> > > > ### Comment · Reviewer_7VBg · 2025-11-28
> > > >
> > > > I acknowledge this aspect. The supplementary ablations you added are appropriate and help clarify the issue. The distinction between internal hallucination signals and semantic-level consistency is technically substantive, and your analysis does make the differentiation between the two more explicit.

---

> > > ### Author Response · Authors · 2025-11-27
> > > **Response to Reviewer 7VBg’s Concerns (4/7)**
> > >
> > > > Concern 3: “I would indeed look forward to more theoretical decomposition.”
> > >
> > > We are very grateful for your interest in the theoretical side of this work. From our perspective, the core question we study is not “can we detect hallucination by looking at multi-turn behavior,” but rather a more structural one: **when a training-based hallucination score is pushed out of its training domain, does it truly collapse, or does it still carry a domain-invariant component that can be recovered if we look at it in the right way?** SpikeScore is our attempt to answer this question.
> > >
> > > Empirically, we find that when these internal scores (from SEP-style probes, SAPLMA-style predictors, and related detectors) are arranged as a trajectory and we analyze their curvature over turns, a robust and transferable pattern emerges, sharp spikes in the trajectory correlate with hallucinations even when the underlying detector alone generalizes poorly across domains. The multi-turn setting is thus a *lens* that makes this structure visible, but the signal itself still comes from internal representations rather than from dialogue-level behavior. We try to formalize this intuition through a spectral-bias view: curvature over time naturally emphasizes **low-frequency, domain-agnostic instability components**, whereas single-step probes tend to exploit **higher-frequency, semantically entangled cues** that are more sensitive to domain shift. Our small-sample training experiments in **Appendix D.1** were designed precisely to probe this decomposition, our method saturates quickly with very few training examples, while single-step probes require far more data and degrade sharply when moved out of domain.
> > >
> > > We fully agree that this is still far from a complete mechanistic theory of hallucination. However, we hope this gives a clearer sense in which our contribution goes beyond a purely empirical recipe: it proposes a concrete way to **decompose what training-based detectors have learned** into a transferable, low-frequency component that can be accessed via temporal dynamics, and a fragile, domain-specific component that lives at the single-step level. We see this as one step toward the kind of theoretical decomposition you are asking for.

---

> > > ### Author Response · Authors · 2025-11-27
> > > **Response to Reviewer 7VBg’s Concerns (5/7)**
> > >
> > > > Concern 4: “For example, is the task design of multi-turn dialogue itself something worth discussing?”
> > >
> > > We completely agree that the design of the multi-turn dialogue protocol is worth explicit discussion. In the revised manuscript we now describe this more systematically in **Appendix G** (prompt library and scheduling), **Appendix D.7** (prompt-selection robustness), and **Appendix D.8** (polite-alignment attack). Below we summarise the key settings and results.
> > >
> > > On the prompt-design side, our goal was to avoid both an overfitted, hand-crafted script and an unconstrained random schedule. We therefore adopt a hybrid strategy. Prompts are organised along two axes, type and strength, and follow a weak-to-strong progression across turns. Within each strength level, we randomly select concrete templates once at the beginning of our experiments. After this initial draw, the exact prompt sequence is fixed and reused for all models and datasets, so that comparisons remain fair. The full prompt inventory and scheduling rules are given in Appendix G.
> > >
> > > To assess how much SpikeScore depends on this particular prompt sequence, we ran a robustness study where we **only** changed the random seed that controls prompt selection (not the model initialisation or sampling seeds), while keeping all other settings identical. We trained on CoQA and evaluated on Math using Llama3.1-8B and Qwen3-8B, reporting AUROC as well as the most frequent turn position at which the maximum second-order difference (“PeakTurn”) occurs across test examples:
> > >
> > > **Table: Prompt-selection robustness on CoQA → MATH**
> > >
> > > ***Llama3.1-8B***
> > >
> > > | Seed     | 42     | 422    | 4222   | 36     | 366    | 3666   |
> > > | -------- | ------ | ------ | ------ | ------ | ------ | ------ |
> > > | AUROC    | 0.7316 | 0.7293 | 0.7358 | 0.7417 | 0.7329 | 0.7410 |
> > > | PeakTurn | 4      | 3      | 6      | 4      | 4      | 5      |
> > >
> > > ***Qwen3-8B***
> > >
> > > | Seed     | 42     | 422    | 4222   | 36     | 366    | 3666   |
> > > | -------- | ------ | ------ | ------ | ------ | ------ | ------ |
> > > | AUROC    | 0.7431 | 0.7504 | 0.7632 | 0.7511 | 0.7398 | 0.7455 |
> > > | PeakTurn | 7      | 6      | 3      | 4      | 5      | 5      |
> > >
> > > ***Across-seed AUROC statistics***
> > >
> > > | Model       | Mean AUROC | Var(AUROC)* |
> > > | ----------- | ---------- | ----------- |
> > > | Llama3.1-8B | 0.7354     | 2.58e-05    |
> > > | Qwen3-8B    | 0.7489     | 6.79e-05    |
> > >
> > > The results show that different prompt selections influence which turn most frequently exhibits the sharpest spike (PeakTurn varies across seeds), indicating that prompt choice affects the conversational trajectory but not the overall discriminative power of the metric. However, the AUROC values remain extremely stable across seeds, with variance on the order of 10⁻⁵. Since SpikeScore only uses the **magnitude** of the maximum second-order difference, not its position, it is largely insensitive to these shifts in spike location. This suggests that the metric does not hinge on a fragile, hand-picked multi-turn script.

---

> > > ### Author Response · Authors · 2025-11-27
> > > **Response to Reviewer 7VBg’s Concerns (7/7)**
> > >
> > > > Concern 5: “Do multi-turn patterns share any underlying mechanisms with other forms of hallucination elicitation?”
> > >
> > >
> > > This is an excellent point! We absolutely view multi-turn interaction not as **a standalone “trick,”** but as one specific instance of **a broader family of stress-tests** designed to surface latent errors. Conceptually, our goal isn't to induce hallucinations for the sake of it, but to use the dialogue to repeatedly probe the model’s stability around a specific belief.
> > >
> > > There is definitely a shared mechanism here, **instability under pressure**. In multi-agent debate, Liang et al. [1] show that having models challenge each other leads to disagreements across debate rounds that can be used to expose incorrect or unstable reasoning. In long-form factuality studies, Wei et al. [2] observe that when a model is asked to produce extended answers on open-domain topics, factual errors tend to accumulate over the course of the generation, and methods such as SAFE explicitly re-query evidence and re-evaluate segments of the same answer to reveal these failures. Self-consistency decoding follows a very similar principle, Wang et al. [3] sample multiple chains of thought for the same question and treat disagreement across sampled reasoning paths as a cue that the model’s belief is unstable, selecting the majority answer as a more reliable estimate. Across these settings, the elicitation method functions by repeatedly probing the model around the same underlying belief and observing how its responses evolve under this sustained pressure.
> > >
> > > Our use of multi-turn dialogue fits naturally into this picture: (1) inconsistency in conversations have been repeatedly documented as a **common failure mode** in real-world logs and multi-turn benchmarks, and (2) it is one of the most **natural ways** to obtain a sequence of internal states around the same question, without modifying decoding algorithms.
> > >
> > > Your comment points toward a compelling vision, **a unified framework where dialogue, debate, and sampling are treated as interchangeable modules targeting the same latent instability**. While formalizing such a general theory is beyond the scope of this single paper, your feedback has helped us clarify exactly how our work fits into that broader landscape. It is a perspective we definitely plan to carry forward in future work.
> > >
> > >
> > > [1] Liang, H., Yin, W., Wang, X., Zhang, T., and Zhang, P. 2024. Encouraging Divergent Thinking in Large Language Models through Multi-Agent Debate. In Proceedings of EMNLP 2024.
> > > [2] Wei, J., Yang, C., Song, X., Lu, Y., Hu, N., Huang, J., Tran, D., Peng, D., Liu, R., Huang, D., Du, C., and Le, Q. V. 2024. Long-Form Factuality in Large Language Models. arXiv preprint arXiv:2403.18802.
> > > [3] Wang, X., Wei, J., Schuurmans, D., Le, Q. V., Chi, E. H., Narang, S., Chowdhery, A., and Zhou, D. 2023. Self-Consistency Improves Chain-of-Thought Reasoning in Language Models. In Proceedings of ICLR 2023.
> > >
> > >
> > > Thank you once again for the depth and sincerity of your engagement with our submission. We have carefully addressed each of your concerns with new experiments, clearer methodological explanations, and additional theoretical analysis. Given the **completeness of these** and **our substantial effort** to respond to every point you raised, **we would be truly appreciative if you could reconsider your rating**. Your feedback has meaningfully strengthened the paper, and we remain grateful for your time, insight, and dedication.

---

> > > > ### Comment · Reviewer_7VBg · 2025-11-28
> > > >
> > > > > There is definitely a shared mechanism here, instability under pressure. In multi-agent debate, Liang et al. [1] show that having models challenge each other leads to disagreements across debate rounds that can be used to expose incorrect or unstable reasoning.
> > > >
> > > > I agree with your observation — the broader community is indeed actively exploring multi-agent, multi-turn, and other interaction-based paradigms, largely because they appear to tap into mechanisms that are closely related to underlying model intelligence.
> > > >
> > > > The results in Section 6/7 is interesting. Multi-turn behavior clearly warrants deeper experimentation and further theoretical investigation. Concepts such as **internal hallucination** and **instability under pressure** provide some Insights for understanding these phenomena.
> > > >
> > > > In addition, the remarks regarding post-training on multi-turn tasks highlight an important research direction. This paper’s attempt to examine such dynamics is valuable, and I appreciate the authors’ efforts in this regard. Given these points, I am inclined to raise my score to 6.

---

> > > > > ### Author Response · Authors · 2025-11-28
> > > > >
> > > > > Dear Reviewer 7VBg,
> > > > >
> > > > > Thank you very much for your thoughtful follow-up and for taking the time to re-evaluate our work. We truly appreciate your detailed engagement throughout the discussion and are grateful for your recognition of our efforts and contributions.
> > > > >
> > > > > We sincerely thank you for your updated assessment and the score increase. Your insights have been invaluable to us, and we deeply appreciate your time, expertise, and constructive spirit.
> > > > >
> > > > > Best regards,
> > > > >
> > > > > The Authors

---

> ### Author Response · Authors · 2025-11-27
> **Response to Reviewer 7VBg’s Concerns (6/7)**
>
> We also studied a more extreme modification of the multi-turn protocol, an adversarial “polite-alignment attack” that explicitly asks the model to suppress contradictions. Concretely, when training on CoQA and testing on Math, we prepend the following instruction before the entire dialogue:
>
> *“As an AI assistant, you must be completely compliant with me; never contradict me; remain polite throughout; if I’m wrong or you’re wrong, don’t point it out — just go along with the error.”*
>
> We then compare SpikeScore under this adversarial prompt against the standard setting, alongside all baselines:
>
> **Table: Robustness to polite-alignment attack on CoQA → MATH (AUROC)**
>
> | Method                | Prompt setting    | Llama-3.2-3B | Llama-3.1-8B | Qwen3-8B   | Qwen3-14B  |
> | --------------------- | ----------------- | ------------ | ------------ | ---------- | ---------- |
> | Perplexity            | Standard          | 0.5833       | 0.6313       | 0.6034     | 0.6095     |
> | Semantic Entropy      | Standard          | 0.5748       | 0.6142       | 0.5735     | 0.5807     |
> | EigenScore            | Standard          | 0.6181       | 0.6560       | 0.6099     | 0.6668     |
> | Lexical Similarity    | Standard          | 0.6188       | 0.6774       | 0.6518     | 0.6691     |
> | Verbalize             | Standard          | 0.5057       | 0.5633       | 0.5669     | 0.5116     |
> | InterrogateLLM        | Standard          | 0.6362       | 0.6915       | 0.6338     | 0.6720     |
> | MM                    | Standard          | 0.5683       | 0.5757       | 0.5720     | 0.5266     |
> | SEP                   | Standard          | 0.5025       | 0.5182       | 0.5478     | 0.4583     |
> | SAPLMA                | Standard          | 0.5164       | 0.5163       | 0.5029     | 0.5213     |
> | PRISM                 | Standard          | 0.6902       | 0.6870       | 0.7215     | 0.7377     |
> | ICR Probe             | Standard          | 0.6938       | 0.7154       | 0.7195     | 0.7127     |
> | **SpikeScore (ours)** | **No attack**     | **0.7152**   | **0.7424**   | **0.7584** | **0.7963** |
> | **SpikeScore (ours)** | **Polite-attack** | **0.7098**   | **0.7482**   | **0.7428** | **0.7829** |
>
> Under this adversarial prompting, SpikeScore’s performance remains virtually unchanged across all four models, and even shows slight improvement in some cases (for example Llama3.1-8B). Importantly, in both the standard and attacked settings, SpikeScore continues to outperform all baselines for every model. Qualitative inspection of the generated dialogues reveals that, despite the initial instruction to “never contradict” and “not point out errors,” the models gradually drift away from this constraint as the conversation progresses. By turns four to five, many models begin revising earlier statements when pressed, and the internal instability captured by SpikeScore persists even when surface-level contradictions are partially suppressed.
>
> Thank you again for highlighting the importance of multi-turn task design.

---

> ### Author Response · Authors · 2025-11-27
>
> Dear Reviewer 7VBg,
>
> Could you please let us know whether we have fully addressed your concerns?
>
> If so, **we would be very grateful if you could consider increasing the score**. If not, we kindly ask for any further, more detailed suggestions so that we can make additional improvements.
>
> Best Regards,
>
> Authors

---

### Official Review · Reviewer_ALVJ · 2025-10-31

**Soundness:** 3
**Presentation:** 3
**Contribution:** 3
**Rating:** 6
**Confidence:** 3

**Summary:**

This paper introduces SpikeScore, a simple yet effective metric to detect hallucinations in LLMs under cross-domain settings. The key observation is that hallucinated answers, when probed in multi-turn self-dialog environments, exhibit sharp uncertainty spikes, whereas truthful responses show smoother trajectories. The authors formalize this phenomenon, propose SpikeScore as the maximum second-order difference in a scoring trajectory, provide theoretical justification through probabilistic separability bounds, and perform extensive cross-domain evaluations across six datasets and four open-source LLMs. Results show consistent improvements over both training-free and training-based baselines, and over cross-domain oriented methods like PRISM and ICR Probe.

**Strengths:**

1. Tackles a realistic and impactful problem setting (generalizable hallucination detection).
2. Novel insight that hallucination triggers instability in multi-turn self-dialogue.
3. Simple but powerful metric (SpikeScore) with no need for additional finetuning.
4. Strong cross-domain results vs powerful baselines (PRISM, ICR probe).

**Weaknesses:**

1. Theory relies on assumptions that may not always hold in real LLM behavior.
2. No evaluation on more complex generative formats (e.g., long-form reasoning beyond QA).
3. Domain choice mainly QA/knowledge tasks; extension to code or vision-language tasks not discussed.

**Questions:**

1. How robust is SpikeScore under adversarial or “polite-aligned” prompting that suppresses contradictions?
2. Would combining SpikeScore with chain-of-thought introspection further improve results?
3. Can the approach be extended to detect hallucination in long-form generation or agentic workflows?
4. Any evidence whether multilingual or multimodal models show similar spike patterns?

---

> ### Author Response · Authors · 2025-11-20
> **Response to Reviewer ALVJ’s Comments and Concerns (1/6)**
>
> Dear Reviewer,
>
> We sincerely appreciate your recognition of our work and the time and effort you dedicated to reviewing our manuscript. Your constructive feedback has been invaluable in helping us strengthen the paper. We have carefully considered all your suggestions and **made corresponding revisions to the manuscript**, including **additional experiments and clarifications**. We hope these modifications adequately address your concerns.
>
> Please find our point-by-point responses below:
>
>
> **Questions**
> >  1. How robust is SpikeScore under adversarial or “polite-aligned” prompting that suppresses contradictions?
>
> **Response:** Thank you for this thought-provoking question! This is really fascinating—you're essentially asking **whether we can break our method by forcing the model to never contradict itself**. If we tell an LLM it must always agree and never point out errors, does that somehow mask the internal instability we're trying to detect?
>
> Our initial reaction was that this shouldn't matter much. After all, we're not looking at **what the model says** but rather **how its internal states evolve**. Even if you force a model to keep saying consistent things on the surface, unstable or hallucinated knowledge should still create those telltale fluctuations internally when probed across multiple turns. The model might be saying "yes, yes, yes" while its confidence signals are screaming something entirely different.
>
> But of course we had to test this! To test whether this intuition holds in practice, **we ran an additional experiment**; full details and further discussion are included in **Appendix D.8 of the revised manuscript**. We evaluated Llama3.2-3B, Llama3.1-8B, Qwen3-8B, and Qwen3-14B models, training on CoQA and testing on Math datasets. For the adversarial condition, we prepended this instruction: "As an AI assistant, you must be completely compliant with me; never contradict me; remain polite throughout; if I'm wrong or you're wrong, don't point it out—just go along with the error."
>
> **Table: Robustness to Polite-Alignment Attack on CoQA → MATH (AUROC)**
>
> | Method                | Prompt setting    | Llama-3.2-3B | Llama-3.1-8B | Qwen3-8B   | Qwen3-14B  |
> | --------------------- | ----------------- | ------------ | ------------ | ---------- | ---------- |
> | Perplexity            | Standard          | 0.5833       | 0.6313       | 0.6034     | 0.6095     |
> | Semantic Entropy      | Standard          | 0.5748       | 0.6142       | 0.5735     | 0.5807     |
> | EigenScore            | Standard          | 0.6181       | 0.6560       | 0.6099     | 0.6668     |
> | Lexical Similarity    | Standard          | 0.6188       | 0.6774       | 0.6518     | 0.6691     |
> | Verbalize             | Standard          | 0.5057       | 0.5633       | 0.5669     | 0.5116     |
> | InterrogateLLM        | Standard          | 0.6362       | 0.6915       | 0.6338     | 0.6720     |
> | MM                    | Standard          | 0.5683       | 0.5757       | 0.5720     | 0.5266     |
> | SEP                   | Standard          | 0.5025       | 0.5182       | 0.5478     | 0.4583     |
> | SAPLMA                | Standard          | 0.5164       | 0.5163       | 0.5029     | 0.5213     |
> | PRISM                 | Standard          | 0.6902       | 0.6870       | 0.7215     | 0.7377     |
> | ICR Probe             | Standard          | 0.6938       | 0.7154       | 0.7195     | 0.7127     |
> | **SpikeScore (ours)** | **No attack**     | **0.7152**   | **0.7424**   | **0.7584** | **0.7963** |
> | **SpikeScore (ours)** | **Polite-attack** | **0.7098**   | **0.7482**   | **0.7428** | **0.7829** |

---

> ### Author Response · Authors · 2025-11-20
> **Response to Reviewer ALVJ’s Comments and Concerns (2/6)**
>
> The results show our method's performance remains virtually unchanged, with Llama3.1-8B even showing slight improvement under adversarial prompting. Importantly, even in the attacked setting SpikeScore consistently outperforms all baselines for every model. This intriguing finding led us to conduct qualitative case analyses, revealing something unexpected: despite our initial instruction to "never contradict" and "don't point out errors", **models increasingly ignored these constraints as conversations progressed**. By **turns 4-5**, most models simply stopped adhering to the suppression directive and began contradicting their earlier statements when pressed.
>
> This suggests that subsequent prompts can override initial constraints, essentially **"washing out" the adversarial instruction** through dialogue momentum. It's a compelling observation about how LLMs handle conflicting instructions—**when faced with incompatible directives, models appear to prioritize more recent or task-specific prompts over blanket behavioral constraints**. This opens interesting questions for future work on instruction adherence and robustness. How can we better understand and control which instructions models choose to follow when directives conflict? Such work could enhance model resistance to instruction-based attacks.
>
> Thank you again for this insightful question. While adversarial prompting isn't central to our main contributions, exploring this angle revealed fascinating patterns in instruction following that have genuinely inspired directions for future research on model controllability and robustness.

---

> ### Author Response · Authors · 2025-11-20
> **Response to Reviewer ALVJ’s Comments and Concerns (3/6)**
>
> >  2. Would combining SpikeScore with chain-of-thought introspection further improve results?
>
> **Response:** Thank you for this excellent question—you really hit on something we grappled with early in this work! **We actually did explore combining SpikeScore with chain-of-thought introspection initially.** The idea seemed natural: if CoT generates intermediate reasoning steps, we could track hallucination scores at each step and potentially get a richer signal about where things go wrong.
>
> But here's the challenge we ran into: **LLMs produce wildly different numbers of reasoning steps depending on problem difficulty and type**. A simple arithmetic problem might generate 2-3 steps while a complex multi-hop reasoning task could spawn 10+ steps. This variability makes it nearly impossible to establish a unified scale for comparing hallucinated versus non-hallucinated responses. How do you fairly compare a 3-step correct answer against a 12-step hallucinated one?
>
> One tempting solution would be to **force the model to generate a fixed number of reasoning steps** for each problem. But this creates a bigger problem—we'd be artificially constraining the model's natural reasoning process. The "forced" structure would fundamentally alter how the model generates its outputs, introducing an experimental confound. **We'd essentially be dictating how the model should think, then testing whether that dictated thinking pattern contains hallucinations.** That's not really a fair or clean experimental setup—it's a bit like making someone walk in a specific pattern and then measuring their natural gait.

---

> ### Author Response · Authors · 2025-11-20
> **Response to Reviewer ALVJ’s Comments and Concerns (4/6)**
>
> >  3. Can the approach be extended to detect hallucination in long-form generation or agentic workflows?
>
> **Response:** Yes, our method can indeed be extended to long-form generation and agentic workflows. SpikeScore is fundamentally post-hoc and works with any generation process. To demonstrate this, **we specifically tested our approach on retrieval-augmented generation (RAG)**, where models autonomously query external knowledge bases, retrieve relevant information, and synthesize it into responses. RAG represents a quintessential agentic workflow since LLMs must operate as agents that interact with external tools and systems to complete complex tasks.
>
> We conducted comprehensive experiments using TriviaQA and RAGTruth benchmarks (full experimental details in **Section 4.3 and Appendix D.6 of the revised manuscript**). Specifically, we built a retrieval-augmented pipeline where all reference passages are treated as independent chunks in a shared corpus, questions serve as queries, both are encoded with bge-large-en embeddings, and we build a Faiss index to retrieve the top-4 passages per query. The concatenated "question + retrieved contexts" is then fed to Llama3.2-3B, Llama3.1-8B, Qwen3-8B, and Qwen3-14B models to produce the initial answer. After this RAG-augmented generation, we apply exactly the same multi-turn continuation protocol and SpikeScore computation as in our main experiments. Critically, all training-based methods (including SpikeScore) were trained only on non-RAG CoQA data, making this a strict test of cross-setting generalization to agentic scenarios.
>
> | Method             | Llama3.2-3B TriviaQA | Llama3.2-3B RAGTruth | Llama3.1-8B TriviaQA | Llama3.1-8B RAGTruth | Qwen3-8B TriviaQA | Qwen3-8B RAGTruth | Qwen3-14B TriviaQA | Qwen3-14B RAGTruth |
> | ------------------ | -------------------- | -------------------- | -------------------- | -------------------- | ----------------- | ----------------- | ------------------ | ------------------ |
> | Perplexity         | 0.5960               | 0.5936               | 0.6220               | 0.6151               | 0.6371            | 0.6284            | 0.6474             | 0.6420             |
> | Semantic Entropy   | 0.6330               | 0.6210               | 0.6173               | 0.6037               | 0.6492            | 0.6388            | 0.6745             | 0.6796             |
> | EigenScore         | 0.6426               | 0.6329               | 0.7085               | 0.6974               | 0.6571            | 0.6292            | 0.7064             | 0.6904             |
> | Lexical Similarity | 0.6248               | 0.6179               | 0.7102               | 0.6958               | 0.6375            | 0.6440            | 0.6542             | 0.6479             |
> | MM                 | 0.5724               | 0.5587               | 0.6020               | 0.5839               | 0.6166            | 0.6044            | 0.6291             | 0.6191             |
> | SEP                | 0.5312               | 0.4988               | 0.5779               | 0.5547               | 0.5161            | 0.5253            | 0.5368             | 0.5239             |
> | SAPLMA             | 0.6041               | 0.5874               | 0.5970               | 0.6011               | 0.6136            | 0.5904            | 0.6399             | 0.6338             |
> | ICR Probe          | 0.7532               | 0.7416               | 0.7682               | 0.7405               | 0.7495            | 0.7426            | 0.7841             | 0.7653             |
> | **SpikeScore(ours)**         | **0.7737**               | **0.7631**               | **0.7947**               | **0.8154**               | **0.8413**            | **0.8291**            | **0.8697**             | **0.8535**             |
>
> Across all four models and both RAG datasets, SpikeScore consistently outperforms the strongest cross-domain baseline (ICR Probe) and all other methods. This demonstrates robust transferability to agentic workflows without any modification to the underlying pipeline or additional training, confirming that our method naturally extends to complex generation scenarios where models operate as agents within structured workflows.

---

> ### Author Response · Authors · 2025-11-20
> **Response to Reviewer ALVJ’s Comments and Concerns (5/6)**
>
> >  4. Any evidence whether multilingual or multimodal models show similar spike patterns?
>
> We appreciate the reviewer’s question regarding multilingual and multimodal settings. For multilingual models, our indicators behave consistently because the underlying modality remains textual. In our experiments on the **Belebele benchmark**, which covers many languages including **German**, we observed spike patterns that closely match those seen in English, as reported in **Section 4**. This supports the view that our internal signals generalize across languages within the text domain [1].
>
> The multimodal case is fundamentally different. In multimodal large vision language models, hallucination is usually defined as a mismatch between the visual content in the image and the generated text, for example mentioning objects or attributes that are not present [2]. In contrast, textual hallucination in language models refers to fluent but factually incorrect or unsupported statements in text only outputs, independent of any image [3,4].
>
> Our work is built around **multi turn dialogue within a purely textual setting**. Unlike text-only dialogue, where multi-turn interactions naturally extend through linguistic context, **multimodal settings do not offer an equivalent mechanism for maintaining coherent turns**. Current benchmarks are built around single image–question pairs or very short VQA exchanges, and **moving to long-horizon trajectories would require redesigning the prompting strategy itself**. One must decide how visual inputs should evolve across turns, whether the model should repeatedly reinterpret the same image, whether it should be asked to revise or reflect on earlier visual descriptions, and how audio or video context could persist across multiple steps. Each of these choices fundamentally changes the task definition and would require substantial new experimentation, meaning there is no ready-made analogue of multi-turn textual dialogue in existing multimodal evaluations [2].
>
> Within this multimodal paradigm, several influential works study hallucination, but they do not provide the kind of hallucination score that our indicators require. POPE evaluates object hallucination by polling whether specific objects mentioned in the response exist in the image, and reports object level accuracy and error statistics rather than a continuous confidence value [4]. H POPE extends this idea to a hierarchical setting for objects and attributes, again producing structured judgments about presence or absence rather than a scalar hallucination score [5]. M HalDetect introduces a fine grained detection dataset with span level labels for hallucinated fragments in image question answer triples, which enables training of classifiers but yields categorical labels tied to those models [6]. Visual Contrastive Decoding and related methods compare outputs under different visual conditions to reduce object hallucination, but their metrics describe improvement in faithfulness rather than providing a reusable uncertainty signal [7]. Recent work that uses large models as detectors for multimodal hallucination similarly tags hallucinated spans or segments without exposing a model independent scalar hallucination measure [8].

---

> ### Author Response · Authors · 2025-11-20
> **Response to Reviewer ALVJ’s Comments and Concerns (6/6)**
>
> Our approach instead relies on indicators extracted from the internal dynamics of text only models that are trained or calibrated using textual hallucination scores. Semantic entropy and its extensions estimate uncertainty over sets of semantically related generations for a fixed text input and have been shown to correlate strongly with hallucination in language models [3,9,10]. Probes such as SEP are then trained to approximate these semantic entropy based scores directly from hidden states in a text model [10]. This entire pipeline assumes that semantic alternatives, clustering, and entropy computation are defined purely in the space of textual outputs. When the correctness of an answer depends jointly on the image and the text, **there is no agreed notion of semantic alternatives or semantic entropy for image text pairs**, and existing multimodal benchmarks do not supply the continuous hallucination scores needed to train or validate such probes. **So, extending our method to multimodal models amounts to a full multimodal uncertainty framework rather than a direct extension of our current method, and it lies outside the scope of the present study.**
>
> We sincerely appreciate the reviewer for raising this point. Although our current study cannot yet address the multimodal extension, we fully agree that this is an important and intellectually rewarding direction. The question of how spike-based internal signals might manifest in image–text models, and how one should define semantic alternatives and uncertainty in that joint space, is both challenging and exciting. We regard this as a valuable suggestion that opens a clear path for future work, and we intend to explore these questions in subsequent research.
>
> References
>
> [1] Bandarkar et al. The Belebele Benchmark: A Parallel Reading Comprehension Dataset in 122 Language Variants, ACL 2024.
>
> [2] Liu et al. A Survey on Hallucination in Large Vision Language Models, 2024.
>
> [3] Farquhar et al. Detecting Hallucinations in Large Language Models Using Semantic Entropy, Nature 2024.
>
> [4] Li et al. Evaluating Object Hallucination in Large Vision-Language Models, EMNLP 2023.
>
> [5] Pham and Schott. H-POPE: Hierarchical Polling-based Probing Evaluation of Hallucinations in Large Vision-Language Models, 2024.
>
> [6] Gunjal et al. Detecting and Preventing Hallucinations in Large Vision-Language Models, AAAI 2024.
>
> [7] Leng et al. Mitigating Object Hallucinations in Large Vision-Language Models through Visual Contrastive Decoding, CVPR 2024.
>
> [8] Cai et al. MHALO: Evaluating MLLMs as Fine-grained Hallucination Detectors, ACL 2025.
>
> [9] Kuhn et al. Semantic Uncertainty: Linguistic Invariances for Uncertainty Estimation in Natural Language Generation, 2023.
>
> [10] Kossen et al. Semantic Entropy Probes: Robust and Cheap Hallucination Detection in LLMs, 2024.

---

### Official Review · Reviewer_4ido · 2025-11-01

**Soundness:** 3
**Presentation:** 3
**Contribution:** 3
**Rating:** 6
**Confidence:** 3

**Summary:**

The paper proposes a method for hallucination detection based on SpikeScore. The score is based on the observation that when LLM hallucinates, contradictions are more often observed in a dialog. SpikeScore is designed to capture the local fluctuation of probability to generate a few consecutive answers in the dialog.

Experiments are done on several public datasets, comparing the SpikeScore method with several training-free, training-based and cross-domain methods. It is shown that SpikeScore outperforms all of them.

**Strengths:**

1. The main idea is based on the observation that in case of hallucination, LLM will produce inconsistent answers in a dialog. The proposed idea is interesting and new. It extends the Consistency-based methods by generating multiple answers in a dialog sequence. This extension is sound.

2. The paper provides a strong motivation from concrete observations.

3. The paper contains theoretical analyses on the method.

4. The experiments show clearly the advantage of SpikeScore.

**Weaknesses:**

1. The experimental results may be presented more clearly. Tables 1 and 2 that describe the main results are quite confusing. The two parts of each table are not explained. It is also unclear how leave-one-out is done for training-based methods. It is said that "training-based methods train on each dataset (columns) while all methods are evaluated on the remaining five datasets". So what is the Mean AURA in SEP under TriviaQA/Llama 3.2-3B? Is this the mean AURA tested on 5 other datasets using the model trained on TriviaQA? In that case, the number is not directly comparable to other training-free methods in the same column, which should have been tested on TriviaQA.

2. To generate dialogs, a set of prompts are designed and organized in several types. It is unclear how they are selected to generate a dialog. Are they randomly chosen? Do they follow some fixed patterns? One would believe that the selection of prompts may influence the SpikeScore. Have you observed this phenomenon? How do you deal with such a variability due to the selection of prompts?

**Questions:**

See the questions in Weakness.

---

> ### Author Response · Authors · 2025-11-20
> **Response to Reviewer 4ido’s Comments and Concerns (1/3)**
>
> Dear Reviewer,
>
> Thank you for your positive assessment of our work and for your thoughtful comments. We greatly value the insights you've provided and have taken them seriously in revising our manuscript. Based on your suggestions, we've **expanded our experimental validation** and **refined several aspects of our presentation**. We believe these updates address the points you raised.
>
> Detailed responses to your specific comments follow:
>
>
> **Weaknesses**
> >  1. The experimental results may be presented more clearly. Tables 1 and 2 that describe the main results are quite confusing. The two parts of each table are not explained. It is also unclear how leave-one-out is done for training-based methods. It is said that "training-based methods train on each dataset (columns) while all methods are evaluated on the remaining five datasets". So what is the Mean AURA in SEP under TriviaQA/Llama 3.2-3B? Is this the mean AURA tested on 5 other datasets using the model trained on TriviaQA? In that case, the number is not directly comparable to other training-free methods in the same column, which should have been tested on TriviaQA.
>
> **Response:** Thank you for pointing out the lack of clarity in the presentation of Tables 1 and 2. We fully agree that the original description of the leave-one-out protocol and the structure of the tables was insufficient, and we have revised the manuscript to make these aspects explicit.
>
> Regarding the comment that the two-part structure of each table was not explained, we clarify that the separation simply reflects space constraints and groups results by model family. This explanation has now been added directly to the **captions of Table 1 and Table 2** in the revised manuscript.
>
> The Mean AUROC reported for SEP under “TriviaQA / Llama 3.2–3B” indeed corresponds to the model trained on TriviaQA using Llama 3.2–3B and evaluated on the other five datasets only, your understanding is correct! This strict setup matches our problem formulation of ***generalizable hallucination detection***: detectors are required to generalize from one source domain to unseen target domains. By training on a single dataset and always testing on the remaining datasets, we measure cross-domain generalization rather than in-domain fitting.
>
> Although training-free methods do not involve any training stage and can be applied to a dataset directly, reporting them under a different evaluation setup from training-based methods would make the numbers in each column reflect different data conditions. In other words, training-based methods would be evaluated in **a “held-out target–dataset setting”**, while training-free methods would be evaluated on **the dataset shown in the column itself**, effectively mixing two incomparable evaluation regimes. To avoid this mismatch and keep all methods under a single, fair experimental structure, we report training-free methods using the same leave-one-out setup in the main tables.

---

> ### Author Response · Authors · 2025-11-20
> **Response to Reviewer 4ido’s Comments and Concerns (2/3)**
>
> We fully agree that this protocol makes the performance of training-free methods less intuitive, since the numbers in each column do not reflect direct in-domain performance. To address this concern, we added a new subsection in the appendix **(Appendix D.5)** that explicitly explains the issue and provides the direct, non–leave-one-out AUROC values for all training-free baselines. The full results are included in **Table 4**, reproduced below in simplified markdown form:
>
> ### **Direct (non–leave-one-out) AUROC of Training-Free Baselines**
> | Method           | Model        | TriviaQA | CommonsenseQA | Belebele | CoQA   | Math  | SVAMP | Average |
> |------------------|-------------|----------|---------------|----------|--------|-------|-------|---------|
> | Perplexity       | Llama-3.2-3B | 0.6336  | 0.6248        | 0.5739   | 0.5908 | 0.5833 | 0.5652 | 0.5953  |
> | Perplexity       | Llama-3.1-8B | 0.6784  | 0.6654        | 0.6325   | 0.6451 | 0.6313 | 0.6024 | 0.6425  |
> | Semantic Entropy | Llama-3.2-3B | 0.6669  | 0.6030        | 0.5683   | 0.6615 | 0.5748 | 0.6537 | 0.6214  |
> | Semantic Entropy | Llama-3.1-8B | 0.7125  | 0.6578        | 0.6358   | 0.7125 | 0.6142 | 0.6874 | 0.6700  |
> | EigenScore       | Llama-3.2-3B | 0.6604  | 0.6871        | 0.6185   | 0.6673 | 0.6181 | 0.6244 | 0.6460  |
> | EigenScore       | Llama-3.1-8B | 0.7051  | 0.7302        | 0.6587   | 0.7397 | 0.6560 | 0.6879 | 0.6963  |
> | Lexical Sim.     | Llama-3.2-3B | 0.6358  | 0.6439        | 0.6327   | 0.6581 | 0.6188 | 0.5958 | 0.6309  |
> | Lexical Sim.     | Llama-3.1-8B | 0.6698  | 0.6947        | 0.7095   | 0.7554 | 0.6774 | 0.6435 | 0.6917  |
> | Verbalize        | Llama-3.2-3B | 0.4655  | 0.4930        | 0.4496   | 0.5519 | 0.5057 | 0.5225 | 0.4980  |
> | Verbalize        | Llama-3.1-8B | 0.5210  | 0.5432        | 0.5013   | 0.5812 | 0.5633 | 0.5712 | 0.5469  |
> | InterrogateLLM   | Llama-3.2-3B | 0.6476  | 0.6378        | 0.6768   | 0.6733 | 0.6362 | 0.6297 | 0.6502  |
> | InterrogateLLM   | Llama-3.1-8B | 0.7158  | 0.6970        | 0.7154   | 0.7054 | 0.6915 | 0.6869 | 0.7020  |
> | Perplexity       | Qwen-8B      | 0.6847  | 0.6125        | 0.5898   | 0.6214 | 0.6034 | 0.5545 | 0.6110  |
> | Perplexity       | Qwen-14B     | 0.6699  | 0.6373        | 0.6256   | 0.6387 | 0.6095 | 0.5999 | 0.6301  |
> | Semantic Entropy | Qwen-8B      | 0.6666  | 0.6217        | 0.6300   | 0.6632 | 0.5735 | 0.6591 | 0.6357  |
> | Semantic Entropy | Qwen-14B     | 0.7239  | 0.6414        | 0.6314   | 0.7071 | 0.5807 | 0.6887 | 0.6622  |
> | EigenScore       | Qwen-8B      | 0.6611  | 0.7067        | 0.6229   | 0.7174 | 0.6099 | 0.6686 | 0.6644  |
> | EigenScore       | Qwen-14B     | 0.6932  | 0.7324        | 0.6367   | 0.7600 | 0.6668 | 0.6752 | 0.6940  |
> | Lexical Sim.     | Qwen-8B      | 0.6173  | 0.6619        | 0.6512   | 0.6641 | 0.6518 | 0.6164 | 0.6438  |
> | Lexical Sim.     | Qwen-14B     | 0.6406  | 0.6690        | 0.6657   | 0.6964 | 0.6691 | 0.6410 | 0.6636  |
> | Verbalize        | Qwen-8B      | 0.4754  | 0.4957        | 0.4977   | 0.5259 | 0.5669 | 0.5502 | 0.5186  |
> | Verbalize        | Qwen-14B     | 0.4645  | 0.5523        | 0.4931   | 0.5968 | 0.5116 | 0.5471 | 0.5276  |
> | InterrogateLLM   | Qwen-8B      | 0.6883  | 0.6843        | 0.7139   | 0.6386 | 0.6338 | 0.6604 | 0.6699  |
> | InterrogateLLM   | Qwen-14B     | 0.6755  | 0.6909        | 0.7094   | 0.7044 | 0.6720 | 0.6458 | 0.6830  |
>
>
> We hope these revisions improve the clarity and interpretability of our experimental results, and we thank the reviewer again for highlighting this issue.

---

> ### Author Response · Authors · 2025-11-20
> **Response to Reviewer 4ido’s Comments and Concerns (3/3)**
>
> >  2. To generate dialogs, a set of prompts are designed and organized in several types. It is unclear how they are selected to generate a dialog. Are they randomly chosen? Do they follow some fixed patterns? One would believe that the selection of prompts may influence the SpikeScore. Have you observed this phenomenon? How do you deal with such a variability due to the selection of prompts?
>
> **Response:** Thank you for this insightful question about prompt selection and its potential influence on SpikeScore. You're absolutely right that prompt design is crucial in any LLM evaluation work, and we appreciate you raising this important methodological consideration.
>
> To address your questions directly: our prompt selection strategy **is neither fully random nor entirely fixed**. We deliberately chose this hybrid approach for good reasons. A completely fixed sequence would make the evaluation unfair, as its effectiveness might only apply to specific scenarios or particular dialogue patterns, potentially overfitting to a narrow set of conversational contexts. On the other hand, pure randomization would compromise methodological comparability across different evaluation runs, making it difficult to ensure consistent experimental conditions.
>
> Specifically, our method establishes a structured progression framework while allowing controlled randomization within that framework. We design prompts along two axes (type and strength) and follow a weak-to-strong scheduling pattern across turns. Within each turn's strength level, we randomly select from the appropriate prompt pool. **Crucially, this random selection happens only once at the beginning of our experiments.** After establishing the prompt sequence through initial randomization, we ***fix*** that exact sequence for all subsequent evaluations to ensure fair comparison across different models and datasets. We provide comprehensive details of our prompt selection procedures and the full prompt library in **Appendix G**.
>
> We completely agree with your concern about robustness to prompt design variations, and evaluating this robustness is indeed meaningful. However, since the theoretical space of possible prompt designs is infinite, **we conducted focused ablation studies within our established framework**. More detailed and comprehensive experimental results have been included in **Appendix D.7 of the revised manuscript**. Specifically, we evaluated on Llama3.1-8B and Qwen3-8B models, training on CoQA and testing on Math datasets, reporting AUROC metrics. We varied the random seed that controls prompt selection (not the tensor computation seed) across multiple runs to assess sensitivity.
>
> To capture fine-grained behavioral changes, we also tracked **which conversation turn most frequently exhibited the maximum second-order difference across different test items**. This tells us where in the dialogue the sharpest fluctuations tend to concentrate under different prompt sequences.
>
> ### **CoQA → Math**
>
> #### **Llama3.1-8B**
>
> | Seed     | 42     | 422    | 4222   | 36     | 366    | 3666   |
> | -------- | ------ | ------ | ------ | ------ | ------ | ------ |
> | AUROC    | 0.7316 | 0.7293 | 0.7358 | 0.7417 | 0.7329 | 0.7410 |
> | PeakTurn | 4      | 3      | 6      | 4      | 4      | 5      |
>
> #### **Qwen3-8B**
>
> | Seed     | 42     | 422    | 4222   | 36     | 366    | 3666   |
> | -------- | ------ | ------ | ------ | ------ | ------ | ------ |
> | AUROC    | 0.7431 | 0.7504 | 0.7632 | 0.7511 | 0.7398 | 0.7455 |
> | PeakTurn | 7      | 6      | 3      | 4      | 5      | 5      |
>
> #### **Across-seed AUROC statistics**
>
> | Model        | Mean AUROC | Var(AUROC)\*  |
> | ------------ | ---------- | ------------- |
> | Llama3.1-8B  | 0.7354     | 2.58e-05      |
> | Qwen3-8B     | 0.7489     | 6.79e-05      |
>
> The results demonstrate that while different prompt selections do shift where the maximum second-order differences concentrate (Peak Turn varies across seeds), indicating that **prompt choice influences the model's output dynamics**, our evaluation metric **remains robust**. Since SpikeScore considers only the magnitude of the maximum second-order difference rather than its position, it effectively neutralizes the variability introduced by different prompt sequences. The AUROC values remain stable across seeds, confirming that our approach maintains consistent discriminative power despite prompt variations within our structured framework.

---

### Official Review · Reviewer_tmaW · 2025-11-01

**Soundness:** 3
**Presentation:** 3
**Contribution:** 3
**Rating:** 8
**Confidence:** 4

**Summary:**

This paper finds that in multi-turn conversations across different domains, subsequent responses triggered by hallucinations typically exhibit greater uncertainty fluctuations compared to factual responses. Based on this observation, this paper proposes a new metric called SpikeScore which quantifies fluctuations through maximum second-order differences. Experiments on multiple datasets demonstrate the effectiveness of the proposed SpikeScore in distinguishing between hallucinated and non-hallucinated responses across domains, and theoretical analysis further provides theoretical guarantees for its generalization capability.

**Strengths:**

1. This paper observes that in multi-turn conversations triggered by hallucinations, LLMs frequently engage in self-correction. By employing the maximum second-order difference to measure local fluctuations, this paper provides an effective and novel perspective for cross-domain hallucination detection.
2. Extensive cross-domain experiments have demonstrated the superiority of the proposed SpikeScore. Theoretical analysis additionally provides validation for its effectiveness in distinguishing between hallucinated and factual responses.

**Weaknesses:**

The claim of cross-domain generalization is not sufficiently supported. Although there are variations in tasks, such as question answering (CommonsenseQA, TriviaQA), reading comprehension (Belebele, CoQA), and mathematical reasoning (Math, SVAMP), the linguistic styles across these datasets may exhibit similarities. More cross-domain scenarios  are expected for evaluation.

**Questions:**

Please see the weaknesses.

---

> ### Author Response · Authors · 2025-11-20
> **Response to Reviewer tmaW’s Comments and Concerns (1/1)**
>
> Dear Reviewer,
>
> Thank you very much for your high recognition of our work! In response to your valuable suggestions, **we have added relevant experiments to the revised manuscript.**
>
> We wish you all the best in your research and life!
>
> Best regards,
> The Authors
>
> **Weaknesses**
> >  The claim of cross-domain generalization is not sufficiently supported. Although there are variations in tasks, such as question answering (CommonsenseQA, TriviaQA), reading comprehension (Belebele, CoQA), and mathematical reasoning (Math, SVAMP), the linguistic styles across these datasets may exhibit similarities. More cross-domain scenarios are expected for evaluation.
>
>
> **Response:** Thank you for raising this important concern about cross-domain generalization. We appreciate your observation that while we test on diverse tasks, the linguistic styles might still share similarities. To address this concern directly, **we conducted additional experiments in a fundamentally different domain: retrieval-augmented generation (RAG)**. More detailed experimental settings and results can be found in **Section 4.3 and Appendix D.6 of the revised manuscript**.
>
> RAG represents a distinct domain shift from our original benchmarks because it introduces several new dimensions: (1) the model must process externally retrieved documents rather than relying solely on parametric knowledge, (2) the linguistic style shifts from direct Q&A to document-grounded reasoning, and (3) the generation context includes potentially noisy or irrelevant retrieved passages that the model must navigate. This creates a substantially different distribution from standard QA or reading comprehension tasks.
>
> We specifically designed experiments using TriviaQA and RAGTruth in RAG settings. We built retrieval pipelines where questions trigger retrieval of top-4 passages via bge-large-en embeddings and Faiss indexing. The concatenated "question + retrieved contexts" is then fed to our test models. Crucially, all training-based methods (including SpikeScore) were trained only on non-RAG CoQA, making this a genuine test of cross-domain generalization.
>
> | Method             | Llama3.2-3B TriviaQA | Llama3.2-3B RAGTruth | Llama3.1-8B TriviaQA | Llama3.1-8B RAGTruth | Qwen3-8B TriviaQA | Qwen3-8B RAGTruth | Qwen3-14B TriviaQA | Qwen3-14B RAGTruth |
> | ------------------ | -------------------- | -------------------- | -------------------- | -------------------- | ----------------- | ----------------- | ------------------ | ------------------ |
> | Perplexity         | 0.5960               | 0.5936               | 0.6220               | 0.6151               | 0.6371            | 0.6284            | 0.6474             | 0.6420             |
> | Semantic Entropy   | 0.6330               | 0.6210               | 0.6173               | 0.6037               | 0.6492            | 0.6388            | 0.6745             | 0.6796             |
> | EigenScore         | 0.6426               | 0.6329               | 0.7085               | 0.6974               | 0.6571            | 0.6292            | 0.7064             | 0.6904             |
> | Lexical Similarity | 0.6248               | 0.6179               | 0.7102               | 0.6958               | 0.6375            | 0.6440            | 0.6542             | 0.6479             |
> | MM                 | 0.5724               | 0.5587               | 0.6020               | 0.5839               | 0.6166            | 0.6044            | 0.6291             | 0.6191             |
> | SEP                | 0.5312               | 0.4988               | 0.5779               | 0.5547               | 0.5161            | 0.5253            | 0.5368             | 0.5239             |
> | SAPLMA             | 0.6041               | 0.5874               | 0.5970               | 0.6011               | 0.6136            | 0.5904            | 0.6399             | 0.6338             |
> | ICR Probe          | 0.7532               | 0.7416               | 0.7682               | 0.7405               | 0.7495            | 0.7426            | 0.7841             | 0.7653             |
> | **SpikeScore(ours)**         | **0.7737**               | **0.7631**               | **0.7947**               | **0.8154**               | **0.8413**            | **0.8291**            | **0.8697**             | **0.8535**             |
>
> SpikeScore maintains strong performance in this entirely new domain, consistently outperforming all baselines despite the significant distribution shift and without any RAG-specific training. This demonstrates genuine cross-domain robustness beyond linguistic style variations, confirming that our method captures fundamental patterns of hallucination that transcend specific task formats or data distributions.

---

### Author Response · Authors · 2025-11-20
**General Response to the Area Chair and All Reviewers**

Dear Reviewers and AC,

We sincerely appreciate your valuable time and effort spent reviewing our manuscript. We are genuinely encouraged by the recognition from multiple reviewers who found our work to provide **"an effective and novel perspective"** (tmaW), demonstrate **"sound"** ideas with **"strong motivation from concrete observations"** (4ido), and tackle **"a realistic and impactful problem"** with a **"simple but powerful metric"** (ALVJ). This positive feedback has motivated us to further strengthen our work through careful revisions. In response to your constructive comments, we have enhanced the manuscript as follows:

1. Additional experiments in **retrieval-augmented generation (RAG)** are now included to demonstrate applicability beyond purely conversational settings and to substantiate cross-domain generalization claims (**Section 4.3, Appendix D.6**).
2. Ablation studies on **prompt-selection strategies** and their influence on SpikeScore stability have been incorporated (**Appendix G**).
3. **Direct AUROC evaluations** for training-free baselines without the leave-one-out protocol are provided for clarity (**Appendix D.5, Table 4**).
4. Robustness under adversarial **“polite-aligned” prompting**, which suppresses contradictions and may obscure internal fluctuations, has been evaluated (**Appendix D.8**).
5. **Comparative analysis on multi-turn consistency training** has been added to verify SpikeScore's robustness against models explicitly optimized for dialogue smoothness (**Appendix D.9**).

Meanwhile, we provide more clarifications of algorithmic details and experimental insights as follows:

1. In response to **Reviewers tmaW, ALVJ, and 7VBg’s** shared concern about cross-domain generalization and applicability beyond standard QA, we conducted RAG experiments on TriviaQA/RAGTruth and showed that SpikeScore trained on CoQA still transfers robustly, demonstrating that our signal is not tied to a single domain or interaction style.
2. In response to **Reviewer 4ido’s** concerns about the presentation of Tables 1–2 and the leave-one-out protocol versus in-domain results, we clarified the table structure and evaluation setup and added a separate table of direct (non–leave-one-out) AUROC for all training-free baselines to make their performance more interpretable.
3. In response to **Reviewer 4ido’s** questions about prompt selection and its impact on SpikeScore, we detailed our structured–randomized prompt framework and ran seed-based ablations, showing that although spike locations shift, AUROC remains stable, demonstrating robustness to reasonable prompt variation.
4. In response to **Reviewer ALVJ’s** question about robustness under adversarial “polite-aligned” prompting that suppresses contradictions, we added a stress-test experiment with an explicit “never contradict” instruction and showed that SpikeScore’s performance remains essentially unchanged and superior to baselines.
5. In response to **Reviewer ALVJ’s** questions on combining SpikeScore with chain-of-thought and extending to long-form or agentic workflows, we explained why variable CoT length makes a unified spike-based comparison problematic without distorting reasoning, and clarified that SpikeScore is a post-hoc detector that can sit on top of agentic/RAG-style pipelines without modifying their workflow.
6. In response to **Reviewer ALVJ’s** question about multilingual and multimodal extensions, we used Belebele results to show that our internal indicators behave consistently across languages in text-only settings, and argued that a true multimodal extension would require a new uncertainty framework and benchmarks with continuous hallucination scores, which we identify as future work.
7. In response to **Reviewer 7VBg’s hypothesis** that our effect might simply reflect multi-turn consistency training, we conducted a comparative experiment using models explicitly trained for consistency (e.g., Qwen-ConsistentChat). The results show that SpikeScore remains effective on these models, confirming that our metric captures internal uncertainty dynamics rather than just surface-level textual consistency artifacts.

In the revised manuscript, these updates are temporarily **highlighted in blue** for your convenience. We believe these revisions substantially strengthen the paper and clarify the significant contributions of SpikeScore.

Thank you very much,

Authors.

---

### Comment · Area_Chair_ig2j · 2025-11-27
**Rebuttal and Discussion Phase**

Dear Reviewers,

Thank you again for your time and effort in reviewing this paper. We are approaching the discussion deadline. I kindly ask you to review the rebuttal and continue the discussion so that we can reach a well-considered decision.

---

### Meta-Review · Area_Chair_zKbi · 2025-12-06

**Summary:**

The paper proposes "SpikeScore," a novel metric for Generalizable Hallucination Detection (GHD). The core insight is that hallucinated responses in Large Language Models (LLMs) exhibit higher uncertainty fluctuations (spikes) during multi-turn self-correction processes compared to factual responses. The authors provide a theoretical framework based on curvature of uncertainty and demonstrate effectiveness across multiple datasets. Initial reviews were generally positive (8, 6, 6), recognizing the novelty and effectiveness of the method. One reviewer (7VBg) initially recommended rejection (2), raising significant concerns about the theoretical depth and whether the metric was simply measuring multi-turn consistency training artifacts.

During the rebuttal, the authors provided extensive additional experiments, including: 1. Evaluations on RAG benchmarks (TriviaQA/RAGTruth) to prove applicability beyond open-ended chat. 2. Ablation studies comparing base models vs. models explicitly trained for multi-turn consistency, demonstrating that SpikeScore remains robust while surface-level consistency metrics fail. 3. Robustness checks against "polite-aligned" adversarial prompting. 4. Clarifications on prompt selection and experimental protocols.
Following these revisions, the dissenting reviewer (7VBg) was convinced by the new evidence and explicitly raised their score to 6.

**Reviewer Concerns:**

Addressed Concerns:

1. Theoretical Validity and Consistency Training (Reviewer 7VBg): The reviewer questioned whether SpikeScore merely detected a lack of consistency training. The authors successfully addressed this by comparing standard models against those fine-tuned for consistency (e.g., ConsistentChat). The results showed SpikeScore relies on internal uncertainty dynamics rather than surface-level coherence, resolving the primary objection.

2. Applicability to Non-Chat Workflows (Reviewers 7VBg, tmaW): Concerns that the method was limited to free-form chat were addressed by new experiments on Retrieval-Augmented Generation (RAG) pipelines, showing strong cross-domain generalization.

3. Robustness to Prompting Strategies (Reviewers 4ido, ALVJ): The authors demonstrated that the metric is robust to variations in prompt selection seeds and adversarial "polite" prompts that attempt to suppress contradictions.

4. Clarity of Results (Reviewer 4ido): Confusion regarding the leave-one-out protocol and table presentation was resolved with revised captions and additional direct AUROC tables.

**Reviewer Scores:**

1. Reviewer tmaW: 8 (The reviewer did not post post-rebuttal, but the initial score was high and concerns were addressed).

2. Reviewer 4ido: 6 (The reviewer did not post post-rebuttal, but clarifications on methodology likely solidified this positive score).

3. Reviewer ALVJ: 6 (The reviewer did not post post-rebuttal, but the robustness experiments addressed their specific questions).

4. Reviewer 7VBg: 6 (Explicitly raised from 2 to 6 in the discussion thread following the consistency ablation results).

---

### Decision · Program_Chairs · 2026-01-26

Accept (Poster)